# Many Circuits, One Mechanism: Input Variation and Evaluation Granularity in Circuit Discovery

## Abstract

Circuit discovery methods identify subgraphs that explain specific model behaviors, and structural differences between discovered circuits are commonly interpreted as evidence of distinct mechanisms. We test this assumption by varying input statistics while holding the task fixed, and show that the resulting structural differences exhibit apparent specialization but do not correspond to functional differences, a pattern we term *phantom specialization*. Using Literal Sequence Copying across four token-frequency bands in five Pythia models (70M–1.4B), we extract 75 circuits and find that structurally distinct circuits implement the same computation: band-specific edges transfer broadly across bands, a shared core recovers at least 99% of circuit performance, and causal interchange interventions confirm that internal representations are interchangeable across frequency bands. Repeated extractions within the same frequency band further suggest that discovery algorithms sample from an equivalence class of valid subgraphs rather than recovering a unique mechanism. Standard evaluation practice obscures this pattern: source-level evaluation inflates apparent faithfulness, while edge-level evaluation reveals the many-to-one mapping from structure to function. Our results show that structural differences between circuits are not sufficient evidence for distinct mechanisms, and that exposing this requires edge-level evaluation and cross-condition transfer tests.[1]

## 1 Introduction

Mechanistic interpretability seeks to reverse-engineer the internal computations of neural networks, much as one would analyze a compiled program to recover its high-level algorithms (Olah, 2022; Elhage et al., 2021). An important object of study in this field is the *circuit*: a subgraph of a model's computational graph that implements a specific behavior (Olah et al., 2020; Conmy et al., 2023; Rai et al., 2024). In this graph, nodes typically correspond to attention heads and MLP blocks, and edges represent the flow of information between components through the residual stream. Early circuits were identified manually through iterative causal interventions, ablating or patching individual components and tracing their effects back from the output, for behaviors such as in-context induction (Olsson et al., 2022), indirect object identification (IOI) (Wang et al., 2023), and greater-than comparison (Hanna et al., 2023). However, this process is slow and difficult to scale. To address this, automated methods have been proposed: ACDC iteratively prunes edges via activation patching (Conmy et al., 2023); attribution-based methods such as EAP-IG approximate edge importance through gradients (Syed et al., 2024; Hanna et al., 2024; Kramár et al., 2024); and edge-masking approaches formulate discovery as differentiable optimization (Bhaskar et al., 2024; Yu et al., 2025). These methods are grounded in the causal abstraction framework, which provides a theoretical foundation for mechanistic interpretability and unifies activation and path patching within a common causal language (Geiger et al., 2025). Despite their differences, they share a common goal: to identify *a* circuit responsible for a given behavior.

Nevertheless, a growing body of work suggests that this shared goal rests on a fragile premise: the assumption that, for a given model and task, a single stable circuit exists to be discovered. This is problematic

---

[1]Our code and data are available at: `https://anonymous.4open.science/r/phantom-specialization-7BD8`.

because treating a discovered circuit as the circuit for a task can obscure the possibility that different extraction choices, prompts, or input distributions reveal different but equally valid mechanisms. Landmark results reinforce this implicit assumption by reporting circuits as singular objects: "the IOI circuit" (Wang et al., 2023), "the induction circuit" (Olsson et al., 2022). Yet circuit discovery already faces several well-known challenges, including fragile faithfulness evaluations (Miller et al., 2024), instability under changes in extraction hyperparameters (Méloux et al., 2025b; Conmy et al., 2023), redundant backup behavior (McGrath et al., 2023), and fundamental non-identifiability; more broadly, recent surveys have also highlighted additional open problems for the field (Sharkey et al., 2025). Méloux et al. (2025a) exhaustively enumerate candidate explanations in toy models and find that multiple structurally distinct circuits, interpretations, and causal alignments can satisfy current mechanistic interpretability criteria simultaneously. Moreover, uit de Bos & Garriga-Alonso (2024) show that the IOI circuit can fail systematically on benign inputs and certain input types, demonstrating that circuit behavior is not uniform across inputs even at evaluation time. These findings raise a further question: if discovered circuits are sensitive to both the extraction method and the evaluation inputs, might they also be sensitive to the input distribution used for extraction? Motivated by the *universality hypothesis* (Olah et al., 2020), existing work has tested circuit consistency across models and settings, finding recurring motifs in toy settings and across scales (Chughtai et al., 2023; Tigges et al., 2024), across languages (Ferrando & Costa-Jussà, 2024), and across tasks (Merullo et al., 2024; Mondorf et al., 2025), often relying primarily on structural overlap to assess similarity (Hanna et al., 2024). Recent work has begun to probe circuit consistency within a single model: Franco et al. (2026) show that per-prompt circuits for IOI cluster into families, and several studies document shared components across related tasks, as well as substantial overlap across reasoning subtasks within a task (Merullo et al., 2024; Lan et al., 2024; Mondorf et al., 2025; Dutta et al., 2024). However, these comparisons vary either the task or the prompt structure, leaving open whether structural differences arise even when only the input *statistics* change while the task itself is held fixed; crucially, it also remains unknown whether such differences are functionally meaningful. We show that this gap matters: when we vary token frequency while holding the task fixed, circuit discovery recovers structurally distinct circuits that nonetheless implement the same computation.

We approach the question of whether circuit structure depends on input statistics along two axes: token frequency and evaluation granularity. Prior work has shown that language model behavior depends systematically on token frequency, that is, how often a token appears in the pretraining corpus, at multiple levels, including contextual representations (Zhou et al., 2021) and numerical reasoning behavior (Razeghi et al., 2022), and even on tasks with no semantic content (Niu et al., 2025). This makes token frequency a natural probe for our first axis: it can be measured precisely from the training corpus, yet varies the input without changing the task definition. We use the Literal Sequence Copying (LSC) task from Niu et al. (2025), a non-semantic copying task in which the model must reproduce a token from an earlier occurrence in the sequence. For example, given the prompt `A B C D ... A B C`, the model must predict `D`. Because LSC minimizes semantic confounds and isolates a narrow copying behavior, it provides a comparatively controlled setting for testing whether circuit structure changes even when the task remains fixed. To systematically test whether input statistics drive circuit differences, we partition the Pythia vocabulary (Biderman et al., 2023) into four frequency bands based on token frequency in The Pile (Gao et al., 2021), extract circuits with ACDC across five model scales (70M to 1.4B), and run three independent extractions per frequency band. This yields five conditions: four frequency bands plus a frequency-weighted control, and 75 circuits total. We compare these circuits along structural, functional, and representational axes (Figure 1). Our second axis is evaluation granularity. Discovered circuits can be evaluated at two levels: *source-level* evaluation preserves all outgoing edges from any node that contributes at least one circuit edge, while *edge-level* evaluation preserves only the specific edges selected by the discovery algorithm. More broadly, the idea that causal analysis may depend on the chosen level of description (Noble, 2011; Hoel et al., 2013) motivates testing whether the causal picture changes across these two granularities. We report the following findings:

$\mathcal{F}$1 Circuits extracted from different frequency bands are structurally distinct, with low-frequency circuits systematically larger.

$\mathcal{F}$2 These structural differences do not reflect functional specialization: band-specific edges transfer broadly across all frequency bands, and a shared core suffices for nearly all circuit performance.

$\mathcal{F}3$ Extraction is noisy across repeated runs, yet this noise is functionally irrelevant: structurally different draws implement the same computation.

$\mathcal{F}4$ Source-level evaluation inflates apparent circuit accuracy by up to 0.57 points relative to edge-level evaluation, masking the absence of specialization and obscuring the many-to-one mapping from structure to function.

These findings lead us to coin the term *phantom specialization*: structural divergence between discovered circuits that does not correspond to functional specialization. It reflects a many-to-one mapping from circuit structure to function: when multiple edges are individually dispensable because computation can route through redundant or compensatory pathways (McGrath et al., 2023), and more broadly because many parameterizations can implement the same function (Bushnaq et al., 2024), greedy pruning must choose which to keep, and small differences in input statistics tip these choices differently, producing structurally distinct circuits that implement the same computation. More broadly, discovery algorithms may sample from an equivalence class of structurally distinct yet functionally interchangeable subgraphs, a form of *circuit degeneracy* analogous to degeneracy in biological systems (Edelman & Gally, 2001). The band-specific edges that ACDC recovers are best understood not as adaptations but as byproducts of the extraction process, that is, as *spandrels* in the sense of Gould & Lewontin (1979), because they boost performance generically across all conditions rather than encoding band-specific computation. Crucially, phantom specialization is detectable only at edge-level granularity. Source-level evaluation collapses this equivalence class of functionally interchangeable circuits into a single apparently faithful macro-state, reminiscent of causal emergence (Hoel et al., 2013), making the phantom invisible ($\mathcal{F}4$).

Taken together, our results establish a many-to-one mapping from circuit structure to circuit function: structurally distinct circuits can implement the same computation, and discovery algorithms can sample from an equivalence class of functionally interchangeable subgraphs. Whether this degeneracy is visible depends on evaluation granularity: source-level evaluation systematically obscures it, whereas edge-level evaluation reveals it. More broadly, this granularity dependence echoes the idea that causal structure may manifest differently at different levels of description (Noble, 2011; Hoel et al., 2013). For circuit discovery, the implication is that structural differences between circuits should not be interpreted as evidence of distinct mechanisms without cross-condition transfer tests. In our setting, edge-level evaluation is the more informative faithfulness metric, and repeated extractions with majority-vote aggregation provide a practical way to identify stable structure. While broader limitations of circuit discovery pipelines have been flagged as open problems (Sharkey et al., 2025), and non-uniqueness has been demonstrated in toy settings (Méloux et al., 2025a), our results show that the same phenomenon persists across five Pythia model scales, from 70M to 1.4B parameters, and across all frequency conditions we study. A single discovered circuit should therefore not be treated as uniquely identifying the underlying mechanism.

Section 2 reviews background and related work. Section 3 describes the experimental setup. Section 4 presents evidence for apparent frequency-dependent specialization. Section 5 tests whether the observed differences reflect genuine specialization, Section 6 discusses implications and limitations, and Section 7 concludes.

## 2 Background and Related Work

### 2.1 Circuit Discovery and Evaluation

Circuit discovery aims to identify computational subgraphs, or *circuits*, that explain how a neural network implements a specific behavior (Olah et al., 2020; Elhage et al., 2021). Throughout this paper, we use *circuit* as an operational term: a sparse, task-relevant subgraph of components and connections that recovers substantial task performance. This definition is intentionally method-relative: discovered circuits depend on how the computational graph is defined, how edges are selected, and how faithfulness is measured. In a transformer, the residual stream serves as a shared communication channel through which attention heads and MLP blocks read and write, which can be represented as a directed acyclic graph (DAG) of information flow (Elhage et al., 2021). Nodes in this graph correspond to attention heads and MLP blocks; edges

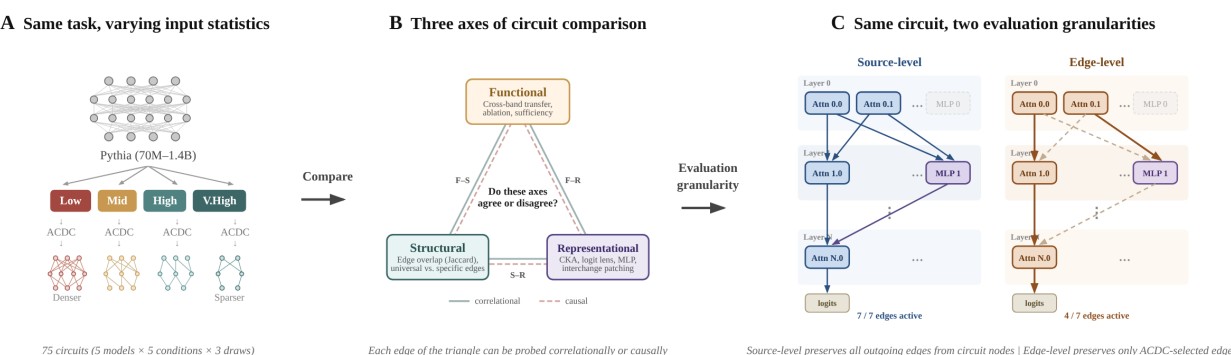

Figure 1: Experimental design and analytical framework. **(A)** ACDC circuit discovery on Literal Sequence Copying across four token-frequency bands plus a control condition in five Pythia models (70M–1.4B) produces 75 structurally distinct circuits (5 models × 5 conditions × 3 draws). **(B)** Circuits are compared along three axes (structural, functional, and representational), connected by both correlational and causal analyses. **(C)** The same circuit can be evaluated at two granularities: source-level preserves all outgoing edges from circuit nodes; edge-level preserves only the specific edges the discovery algorithm selected.

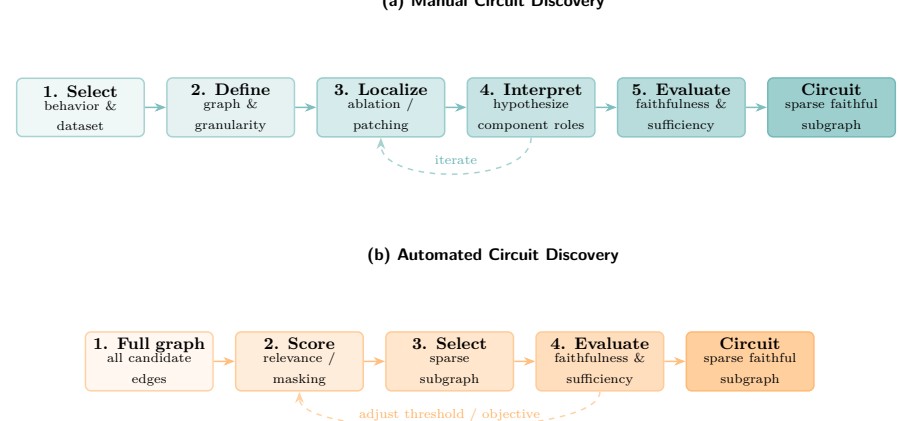

Figure 2: Two paradigms for circuit discovery. **(a)** Manual discovery follows a five-step workflow (Rai et al., 2024): select a target behavior, define the graph and granularity, localize important components via intervention, interpret their roles, and evaluate the result, iterating between localization and interpretation until a stable working hypothesis emerges (Olsson et al., 2022; Wang et al., 2023; Hanna et al., 2023). **(b)** Automated methods start from the full computational graph containing all candidate edges, score each edge's relevance through activation patching (Conmy et al., 2023), gradient attribution (Syed et al., 2024), or differentiable masking (Bhaskar et al., 2024), select a sparse subgraph, and evaluate faithfulness, iterating over the sparsity threshold or objective until the result stabilizes.

represent that one component reads from the output of another through the residual stream. A circuit is a task-relevant subgraph of this DAG (Conmy et al., 2023; Rai et al., 2024).

Most circuit discovery methods build on *activation patching* (Vig et al., 2020; Meng et al., 2022): replacing a component's activation with one from a different input and observing whether the model's output changes in the predicted way. The causal abstraction framework (Geiger et al., 2025) subsequently formalized this practice, grounding interchange interventions in causal theory. Activation patching involves three forward passes: (1) a *clean run* on the original prompt, caching all intermediate activations; (2) a *corrupt run* on a modified input from which the signal relevant to the target behavior has been removed; and (3) a *patch run* in which the activation of a specific component is swapped between runs while the rest of the computation proceeds normally. Patching can be applied in two directions: noising (replacing a clean component with

its corrupt value to test necessity) and denoising (restoring a corrupt component to its clean value to test sufficiency) (Rai et al., 2024; Heimersheim & Nanda, 2024). The choice of corruption method (zero, mean, or resampling ablation) can materially affect patching results, and may therefore change the recovered circuit for the same model and task (Heimersheim & Nanda, 2024; Méloux et al., 2025b).

For an extended treatment of circuit discovery methods and evaluation, a brief review of activation patching for unfamiliar readers, as well as a detailed account of the corruption method, see Appendix A.

Automated methods algorithmically search for sparse, task-relevant subgraphs (Figure 2b). Conmy et al. (2023) introduced ACDC, which automates circuit discovery by iteratively pruning edges whose removal least affects the model's output under activation patching. Subsequent methods fall into several families: attribution-based methods approximate edge effects via gradient-based scores, including EAP (Syed et al., 2024), EAP-IG (Hanna et al., 2024), and AtP* (Kramár et al., 2024); on the MIB benchmark, EAP-IG-inputs achieves the best average performance, while related attribution-based methods remain competitive (Mueller et al., 2025); edge-pruning and masking approaches formulate discovery as differentiable optimization over computation-graph masks, including Edge Pruning (Bhaskar et al., 2024) and DiscoGP (Yu et al., 2025); and single-forward-pass methods such as ACC++ (Franco et al., 2026) trace information flow without patching. Despite differences in search procedure, these methods are typically used to produce a single explanatory subgraph for a target behavior, a framing whose limitations have recently become more explicit (Sharkey et al., 2025; Méloux et al., 2025a).

This assumption has received limited scrutiny. Méloux et al. (2025b) show that gradient-based attribution scores exhibit high variance across inputs; Conmy et al. (2023) note that ACDC's behavior is sensitive to threshold and metric choice; and Sharkey et al. (2025) emphasize broader open problems in current circuit discovery pipelines. Franco et al. (2026) offer a complementary perspective: using ACC++, a single-forward-pass tracing method based on SVD of QK matrices rather than activation patching, they show that per-prompt circuits for IOI cluster into recurring "prompt families," suggesting that apparent instability in task-level circuit discovery partly reflects genuine prompt-specific subcircuit variation. Our work provides a systematic empirical test of this concern from the input-distribution side: by running multiple independent extractions on distinct frequency bands within the same task, we show that the instability and non-uniqueness suggested by prior work can manifest as structurally distinct but functionally interchangeable circuits, precisely the scenario that makes structural comparison unreliable.

Circuit evaluation raises complementary concerns. Faithfulness can be operationalized via logit difference (Wang et al., 2023), KL divergence (Conmy et al., 2023; Zhang & Nanda, 2024), or path patching (Goldowsky-Dill et al., 2023), but aggregate faithfulness can mask per-input failures (Miller et al., 2024), and many published circuits do not pass strict sufficiency-and-necessity tests (Shi et al., 2024; uit de Bos & Garriga-Alonso, 2024).

A key methodological distinction is evaluation granularity. Most circuit studies evaluate at the level of components or source nodes, preserving all outgoing connections from selected components rather than only the specific discovered edges. In this paper, we refer to this as *source-level* evaluation, and reserve *edge-level* evaluation for tests that preserve only the specific connections identified by the discovery algorithm. This distinction matters because circuit faithfulness scores can be sensitive to ablation methodology (Miller et al., 2024). We show that source-level evaluation systematically inflates apparent circuit accuracy (Section 5.2.1) and introduce cross-condition transfer as a complementary evaluation axis.

## 2.2 Circuits Across Conditions

The mechanistic interpretability literature has produced detailed accounts of circuits for specific tasks, including Indirect Object Identification (Wang et al., 2023), in-context copying via induction heads (Olsson et al., 2022), greater-than comparison (Hanna et al., 2023), copy suppression (McDougall et al., 2024), successor operations (Gould et al., 2024), and extractive question-answering (Basu et al., 2025), among others. A recurring question is whether the same motifs appear across models. Olah et al. (2020) proposed the *universality hypothesis*; subsequent empirical work found consistent IOI circuits across Pythia scales (Tigges et al., 2024), cross-lingual agreement circuits in Gemma 2B (Team et al., 2024a; Ferrando & Costa-Jussà, 2024), and recurring successor heads across model families (Gould et al., 2024).

Several studies have begun comparing circuits across conditions, but each addresses a different slice of the problem and leaves specific gaps that our design targets. Hanna et al. (2024) argued that structural overlap alone is insufficient for comparing circuits, advocating faithfulness-based comparison; however, they compare circuits found by different *methods*, not circuits found on different *inputs*, and do not quantify the source-level versus edge-level evaluation gap. We organize the remaining empirical comparisons by the axis of variation.

**Across tasks.**  Merullo et al. (2024) showed that IOI and Colored Objects circuits share ∼78% of attention heads in GPT-2 Medium (Radford et al., 2019), but used source-level (head) granularity, a single extraction per task, no systematic transfer matrix, and no statistical controls. Dutta et al. (2024) found that within chain-of-thought reasoning in Llama-2 7B (Touvron et al., 2023), the same induction-like attention heads serve decision-making, copying, and inductive reasoning subtasks, with heads identified for induction alone retaining >90% accuracy across all subtask types; however, they did not test whether the *structural differences* between subtask circuits are functionally meaningful. Lan et al. (2024) identified shared sub-circuits for sequence continuation tasks (numerals, number words, months) in GPT-2 Small and Llama-2 7B with partial cross-task transfer, but at node-level granularity and without multiple extractions or causal confirmation. Mondorf et al. (2025) found both shared and task-specific substructure for string-edit operations, again at node level and across *different tasks* rather than within a single task.

**Across models, scales, and training time.**  Tigges et al. (2024) tracked IOI and successor circuits across 300B tokens of Pythia training and across scales up to 2.8B, finding algorithmic stability despite component-level fluctuations; however, they vary *time and scale*, not input distribution, and use source-level (head) granularity without testing whether the structural variation they document is functionally irrelevant. Ferrando & Costa-Jussà (2024) found consistent subject–verb agreement circuits across English and Spanish in Gemma 2B, including a language-independent number direction, but studied a single model with a single extraction per language and no edge-level evaluation.

**Within a single task.**  Most directly, Franco et al. (2026) showed that even within a single task (IOI), per-prompt circuits are not unique: different prompt templates induce systematically different circuits in GPT-2 Small, Pythia-160M, and Gemma-2 2B (Team et al., 2024b). Crucially, however, their prompt-template variation changes the *task structure* (e.g., name order), and they report systematically different prompt-family mechanisms without testing cross-condition transfer or functional equivalence. Sun (2025) showed that circuit stability across input instances predicts generalization, but did not test whether structural *instability* between conditions translates to functional difference. Mahaut & Franzon (2025) found that superficially similar repetition behaviors can arise from qualitatively different internal processes: an instance of genuine specialization that contrasts with the phantom we identify.

Across this body of work, three specific gaps persist. First, most comparisons vary the *task* or *prompt structure*, not the input *statistics* within a fixed task; the prompt-family analysis of Franco et al. (2026) also varies prompt structure alongside surface form. Second, to our knowledge, all studies cited above evaluate at source or node level; none systematically compare edge-level with source-level evaluation or quantify the resulting inflation. Third, none combine multiple independent extractions, cross-condition transfer matrices, and causal confirmation to distinguish genuine specialization from extraction artifacts. Our experimental design addresses all three gaps simultaneously.

## 2.3   Token Frequency and Model Internals

Token frequency, governed by Zipfian distributions (Zipf, 1949), is known to affect language models at multiple levels of analysis. At the representation level, Zhou et al. (2021) showed that low-frequency words occupy more identifiable but less diverse regions of BERT's (Devlin et al., 2019) embedding space, introducing systematic distortions in similarity metrics. Merullo et al. (2025) found that linear representations of factual relations in OLMo (Groeneveld et al., 2024) and GPT-J (Wang & Komatsuzaki, 2021) form only when subject-object co-occurrences exceed a frequency threshold, linking representational structure directly to pretraining statistics. At the behavioral level, Razeghi et al. (2022) demonstrated that few-shot numerical reasoning accuracy correlates with operand frequency in pretraining data, and Niu et al. (2025) showed that induction-head copying degrades for rare tokens even though the task logic is frequency-invariant.

Pinto et al. (2024) revealed that weight decay disproportionately depreciates low-frequency tokens, a bias invisible in aggregate training loss. At the neuron level, Liu et al. (2025) showed that rare-token processing in Pythia and GPT-2 emerges through distributed specialization: coordinated but spatially scattered MLP neurons within the shared architecture, with no evidence of dedicated modules or attention routing. However, all of this work examines frequency effects at the level of representations, outputs, or individual neurons, not at the level of the computational graph. Our work extends to the circuit level, using token frequency as a controlled perturbation axis for circuit discovery and testing whether observed structural differences reflect genuine functional specialization.

## 2.4 Redundancy, Non-Uniqueness, and Multiple Valid Mechanisms

Several lines of evidence suggest that transformer computations are not implemented by a single, uniquely determined pathway. Méloux et al. (2025a) provide the most systematic demonstration of this problem: using small MLPs trained on Boolean functions, they exhaustively enumerate all valid explanations under both circuit-first and algorithm-first MI strategies and find overwhelming non-identifiability: multiple circuits with zero error, multiple interpretations per circuit, and multiple causally aligned algorithms per network, with the number of valid explanations growing dramatically with model size. McGrath et al. (2023) described the *Hydra effect*: when important attention heads are ablated, backup heads compensate by increasing their contribution, indicating built-in redundancy. Wang et al. (2023) observed similar backup behavior in the IOI circuit, and Ortu et al. (2024) showed that factual recall and counterfactual reasoning engage competing mechanisms that dynamically trade off depending on context. Dutta et al. (2024) documented this redundancy at scale in Llama-2 7B: during chain-of-thought reasoning, multiple parallel pathways simultaneously write the answer token to the output residual stream, each collecting information from different segments of the context (i.e., generated CoT, question, and few-shot examples), a direct instance of functionally interchangeable parallel circuits in a production-scale model. At the parameter level, Bushnaq et al. (2024) connected this redundancy to degeneracy in the loss landscape, showing that mechanistically distinct parameter configurations can achieve equivalent loss. This parallels the biological concept of *degeneracy*: structurally different elements performing the same function (Edelman & Gally, 2001).

An alternative to edge-level circuit discovery is to define circuits over learned interpretable features rather than architectural components, using sparse autoencoders (Marks et al., 2025) or transcoders (Dunefsky et al., 2024). Such *feature-level* circuits may resolve ambiguities arising from polysemantic components, precisely the setting in which phantom specialization is most likely to occur. Whether feature-level circuits exhibit the same non-uniqueness is an important open question (Section 7).

Our notion of *phantom specialization* extends these observations from individual components or parameters to complete circuits: under controlled input variation, the discovery algorithm samples from an equivalence class of structurally distinct but functionally interchangeable circuits. This is distinct from the Hydra effect (compensation after intervention) and from loss-landscape degeneracy (parameter-level non-uniqueness). Franco et al. (2026) provide independent corroboration from a different method and perturbation axis: their ACC++ traces reveal that the same canonical IOI component (e.g., the name-mover head) can carry entirely different internal signals across prompt templates while preserving its high-level role, a signal-level analog of our finding that structurally distinct circuits implement identical functions. Our work can also be seen as an empirical, large-scale answer to the open question posed by Méloux et al. (2025a): they ask whether non-identifiability persists beyond toy models, and our findings in Pythia models up to 1.4B parameters suggest that it does.

# 3 Experimental Setup

## 3.1 Data and Task

**Frequency bands.** We measure token frequency in the Pythia training corpus, express it as occurrences per million tokens, and partition the core range (percentiles 1–99, excluding sparse tails) into five equal-width bands on a $\log_{10}$ scale, so that each band spans the same multiplicative frequency ratio (approximately 3.5×), keeping within-band variation below 4× while retaining at least 500 tokens per band (subsection B.3). The

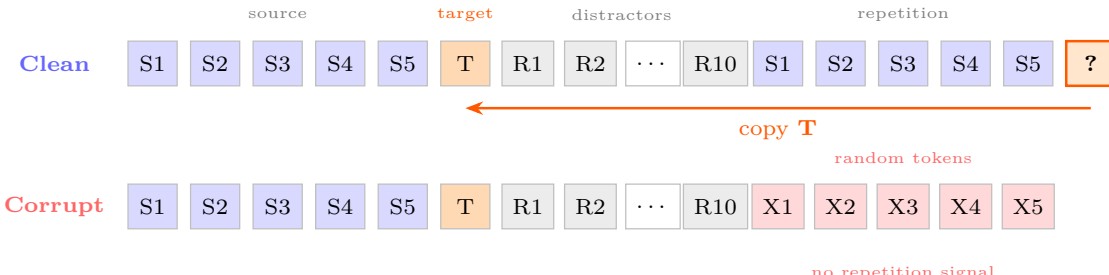

Figure 3: LSC sequence structure and corruption. **Top (clean):** the model observes a source prefix $S_{1-5}$ followed by target $T$, a distraction segment $R_{1-10}$, and a repetition of the source prefix, then must predict $T$ at the final position. All 16 tokens are sampled from the same frequency band. **Bottom (corrupt):** the repeated source prefix is replaced with random tokens $X_{1-5}$ from the same frequency band, destroying the repetition signal while leaving all other positions unchanged.

final design comprises five core bands (*very_low* through *very_high*), two exploratory tail bands, and one frequency-weighted *control* condition (Table 13).

**Confound control.** Token frequency co-varies with other token properties: rare tokens tend to be longer and more often capitalized (e.g., proper nouns), either of which could independently affect model behavior. To isolate frequency as the primary axis of variation, we restrict the token pool to the 26,863 tokens that represent standalone words (identified by the BPE space prefix) and contain only unaccented Latin letters (a–z/A–Z), excluding subword fragments, digits, punctuation, and non-Latin scripts (see subsection B.2 for the full filtering pipeline). Within this pool, character length and capitalization still vary across frequency bands; both are controlled at the task design stage by restricting to lowercase tokens and length-matching across bands (Section 3.1; subsection B.4). Although character length is not directly visible to the model once a word is represented as a single token, it remains correlated with other token-level properties that may vary across frequency bands. We therefore length-match across bands as a conservative proxy control to better isolate frequency as the manipulated variable. Our filtering and matching steps isolate token frequency from other properties that co-vary with rarity in natural text, so the conclusions should be interpreted as evidence about controlled frequency variation rather than the full natural bundle of rare-token effects.

**Task: Literal Sequence Copying.** We use *Literal Sequence Copying* (LSC), the non-semantic induction task of Niu et al. (2025), who showed that copying accuracy degrades for rare tokens even though the task requires only pattern matching. Each input sequence (Figure 3) contains a five-token source prefix $S_{1-5}$, a target token $T$, a ten-token distraction segment $R_{1-10}$, and a repetition of the source prefix. The model's task is to predict $T$ by copying it from an earlier occurrence of the pattern (Figure 3); no instructions or demonstrations are provided, so the model must rely on its learned copying mechanism through autoregressive next-token prediction. All 16 unique tokens in each sequence (5 source tokens, 1 target, 10 distractors; the 5 source tokens are repeated, yielding 21 positions) are sampled randomly without replacement from the same frequency band, making accidental sequential patterns negligible. Because LSC is non-semantic, frequency can vary across examples without changing the task itself.

To control the confounds identified above, we restrict LSC pools to lowercase tokens and length-match across bands, yielding 703 tokens per band for four core bands (low through very_high). The very_low band (97 lowercase tokens after filtering) is too small for reliable length-matching and is excluded from both the matched pools and circuit discovery. The control condition draws from all four matched bands (2,812 tokens total), weighting each token by its pretraining frequency so that the resulting sequences reflect the model's natural training distribution. Circuit discovery thus operates on five conditions: four frequency bands plus control (subsection D.9 verifies that the control circuit does not differ significantly from frequency-specific circuits).

We generate 1,500 sequences per band (70/15/15 train/validation/test) across three independent *draws*: each draw samples a fresh set of tokens from the pool and generates new sequences, yielding both a new dataset and (after extraction) a new circuit. Generation procedure and seeds in subsection B.5.

## 3.2 Models

We use five models from the Pythia suite (Biderman et al., 2023): Pythia-70m, 160m, 410m, 1b, and 1.4b (Table 1, left). Pythia is well suited for this study because it was trained exclusively on The Pile and its tokenizer is publicly available, enabling precise frequency measurements that match the model's training distribution. Token-frequency extraction details in subsection B.1. We use the standard (undeduped) variant throughout, as the deduped models performed worse on LSC in preliminary evaluations. The five scales span a $20\times$ parameter range, allowing us to test whether circuit-level patterns hold across model capacity. All larger models achieve near-ceiling accuracy ($\geq 93\%$ top-1; Section 4.1), confirming that the task is well within their capacity. Pythia-70m is a boundary case: its base accuracy is low (30–67%) and several metrics diverge from the four larger models. We include it to examine whether phantom specialization extends to low-capacity models, but qualify results throughout when Pythia-70m diverges from the larger models.

## 3.3 Circuit Extraction

We discover circuits using the ACDC algorithm (Conmy et al., 2023), as implemented in the AutoCircuit library (Miller et al., 2024). We use AutoCircuit rather than the original ACDC codebase because the latter exhibited metric collapse on Pythia models during preliminary testing; we refer the reader to section C.2 for the documented failure modes. ACDC was preferred over EAP and EAP-IG (Syed et al., 2024; Hanna et al., 2024), which rank edges independently by gradient-based scores and select the top-$k$, potentially producing disconnected edge sets that do not form functional circuits: at ACDC-matched circuit sizes, EAP-IG achieves only 2–24% of base accuracy for models $\geq 160M$, versus 79–99% for ACDC (see subsection C.9 for the full comparison). ACDC requires paired clean and corrupted inputs. For LSC, the corrupted input replaces the repeated source prefix with random tokens from the same frequency band, destroying the repetition signal while leaving all other positions unchanged (Figure 3; subsection C.1). Edges whose resample-ablated activations change the output by less than a threshold $\tau$ are pruned. Formally, let $\mathcal{G} = (V, E)$ denote the full computational graph, where $V$ comprises attention heads and MLP blocks and $E$ the connections between them. Given a clean input $x$ and its corrupted counterpart $\tilde{x}$, *resample ablation* of edge $(u, v)$ replaces the activation sent from $u$ to $v$ with the value $u$ computed on $\tilde{x}$. For a set of retained (active) edges $C \subseteq E$, let $p_C(\cdot \mid x, \tilde{x})$ denote the model's output distribution when edges in $C$ carry clean activations and all others carry resample-ablated activations. ACDC iteratively prunes each edge $e$ from the current circuit $C$ for which

$$\mathrm{KL}\big(p_{\mathrm{base}}(\cdot \mid x) \,\big\|\, p_{C \setminus \{e\}}(\cdot \mid x, \tilde{x})\big) < \tau, \tag{1}$$

where $p_{\mathrm{base}} \equiv p_E$ is the unablated model. We use factorized patching, which allows ACDC to prune individual edge connections rather than all outputs of a component at once (subsection C.2). Divergence is measured at the final sequence position.

The pruning threshold $\tau$ controls the minimality-faithfulness trade-off. Since no reference circuit for LSC on Pythia exists, we select $\tau$ from stand-alone circuit properties. Crucially, we perform threshold selection on the *control* band rather than on any frequency-specific band: because the control condition samples tokens with pretraining-frequency weights, it is agnostic to the frequency partition and prevents overfitting $\tau$ to a particular band's characteristics. The selected $\tau^*$ is then applied uniformly to all five conditions, so any structural differences between the resulting circuits cannot be attributed to per-band threshold tuning. We sweep 11 log-uniformly spaced thresholds ($10^{-2}$ to $10^{-6}$) on the control band, training on 256 examples and evaluating on the validation split (225 examples). The Pareto frontier over edge fraction and KL divergence (Figure 15) identifies non-dominated operating points, from which we select one threshold $\tau^*$ per model (Table 1; per-model selection criterion in subsection C.3). All selected points achieve 0% ablation accuracy, confirming that the discovered edges are necessary for the task. Adjacent thresholds (2–3$\times$ range in circuit size) yield at most 5.9 percentage points change in transfer efficiency (see subsection C.8), confirming that our conclusions are robust to threshold selection. The full extraction pipeline requires approximately 736

Table 1: Model architecture and selected ACDC thresholds. The left columns show fixed architectural properties; the right columns show circuit extraction results. Pythia-1b has fewer layers and heads than Pythia-410m despite having more parameters, resulting in a much smaller computational graph (10K vs. 81K edges) and a higher edge fraction. See Table 14 in the appendix for additional metrics including circuit accuracy, retention, and ablation accuracy.

| Model | Architecture | | | | Circuit extraction | | |
|---|---|---|---|---|---|---|---|
| | Layers | Heads | $d_{\text{model}}$ | Graph edges | $\tau^*$ | Edge % | KL |
| Pythia-70m | 6 | 8 | 512 | 1,324 | $1.58 \times 10^{-3}$ | 32.9 | 0.24 |
| Pythia-160m | 12 | 12 | 768 | 11,467 | $6.31 \times 10^{-4}$ | 12.2 | 0.28 |
| Pythia-410m | 24 | 16 | 1024 | 80,581 | $2.51 \times 10^{-4}$ | 4.3 | 0.29 |
| Pythia-1b | 16 | 8 | 2048 | 10,009 | $1.58 \times 10^{-3}$ | 9.4 | 0.48 |
| Pythia-1.4b | 24 | 16 | 2048 | 80,581 | $6.31 \times 10^{-4}$ | 2.6 | 0.50 |

GPU-hours, which constrains the number of independent draws and threshold settings we can explore. Our robustness checks confirm stability across adjacent thresholds and three draws per frequency band, but additional extractions could further characterize the equivalence class of valid subgraphs.

With $\tau^*$ fixed per model, we run ACDC across all five conditions (four frequency bands plus control) and three draws (75 circuits total),[2] evaluating each circuit on the held-out test split (225 examples) using four standard metrics[3] (Section 3.4). The validation split is used exclusively during threshold selection, preventing information leakage between the two phases. Each circuit is also evaluated on the test splits of the other four bands, yielding a $5 \times 5$ cross-band transfer matrix. As a robustness check, we compare each circuit against 100 random baselines, each containing the same number of edges sampled uniformly from the full graph (without connectivity constraints). Every discovered circuit outperforms all 100 random baselines (subsection C.6).

### 3.4 Evaluation Framework

We evaluate every circuit using four standard metrics (Wang et al., 2023). For a circuit $C \subseteq E$ and test set $D$:

$$\text{Faithfulness:} \quad \mathbb{E}_{x \in D}\big[\text{KL}\big(p_{\text{base}}(\cdot \mid x) \,\|\, p_C(\cdot \mid x, \tilde{x})\big)\big], \tag{2}$$

$$\text{Sufficiency:} \quad \text{Acc}(C, D) = \tfrac{1}{|D|} \sum_{(x,y) \in D} \mathbf{1}\big[\arg\max p_C(\cdot \mid x, \tilde{x}) = y\big], \tag{3}$$

$$\text{Necessity:} \quad \text{Acc}_{\text{abl}}(C, D) = \text{Acc}(E \setminus C, \, D), \tag{4}$$

$$\text{Minimality:} \quad |C| \,/\, |E|. \tag{5}$$

Each metric is computed at two levels of granularity, which determine how activations flow through the graph. In *edge-level* evaluation, for each $(u, v) \in E$ the activation from $u$ to $v$ is clean (computed on $x$) if $(u, v) \in C$, and resample-ablated (computed on $\tilde{x}$) otherwise. In *source-level* evaluation, let $V_C = \{u \in V : \exists v, (u, v) \in C\}$ be the set of nodes contributing at least one circuit edge. All outgoing edges from any $u \in V_C$ carry clean activations, even those not in $C$; only edges originating from nodes outside $V_C$ are ablated. Source-level evaluation is therefore more permissive: it allows information to flow through pathways that ACDC did not select but that originate from circuit-participating components. Throughout, we report edge-level as the primary metric and use source-level comparisons to expose how evaluation granularity affects apparent circuit accuracy (Section 5.2.1). All statistical tests use non-parametric methods (Kruskal-Wallis (Kruskal & Wallis, 1952), Mann-Whitney U (Mann & Whitney, 1947), Wilcoxon signed-rank (Wilcoxon, 1945), Jonckheere-Terpstra (Jonckheere, 1954; Terpstra, 1952), Spearman (Spearman, 1904)) with Benjamini-Hochberg false discovery rate correction (Benjamini & Hochberg, 1995) at $\alpha = 0.05$.[4]

---

[2]Per-condition extraction details in subsection C.4.
[3]Defined in subsection C.5.
[4]Test catalogue and FDR procedure in subsection D.11.

# 4 Evidence for Frequency-Dependent Specialization

We characterize the 75 extracted circuits through four analyses, corresponding to the four subsections below. First, we evaluate each of the five models on all five conditions (four frequency bands plus control) to establish **base model performance** and assess how accuracy varies across frequency bands. Second, we validate **circuit quality** by measuring faithfulness (KL divergence), sufficiency (circuit accuracy), necessity (ablation accuracy), and minimality (edge fraction), and compare each circuit against 100 random baselines. Third, we perform a **structural comparison** by measuring edge overlap between all pairs of circuits (Jaccard similarity: shared edges divided by total unique edges) and testing whether circuits from the same frequency band overlap more than circuits from different bands, with bootstrap confidence intervals to assess reliability. Fourth, we measure **cross-band transfer** by evaluating each circuit on all other bands' test data, yielding a $5\times5$ transfer matrix per model and draw, and test for directional asymmetry between low- and high-frequency circuits.

## 4.1 Base Model Performance

Before examining circuits, we establish the base model's sensitivity to token frequency (full per-model breakdown in subsection D.1). Pythia-70m shows a pronounced frequency gradient: top-1 accuracy rises from 30.1% on the low band to 67.0% on very_high. All larger models achieve at least 92.9% across bands, with near-ceiling top-5 accuracy ($\geq$99%). The frequency effect on base accuracy is statistically significant only for Pythia-70m (Kruskal-Wallis $H = 13.5$, $\eta^2 = 0.95$, $p_{\text{BH}} = 0.036$). The task itself is therefore largely frequency-invariant for models with sufficient capacity, so any circuit-level frequency effects in the larger models cannot be attributed to differences in task difficulty.

## 4.2 Circuit Size and Sparsity

Circuit sparsity increases with model scale: ACDC retains 30.9% of edges for Pythia-70m, 12.3% for Pythia-160m, 4.4% for Pythia-410m, 9.1% for Pythia-1b, and 2.8% for Pythia-1.4b (Pythia-1b has fewer layers and heads than Pythia-410m, so its computational graph is much smaller, making the edge fraction higher despite retaining fewer edges in absolute terms; Table 1). For models $\geq$160M, low-frequency circuits are consistently larger than high-frequency circuits: for example, in Pythia-410m, low-frequency circuits retain 4.9% of edges versus 4.1% for very_high, and in Pythia-1.4b, 3.2% versus 2.5% (see subsection D.2). Pythia-70m shows the opposite trend (low 29.4% vs. very_high 31.7%), likely reflecting its low base accuracy rather than a frequency-specific effect. Although this size gradient is consistent across the four larger models, the absolute differences are small, foreshadowing the functional equivalence established in Section 5.

## 4.3 Circuit Quality Validation

All 75 circuits pass standard quality checks. When evaluated at edge level on the band used for extraction, each circuit recovers at least 80% of the base model's accuracy across all model–band combinations (subsection D.3), and KL divergence remains low (0.21–0.71 across all 75 circuits; Table 1 reports per-model values for the control band). Every circuit outperforms all 100 random baselines. Ablation accuracy is 0% for 69 of 75 circuits and at most 0.44% (1/225) for the remaining six: removing the circuit eliminates task performance, confirming that the discovered edges are necessary (see subsection D.4). These results establish that the circuits are meaningful subgraphs rather than artifacts of the extraction procedure, and that subsequent comparisons across frequency bands start from a common baseline of circuit quality.

## 4.4 Circuits Show Structural Differences

Having established that all circuits are individually faithful, we now ask whether circuits extracted from different frequency bands differ structurally. We measure similarity using the Jaccard index (Jaccard, 1901) over edge sets, $J(C_i, C_j) = |C_i \cap C_j| \,/\, |C_i \cup C_j|$, and compare two types of circuit pairs: *within-band* (same frequency band, different draws) and *between-band* (different frequency bands). Because Jaccard is symmetric and does not capture subset relationships, we complement it with directed containment analysis

(subsection E.10), which confirms that low-frequency circuits contain more high-frequency edges than the reverse.

Within-band similarity exceeds between-band similarity in all five models, with gaps of 0.013–0.032 (absolute Jaccard values range from 0.38 to 0.79, so circuits overlap substantially regardless of band; the gap corresponds to only 8–55 additional shared edges out of circuits containing 400–3,600 edges; Table 27; subsection E.6). Bootstrap 95% confidence intervals (Efron, 1987) ($N$=10,000) exclude zero in all models, confirming that the effect is reliable. However, the absolute magnitudes are small: for models ≥160M, the largest gap is 0.032 (95% CI $[0.024, 0.041]$) and the smallest is 0.013 ($[0.003, 0.023]$). Pythia-70m shows the same gap (0.032, $[0.023, 0.042]$) but on a lower base accuracy, where structural variation is more likely to carry functional consequences (Section 5.1.1).

To calibrate these differences, we compare them against extraction noise: the between-band gap is only 0.51–1.56 SD of the cross-draw Jaccard variation, comparable to the variation from running ACDC twice on the same data. This indicates that structural differences between bands should be interpreted with caution. A formal power analysis (subsection E.7) confirms that 2–5 draws per frequency band are needed to reliably detect gaps of this magnitude; our 3-draw design achieves ≥0.89 power for four of five models. Attention edges carry the largest share of the structural gap (75–93% of band-specific edges are attentional; subsection E.9).

We can further decompose the edge sets by how many conditions share each edge. Edges present in all five conditions form the *universal core*; edges appearing in only one condition are *band-specific*. The universal fraction (computed over the union of all draws) decreases from 65.5% in Pythia-70m to 24.3% in Pythia-410m (subsection E.5), meaning that larger models appear to have more band-specific circuitry (Figure 5 visualizes the full edge-sharing spectrum across all five tiers). Taken together, these results suggest that ACDC recovers structurally different circuits for different frequency bands, apparent evidence of frequency-dependent specialization.

### 4.5 Asymmetric Transfer

Cross-band transfer is high overall, with small generalization gaps (same-band accuracy minus mean cross-band accuracy) that are often negative, meaning circuits frequently perform better on other bands than on their own, particularly when transferring from harder low-frequency to easier high-frequency bands. Detailed generalization gaps in subsection D.5. However, the transfer is asymmetric: low-frequency circuits generalize to high-frequency inputs more effectively than the reverse. For Pythia-1.4b, a circuit extracted on the low band achieves 92.1% accuracy when evaluated on very_high inputs, whereas the reverse direction achieves only 71.7%, a gap of 20.4 percentage points; Pythia-160m shows the same pattern with a smaller gap (91.8% vs 86.1%; subsection D.6). This pattern is consistent across all models above 70m and aligns with the size gradient from Section 4.2: larger (low-frequency) circuits contain a superset of edges used by smaller (high-frequency) circuits, so they transfer well to easier inputs. The structural asymmetry, where low-frequency circuits contain high-frequency edges more than the reverse, is statistically significant for Pythia-160m, 410m, and 1b, and the same pattern holds for Pythia-1.4b. subsection F.7 analyzes predictors of this asymmetry. The asymmetry appears consistent with specialization: low-frequency circuits include more of the computation needed for high-frequency inputs, but not vice versa.

Taken together, Sections 4.1–4.5 present what appears to be strong evidence for frequency-dependent circuit specialization: circuits differ structurally, the differences are systematic (low-frequency circuits are larger and more inclusive), and these differences produce asymmetric transfer. We next test whether this structural divergence reflects genuine functional specialization.

## 5    Phantom Specialization

We now subject the apparent specialization from Section 4 to a systematic falsification battery designed to test whether the observed structural divergence reflects genuine frequency-dependent mechanisms or artifacts of the extraction process. The battery comprises four independent lines of evidence: (i) targeted functional tests of whether band-specific edges provide band-specific or generic computation (subsection 5.1);

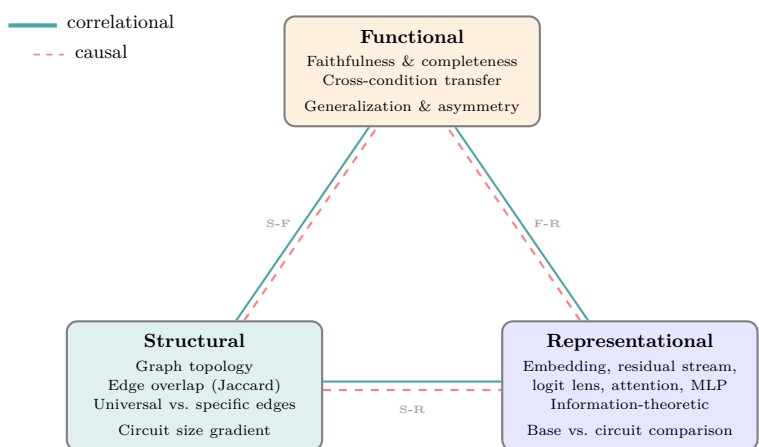

Figure 4: The similarity triangle: three axes for comparing circuits, each probed both correlationally (metric-pair correlations) and causally (interventions). **S–F:** do structurally different circuits produce different outputs? Tested via cross-band transfer (causal) and Spearman correlations between structural and functional metrics (correlational; 52/80 pairs exhibit Simpson's paradox). **S–R:** does structural divergence predict representational divergence? Tested via interchange patching and Boundless DAS (causal) and metric-pair correlations (correlational; 68/80 reversals). **F–R:** do functional and representational metrics agree? Tested via the same causal interventions and 64 metric-pair correlations (45/64 reversals). Phantom specialization corresponds to low structural similarity coexisting with high functional and representational similarity; genuine specialization would show low similarity on all three axes.

(ii) identification of methodological confounds that inflate apparent specialization (subsection 5.2); (iii) representational similarity analysis across four metrics to test whether circuit pruning alters internal geometry (subsection 5.3); and (iv) causal interchange interventions that directly test whether band representations are interchangeable (subsubsection 5.3.2). As a cross-method check, we repeat the core analysis using EAP and EAP-IG circuits, which select largely different edges (Jaccard 0.28–0.60 with ACDC), providing a method-independent control (developed in subsection 5.1, "Three additional controls").

These analyses span three independent axes (Figure 4): **structural** (edge overlap via Jaccard indices; Section 4.4), **functional** (output equivalence via cross-band transfer; Sections 4.3–4.5), and **representational** (internal activations across layers). Each axis can independently confirm or refute specialization; genuine specialization would require low cross-band transfer, band-specific necessity under targeted ablation, *and* robust representational divergence tied to distinct causal pathways. Phantom specialization is diagnosed when structural divergence coexists with high functional and representational similarity, a dissociation that no single test could establish but that the convergence of all four lines of evidence renders unambiguous.

## 5.1 Band-Specific Edges Are Functionally Generic

### 5.1.1 Edge Sharing and Band-Specific Transfer

To test whether structural differences reflect genuine specialization, we decompose each circuit by its *sharing count*: for an edge $e$ and the set of conditions $\mathcal{B}$, let $\kappa(e) = |\{b \in \mathcal{B} : e \in C_b\}|$ denote the number of conditions whose circuit contains $e$. Edges with $\kappa = |\mathcal{B}|$ form the *universal core*; edges with $\kappa = 1$ are *band-specific*. Figure 5 shows the full sharing spectrum. The universal core forms a single connected subgraph covering 64–91% of circuit nodes; connectivity matters because a disconnected core would suggest structurally unrelated computations, whereas a connected core implies a coherent shared mechanism. Its edges are highly stable across independent ACDC draws (94–98% appear in all three draws). Band-specific edges, by contrast, are overwhelmingly attentional (75–93%), and only 1–2% are stable across draws.[5] This instability suggests that band-specific edges largely reflect extraction noise rather than meaningful specialization. The canonical

---

[5] Per-component stability table in subsection E.14.

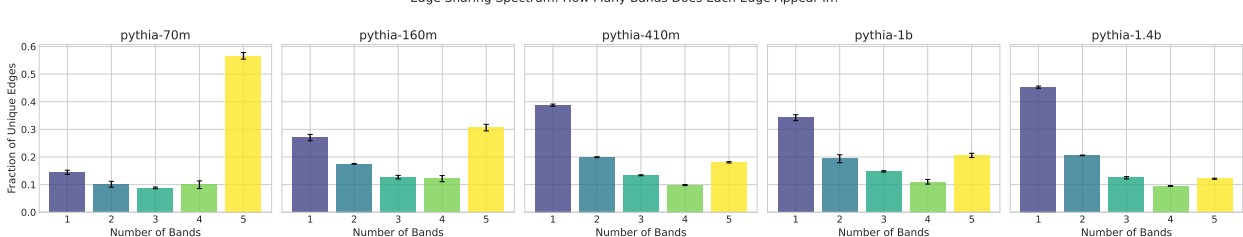

Figure 5: Edge sharing spectrum by model. Each bar shows the fraction of edges appearing in exactly $k$ of five frequency bands. Smaller models concentrate edges in the universal tier ($k=5$); larger models distribute more edges across partial-sharing tiers. Band-specific edges ($k=1$) are unstable across draws and functionally inert in isolation.

copy mechanism, where previous-token heads (which attend to the immediately preceding token) compose with induction heads (which match repeated prefixes and copy the following token) (Olsson et al., 2022), is 85–100% universal, while band-specific edges are disproportionately BOS-sink and diffuse-attention heads whose contribution is substantial but band-agnostic (subsection H.6; BOS-sink ablation in subsection H.7).

The universal core alone recovers 13.7–71.4% of full circuit accuracy (decreasing monotonically with model scale; see Table 5), while band-specific edges in isolation achieve 0% accuracy, establishing that the core is both partially sufficient and strictly necessary (sufficiency evidence in subsection H.1, complement-ablation in subsection H.9). The remaining gap is closed not by band-specific edges but by edges shared across three or four bands: relaxing to $k{\geq}3$ recovers $\geq$98% of full circuit accuracy for models above 70m (Section 5.1.2). Majority-vote aggregation across draws increases accuracy by $\sim$5 percentage points without changing transfer efficiency ($\leq$0.012 difference), confirming that extraction noise rather than latent band-specific structure drives cross-draw variation.

If band-specific edges encoded frequency-specialized computations, they should preferentially benefit the band on which they were discovered. We test this by adding each band's specific edges ($\kappa = 1$) to the universal core and evaluating the resulting circuit on all five conditions. Let $C_{\mathrm{core}}$ denote the universal core and $C_b^*$ the band-specific edges for condition $b$. The *same-band boost* is $\Delta_b = \mathrm{Acc}(C_{\mathrm{core}} \cup C_b^*, D_b) - \mathrm{Acc}(C_{\mathrm{core}}, D_b)$, and the *cross-band boost* to condition $b'$ is $\Delta_{b \to b'} = \mathrm{Acc}(C_{\mathrm{core}} \cup C_b^*, D_{b'}) - \mathrm{Acc}(C_{\mathrm{core}}, D_{b'})$. We define *transfer efficiency* as the ratio of mean cross-band boost to same-band boost:

$$\mathrm{TE}(b) \;=\; \frac{\frac{1}{|\mathcal{B}|-1} \sum_{b' \neq b} \Delta_{b \to b'}}{\Delta_b} \,. \tag{6}$$

TE = 1 indicates that band-specific edges provide a purely generic benefit; TE $\ll$ 1 would indicate genuine specialization. The observed boost is nearly uniform across test bands (Table 2; Figure 45 in subsection H.2): transfer efficiency ranges from 81.4% (Pythia-70m; 95% CI [74, 91]) to 97.0% (Pythia-1.4b; [94, 99]). In plain terms, edges labeled as "low-frequency-specific" boost performance on high-frequency inputs nearly as much as on low-frequency inputs; if these edges truly encoded band-specific computation, one would expect cross-band benefit to be substantially smaller than same-band benefit, but it is not. The same-band advantage is statistically significant ($p < 0.03$) but small-to-medium in effect size (Cohen, 2013) (Cohen's $d = 0.22$–0.71), and random edges of the same count contribute only 1–3% boost. The absolute same-band gap is approximately constant at 0.016–0.029 points across scales, even as the generic boost grows from 0.12 to 0.76, indicating a fixed structural offset rather than growing specialization. Under zero ablation, this residual vanishes entirely: the same-vs-cross boost difference drops to $\leq$0.012 points for all models, with Cohen's $d$ falling from 0.22–0.71 (resample) to $\leq$0.18 (zero; subsection H.12, Table 61), confirming it as an artifact of resample ablation rather than genuine band-specific computation. This zero-ablation diagnostic is critical: it rules out the possibility that the small same-band advantage reflects a real but weak specialization signal, attributing it instead to a known property of the corruption procedure. Band-specific edges are 5–263$\times$ more common than a noise-only null model predicts, indicating that ACDC's pruning is genuinely sensitive to input-distribution differences, but these structurally real differences are functionally

Table 2: Cross-band transfer of band-specific edges. Transfer efficiency is the ratio of cross-band to same-band accuracy boost; $d$: Cohen's $d$ for the same-vs-cross difference; $p$: permutation test (Fisher, 1966). 95% bootstrap percentile CIs ($N$=10,000) in brackets.

| Model | Same Boost | Cross Boost | Random | Transf. Eff. [95% CI] | $d$ | $p$ |
|---|---|---|---|---|---|---|
| Pythia-70m | $0.122 \pm 0.055$ | $0.099 \pm 0.035$ | 0.014 | 0.814 [.74, .91] | 0.49 | 0.004 |
| Pythia-160m | $0.304 \pm 0.059$ | $0.282 \pm 0.048$ | 0.028 | 0.926 [.89, .96] | 0.42 | <0.001 |
| Pythia-410m | $0.450 \pm 0.080$ | $0.434 \pm 0.064$ | 0.024 | 0.964 [.95, .98] | 0.22 | 0.006 |
| Pythia-1b | $0.513 \pm 0.091$ | $0.485 \pm 0.064$ | 0.027 | 0.944 [.92, .97] | 0.36 | 0.001 |
| Pythia-1.4b | $0.756 \pm 0.027$ | $0.733 \pm 0.037$ | 0.010 | 0.970 [.94, .99] | 0.71 | 0.028 |

Table 3: Cross-method comparison at ACDC-matched circuit size (1.0×). Δ: same-band minus cross-band accuracy (positive would indicate specialization). Jaccard: edge overlap with the corresponding ACDC circuit. EAP-IG circuits require 3–5× more edges for comparable faithfulness (subsection C.9).

| Method | Model | Same Acc | Δ | Jaccard |
|---|---|---|---|---|
| ACDC | 160M | 0.81 | −0.05 | — |
| ACDC | 410M | 0.83 | −0.04 | — |
| ACDC | 1.4B | 0.85 | +0.01 | — |
| EAP-IG | 160M | 0.15 | +0.01 | 0.42 |
| EAP-IG | 410M | 0.09 | −0.01 | 0.32 |
| EAP-IG | 1.4B | 0.02 | −0.03 | 0.28 |

inert (zero standalone accuracy, ≤2% stable across draws). The structural variation is the expected part; its functional irrelevance is the surprise.

Three additional controls corroborate this conclusion. First, per-example agreement between same-band and cross-band circuits is 83–93% for models ≥160M, with no same-band advantage, ruling out the possibility that aggregate transfer masks systematic per-input divergence. Second, an **independent EAP-IG analysis** (Hanna et al., 2024) corroborates the null on the same 75 extraction conditions (Table 3): same-band advantage stays at most 4.6 percentage points across all five models and ten size multipliers (with inconsistent sign), Jaccard overlap with ACDC is only 0.28–0.60, and the result holds from 1% to 100% of edges. Even at the smallest size where EAP-IG circuits achieve ≥50% faithfulness (1.5×–5× ACDC edges), same-band advantage stays in [+0.8, +4.6] percentage points; this rules out the alternative explanation that the null is an artifact of comparing incoherent EAP-IG circuits to coherent ACDC ones (subsection C.9). Third, adjacent ACDC thresholds (2–3× range in circuit size) yield at most 5.9 percentage points change in transfer efficiency, confirming that our conclusions are robust to the threshold selection (subsection C.8).

This is the defining signature of *phantom specialization*: the structural differences documented in Section 4.4 do not translate into band-specific functional advantages.

### 5.1.2 Shared-Core Sufficiency

Relaxing the sharing requirement from strict universality ($k$=5, all conditions) to majority sharing ($k$≥3, at least three of five conditions) recovers ≥98% of full circuit accuracy for Pythia-160m and above (95% CI lower bound; Figure 6). Pythia-70m is excluded from this bound due to its low base accuracy (30–67%), which makes gap-closure ratios unstable; its lower transfer efficiency (81%) likely reflects noisier extraction rather than a genuinely different mechanism. The critical threshold is predominantly $k$=3;[6] truly band-specific edges ($k$=1) contribute negligible additional accuracy. Universal-core retention ($k$=5) declines monotonically with model size (71% → 67% → 53% → 45% → 14% from 70M to 1.4B), consistent with increasing pathway redundancy: ACDC selects different routes across extractions, reducing the strict intersection while preserving the shared computation. The majority-shared core ($k$≥3) closes this gap entirely, and $k$≥3 edge

---

[6]Per-band critical-$k$ in subsection H.3.

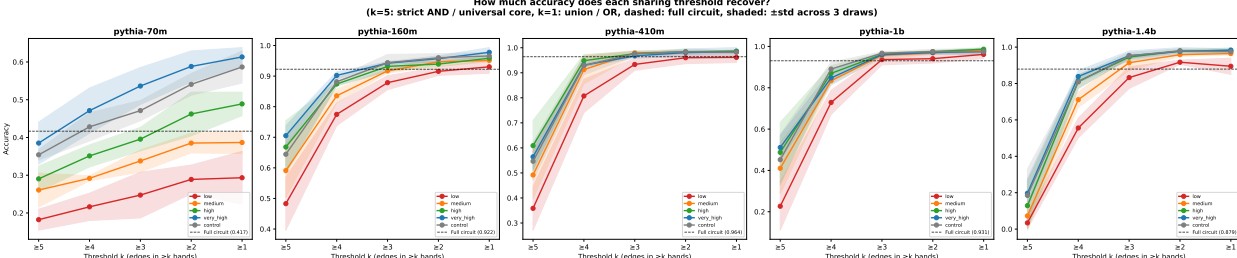

Figure 6: Circuit accuracy as a function of sharing threshold $k$. Lower $k$ includes more edges; $k=5$ is the strict universal core. Accuracy reaches $\geq 95\%$ of the full circuit at $k=3$ for most model×band combinations.

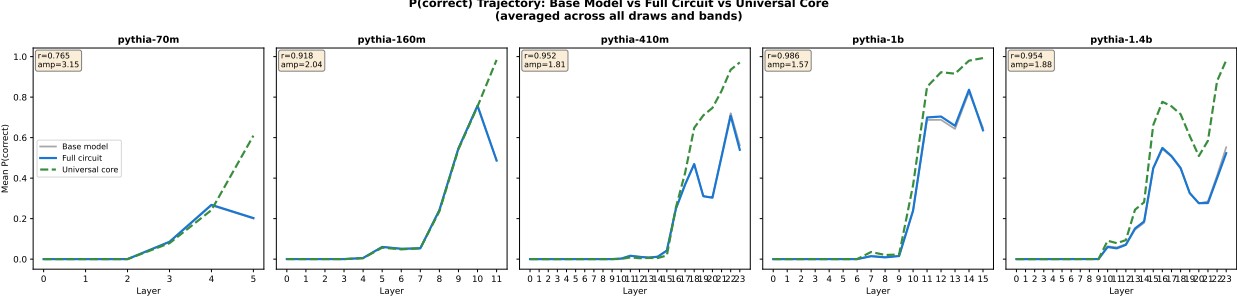

Figure 7: Logit lens trajectories: universal core (dashed) vs. full circuit (solid), with the base model (gray) for reference. The core follows the same trajectory shape at lower P(correct) values, reaching its peak 0.5–2.2 layers later than the full circuit. Correlation increases with model scale ($r = 0.77$ for Pythia-70m to 0.99 for Pythia-1b). Note that the full circuit's P(correct) drops at the final layer while the core's continues to rise; this likely reflects the full circuit's additional edges transforming the residual stream away from the unembedding direction at the last layer.

sets are themselves band-agnostic (pairwise Jaccard 0.656 for Pythia-1.4b, nearly double the full-circuit between-band Jaccard of 0.366).[7] Pythia-1.4b exemplifies this structure–function dissociation most sharply: within-band draw sharing drops to 21–27% (vs. 68–73% for Pythia-70m; subsection H.5), yet cross-band transfer matches or exceeds the other models, demonstrating that maximal structural divergence is fully compatible with functional interchangeability.

The universal core is not merely a set of overlapping edges: it is enriched for the components of the canonical copy mechanism identified by Olsson et al. (2022). Previous-token attention heads are significantly enriched among universal heads (odds ratio 3.7–14.4, $p < 0.05$), and induction heads are enriched in the two larger models ($p < 0.04$). Per-mechanism breakdown including BOS-sink ablation in subsection H.6. Notably, in Pythia-1b every induction head is classified as universal (OR $= \infty$), and in Pythia-410m seven of eight are (OR $= 10.3$): the core mechanism for the task is largely frequency-invariant. BOS-attending (sink) heads, which do not contribute to the copy computation, are depleted (OR $= 0.10$–$0.38$, $p < 0.002$). This composition, previous-token heads feeding induction heads, is the canonical mechanism for sequence copying (Olsson et al., 2022).

Logit lens (Nostalgebraist, 2020) trajectories of the universal core correlate highly with those of the full circuit: Pearson (Pearson, 1895) $r = 0.77$ (Pythia-70m), 0.92 (Pythia-160m), 0.95 (Pythia-410m), and 0.99 (Pythia-1b; Figure 7). subsection H.10 establishes that the core reaches its peak P(correct) 0.5–2.2 layers later than the full circuit, and its per-layer P(correct) values are 1.6–3.1× lower, consistent with a weaker version of the same algorithm rather than a qualitatively different computation.

---

[7]$k{\geq}3$ edge-set composition in subsection H.4.

Table 4: Draw stability by edge sharing level (% of edges appearing in 1, 2, or 3 of 3 independent ACDC draws). Band-specific: edges unique to one of five frequency bands; universal: edges shared across all five.

| | Band-specific | | | Universal | | |
|---|---|---|---|---|---|---|
| Model | 1 | 2 | 3 | 1 | 2 | 3 |
| Pythia-70m | 87 | 12 | 1 | 8 | 10 | 82 |
| Pythia-160m | 87 | 12 | 1 | 12 | 17 | 71 |
| Pythia-410m | 89 | 10 | 1 | 17 | 22 | 60 |
| Pythia-1b | 93 | 6 | 1 | 16 | 25 | 59 |
| Pythia-1.4b | 90 | 10 | 1 | 19 | 24 | 57 |

### 5.1.3 Cross-Draw Stability

Phantom specialization is not limited to cross-band comparisons; it also appears across independent extractions on the *same* band. Extraction noise is substantial: across three independent ACDC draws on the same band and model, 68–73% of edges appear in all three draws for Pythia-70m, but only 29–31% for Pythia-410m (Table 4; subsection H.5). Band-specific edges are particularly unstable: 87–93% appear in only one of three draws, compared with 57–82% of universal edges appearing in all three (Table 4). Yet cross-draw transfer, evaluating a circuit on data from the same band but a different draw, yields accuracy ratios of 0.987–1.004 (aggregate 95% CI [0.990, 1.007], spanning 1.0), indistinguishable from same-draw evaluation. This further demonstrates that structural variation does not imply functional variation: circuits that share as few as 29% of their edges produce equivalent outputs. The combination of high cross-draw instability (49–71% of edges absent from at least one draw for models ≥160M) and near-perfect cross-draw transfer implies that a single ACDC extraction is composed of a minority of truly essential edges and a substantial fraction of functionally inert edges that happened to survive greedy pruning in that particular run. This places a quantitative bound on the fraction of functionally inert edges in any single discovered circuit: mechanistic conclusions drawn from single-extraction analyses rest on an edge set where potentially half or more of the edges are not required for the circuit's function. Multiple extractions and majority-vote aggregation are therefore necessary to distinguish stable circuit structure from extraction noise.

### 5.2 Methodological Confounds

Having established that band-specific edges are functionally generic, we now identify two methodological factors that have obscured this finding in prior work.

### 5.2.1 Evaluation Granularity

The standard evaluation granularity used in much circuit-discovery work systematically obscures the phantom: source-level evaluation (Section 3.4) inflates apparent circuit accuracy. When the universal core is evaluated at source-level, preserving all outgoing edges from universal nodes rather than only the edges ACDC selected, accuracy jumps to 0.51–0.99, compared with 0.30–0.62 at edge-level (Figure 8). The inflation ranges from 0.22 (Pythia-70m) to 0.57 (Pythia-1b) accuracy points. At source-level, even an incomplete circuit can appear nearly faithful because information flows through unselected pathways that happen to originate from circuit-participating components. This granularity dependence connects to the broader principle that causal structure manifests differently at different scales of analysis (Noble, 2011; Hoel et al., 2013). We develop this connection in Section 6.1. The implication is that any study comparing circuits across conditions must evaluate at edge level to avoid this confound.

Table 5 consolidates the phantom specialization metrics across model scale. Two trends run in opposite directions: the universal fraction decreases from 73.7% (Pythia-70m) to 36.4% (Pythia-410m), making circuits *appear* more specialized at scale, while transfer efficiency *increases* from 81.5% [74, 91] to 96.4% [95, 98], meaning that whatever band-specific edges exist become more generic (subsection H.8). At the same time, source-level inflation grows with scale (the source-minus-edge accuracy gap widens from 0.22 to 0.57), so the methodological artifact that masks the phantom becomes more severe for larger models. Cross-draw

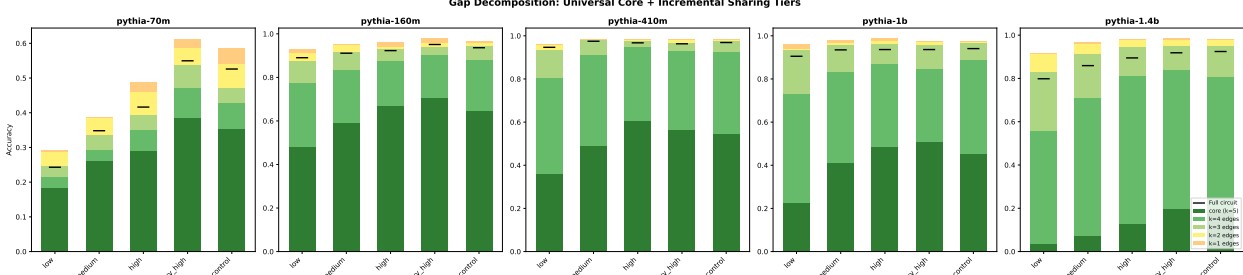

Figure 8: Performance gap decomposition. The gap between the universal core (edge-level) and full circuit grows with model scale: 0.12 (Pythia-70m), 0.30 (Pythia-160m), 0.45 (Pythia-410m), and 0.51 (Pythia-1b). Each bar decomposes this gap by sharing tier $k$ (number of frequency bands containing the edge; $k$=1: band-specific; $k$=5: universal). Including $k{\geq}3$ edges closes nearly all of the gap; $k$=1 (band-specific) edges contribute negligibly. Source-level evaluation (top bar per model) dramatically inflates apparent accuracy compared with edge-level.

Table 5: Scaling synthesis of phantom specialization metrics. Universal fraction (per-draw average) decreases with scale (circuits *look* more specialized), but transfer efficiency increases (band-specific edges become more generic), and cross-draw transfer remains near-perfect. Crit. $k$: minimum sharing threshold at which accuracy reaches $\geq$95% of the full circuit (mean over bands). 95% bootstrap CIs in brackets.

| Model | Univ. % | Retention | Transf. Eff. [CI] | Draw Transf. [CI] | Crit. $k$ |
|---|---|---|---|---|---|
| Pythia-70m | 73.7 | 0.714 | 0.814 [.74, .91] | 0.990 [.94, 1.05] | 2.8 |
| Pythia-160m | 50.7 | 0.669 | 0.926 [.89, .96] | 0.998 [.99, 1.01] | 3.0 |
| Pythia-410m | 36.4 | 0.533 | 0.964 [.95, .98] | 0.999 [.99, 1.01] | 3.6 |
| Pythia-1b | 38.9 | 0.447 | 0.944 [.92, .97] | 1.004 [.99, 1.01] | 3.0 |
| Pythia-1.4b | 27.2 | 0.137 | 0.970 [.94, .99] | 1.017 [1.00, 1.03] | 3.0 |

transfer remains near-perfect across all scales (ratio 0.987–1.004; subsection H.8), confirming that circuits are functionally reproducible even when structurally variable.

### 5.2.2 Variance Decomposition

Across all three axes of the similarity triangle, model scale is the dominant source of variance. Unified variance decomposition assigns the model factor the largest share for every structural, functional, and representational metric examined (subsection G.4). The pooled functional-only decomposition attributes 93.1% of variance to model and 2.6% to band, but this pooled band effect is driven primarily by Pythia-70m's frequency gradient (base accuracy 30–67%). The more informative statistic is the *within-model* band effect: the frequency effect on base accuracy is statistically significant only for Pythia-70m ($p_{\mathrm{BH}} = 0.036$; Section 4.1), and circuit accuracy likewise shows a band gradient only for that model. Model scale bundles several confounded factors (graph topology, baseline performance, and pathway redundancy) that cannot be disentangled with five models. However, baseline performance is likely the dominant driver rather than graph topology: Pythia-1b and Pythia-160m have similarly sized computational graphs (10K and 11K edges) yet both behave like the larger models on functional metrics, while Pythia-70m diverges primarily because the task is difficult for it (30–67% base accuracy).

This dominance produces pervasive Simpson's paradox (Simpson, 1951): correlations that appear strong when pooling across models reverse or vanish within individual models. subsection G.1 documents this Simpson's paradox across all structural-functional metric pairs. Along the structure–function edge of the triangle, 52 of 80 metric pairs exhibit sign reversal; along structure–representation, 68 of 80 pairs reverse (subsection G.3). For example, universal fraction correlates strongly with peak probe layer across models ($\rho = -0.96$), but the within-model correlation is near zero ($\rho = -0.12$). Figure 9 illustrates this pattern for one representative metric pair: universal edge fraction versus circuit size fraction. The pooled correlation is

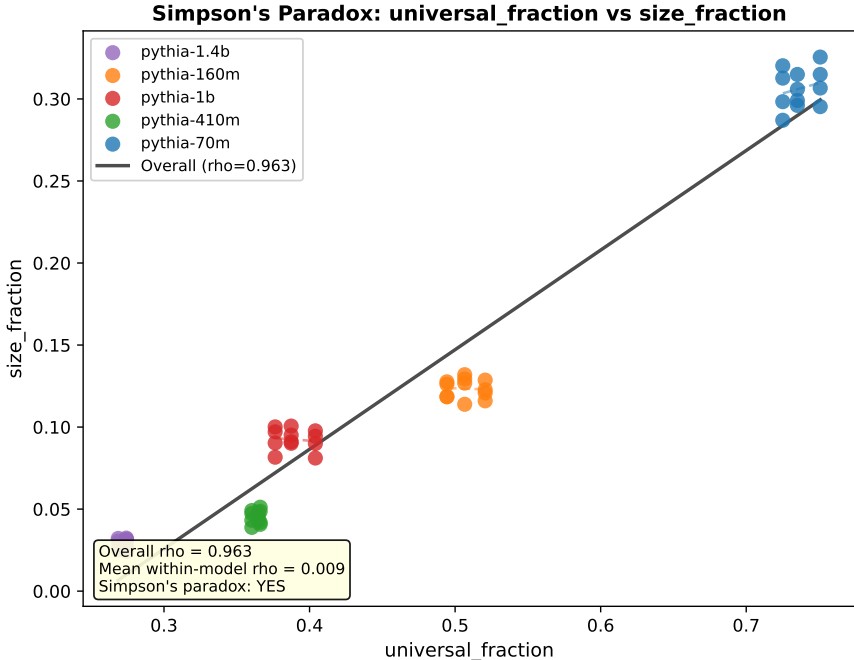

Figure 9: Simpson's paradox in cross-perspective correlations. Universal edge fraction and circuit size fraction correlate strongly when pooling across models (black line; $\rho = 0.96$), but per-model regressions (colored) are near-zero ($\bar{\rho} = 0.01$). Model identity, not frequency band, drives the apparent association. This pattern recurs in 52/80 structure–function and 68/80 structure–representation metric pairs.

near-perfect ($\rho = 0.96$), yet per-model regressions are essentially flat (mean within-model $\rho = 0.01$). The same reversal pattern holds across the majority of structure–function and structure–representation pairs.

These reversals are directly relevant to phantom specialization: they show that the apparent structural–functional relationship between frequency and circuit organization is an artifact of pooling across model scales, not a within-model phenomenon. Any study comparing circuits across conditions must control for model identity to avoid this confound.

## 5.3 Representational and Causal Confirmation

The preceding analyses established the phantom along the structural–functional edge of the similarity triangle; we now examine the representational vertex for independent confirmation.

### 5.3.1 Representational Similarity

Across five representational metrics, the circuit retains the base model's representational geometry (Table 6); band-level differences in internal representations are a property of the model, not an artifact of circuit pruning.

### 5.3.2 Interchange Interventions

The representational metrics above are correlational; we now test whether band representations are *causally* interchangeable using two complementary intervention methods. Interchange patching (Geiger et al., 2024) replaces one band's residual-stream activations with another's at the prediction position and measures whether the model's output follows the source band's target (Interchange Intervention Accuracy, IIA). For models ≥160M, IIA is ≥0.95 across all 20 cross-band pairs, with the same-band versus cross-band difference at most 0.004 (Table 7; Figure 37 in subsubsection F.9.1); a layer-sweep positive control confirms

Table 6: Representational similarity between full circuit and base model across five metrics. Each row is independently consistent with the phantom-specialization conclusion: pruning preserves the base model's representational geometry.

| Metric | Result (circuit vs. base) | Detail |
|---|---|---|
| Embedding geometry | shared $k$NN/probe structure | subsection F.1 |
| Residual-stream CKA (Kornblith et al., 2019) | median $\geq 0.97$ | subsection F.2 |
| Attention entropy (Zhai et al., 2023) | base/circuit aligned | subsection F.4 |
| Band-identity MI (Shannon, 1948) | $< 10\%$ fractional loss | subsection F.6 |
| MLP neuron selectivity | Spearman $\rho = 0.71$–$0.98$ | subsection F.5 |

Table 7: IIA at peak layer: same-band versus cross-band. The same–cross difference is negligible ($\leq 0.004$) for all models, causally confirming that band identity does not determine representational format. Pythia-70m IIA is low overall (0.47), reflecting limited capacity rather than band sensitivity.

| | Pythia-70m | Pythia-160m | Pythia-410m | Pythia-1b | Pythia-1.4b |
|---|---|---|---|---|---|
| Peak layer (of total) | L5/6 | L11/12 | L23/24 | L13/16 | L21/24 |
| Same-band IIA | 0.466 | 0.962 | 0.984 | 0.986 | 0.968 |
| Cross-band IIA | 0.466 | 0.962 | 0.984 | 0.985 | 0.972 |
| Difference | +0.000 | +0.000 | +0.000 | +0.001 | −0.004 |

that the method has full dynamic range (IIA from 0 to $\geq 0.92$; section F.9.1). This establishes that band representations are not merely similar but causally interchangeable: swapping one band's internal state for another's preserves the model's computation with near-perfect fidelity. Boundless DAS (Wu et al., 2023) further localizes the causal information distinguishing bands to just 12–31 dimensions (0.6–6.1% of $d_{\mathrm{model}}$), and this fraction decreases with model scale (Table 8). Full details are in subsection F.9. Together, these causal results establish that the model employs a shared representational format across frequency bands: band identity is encoded in a tiny subspace of an otherwise band-generic representation, and this subspace is neither necessary nor sufficient for the copy computation.

### 5.3.3 A Genuine Frequency Effect: Processing Dynamics

Despite this broad representational invariance, one genuine frequency effect emerges, not in circuit structure, but in processing dynamics within the same circuit. Logit lens analysis reveals that the base model's output distribution converges to the final prediction later for low-frequency tokens (fractional depth 0.78–0.90) than for high-frequency tokens (0.61–0.72), and this gradient is consistent across all five models (subsection F.3). Circuit extraction preserves this convergence timing almost exactly (shift $< 0.05$ fractional layers), indicating that the frequency effect reflects the model's processing dynamics rather than an artifact of circuit pruning.

The convergence gap grows with model scale: from ∼0.06 fractional depth for Pythia-70m to ∼0.29 for Pythia-1.4b (Table 36); tuned-lens calibration (Belrose et al., 2023) reduces the scaling to ∼3× but preserves the relative ordering across bands in all five models (section F.3), confirming that the delay reflects genuine computational requirements. Neither component-usage shifts nor embedding-norm gradients explain the delay; the mechanistic cause remains open, though late-layer MLP contributions are slightly higher for low-frequency tokens, suggesting a *shared mechanism, differential effort* pattern consistent with phantom specialization.

Frequency thus modulates activation dynamics within a shared circuit rather than routing computation through different circuits: it influences *how* the model arrives at its prediction (later convergence for rare tokens) without changing *which* circuit it uses. This dissociation, representational sensitivity to frequency alongside structural and functional invariance, is a concrete instance of the broader phantom specialization phenomenon.

Table 8: Boundless DAS effective dimensions at peak layer. The subspace encoding band-distinguishing information decreases from 6.1% to 0.6% of $d_{\text{model}}$ with scale, while achieving perfect IIA in all cases.

| Model | Peak layer | $d_{\text{model}}$ | Eff. dim (low→high) | Eff. dim (high→low) | % of $d_{\text{model}}$ | IIA |
|---|---|---|---|---|---|---|
| Pythia-70m | L5 | 512 | 31 | 31 | 6.1 | 1.0 |
| Pythia-160m | L11 | 768 | 19 | 20 | 2.5 | 1.0 |
| Pythia-410m | L23 | 1024 | 17 | 18 | 1.7 | 1.0 |
| Pythia-1b | L13 | 2048 | 13 | 12 | 0.6 | 1.0 |
| Pythia-1.4b | L21 | 2048 | 12 | 12 | 0.6 | 1.0 |

## 5.4 Convergence of Evidence

The four lines of evidence developed in this section converge on a single conclusion: the structural specialization documented in Section 4 is phantom. *Functionally*, band-specific edges transfer 81–97% across all frequency bands, and majority-shared edges ($k{\geq}3$) recover $\geq$99% of full circuit accuracy. *Representationally*, residual-stream CKA remains $\geq$0.97 and MLP selectivity correlations reach $\rho = 0.71$–0.98 between base model and circuit. *Causally*, interchange patching yields IIA $\geq$0.95 across all band pairs, and Boundless DAS localizes band-distinguishing information to $\leq$6% of model dimensions. *Methodologically*, the zero-ablation diagnostic eliminates the residual same-band advantage, source-level inflation is quantified at 0.22–0.57 points, and an independent EAP-IG analysis finds no band specificity at any circuit size. No single test would be decisive; it is the independent convergence across structural, functional, representational, and causal axes, each capable of falsifying the phantom hypothesis had specialization been genuine, that warrants confidence in the conclusion.

# 6 Discussion

## 6.1 Why the Phantom Exists

Phantom specialization arises because the mapping from circuit structure to circuit function is many-to-one. When multiple edges are individually dispensable because the model implements the same function through redundant pathways (McGrath et al., 2023; Bushnaq et al., 2024; Rohweder et al., 2026), greedy pruning algorithms must choose which to keep, and small differences in input statistics can tip these choices differently, producing structurally distinct circuits that implement the same computation. Band-specific edges can be understood as *circuit spandrels* (Gould & Lewontin, 1979): structural byproducts of the extraction process rather than functional adaptations. Independent neuron-level evidence supports this conclusion: Liu et al. (2025) find that rare-token processing relies on coordinated subnetworks within shared MLP layers rather than modular separation, the neuron-level analog of our circuit-level finding that frequency does not route computation through different circuits.

Three distinct sources of circuit variation should be distinguished: (i) *extraction instability* (same frequency band yields different edge sets across runs), (ii) *non-uniqueness of minimal subgraphs* (different conditions recover functionally equivalent circuits, the empirical analog of the non-identifiability demonstrated exhaustively in toy models by Méloux et al., 2025a), and (iii) *model-level degeneracy* (the model supports multiple independent pathways; Edelman & Gally, 2001). Table 9 maps our results to these sources. Our evidence directly establishes (i) and (ii); interpretation (iii) is consistent but not conclusively demonstrated.

Cross-draw analysis confirms extraction instability is substantial (16–46% of edges in only one draw), and the EAP-IG analysis in subsection 5.1, which selects largely different edges (Jaccard 0.28–0.60 with ACDC), likewise finds no band specificity, providing partial evidence for non-uniqueness. Per-example agreement (83–93% for models $\geq$160M, with no same-band advantage) confirms that coarse-metric agreement is not masking systematic divergence. All three sources likely contribute; what the evidence conclusively establishes is a many-to-one mapping from extracted graph to measured function, regardless of its precise origin. As motivated in the Introduction, there is no privileged level of causation in complex systems (Noble, 2011). Noble argues, from cardiac cell modeling, that downward causation is not merely compatible with lower-

Table 9: Mapping of empirical results to sources of circuit variation: (i) extraction instability, (ii) non-uniqueness of minimal subgraphs, (iii) model-level degeneracy. ✓: directly supports; ∼: consistent but not decisive (results marked ∼ for source (iii) are equally consistent with source (ii); only targeted multi-circuit ablation would provide evidence specifically for (iii)); —: does not bear on this source.

| Empirical result | (i) | (ii) | (iii) |
|---|---|---|---|
| Cross-draw Jaccard 0.29–0.73 (Sec. 5.1.3) | ✓ | — | — |
| 16–46% draw-exclusive edges | ✓ | — | — |
| Union circuits: +5 percentage points acc., ≤0.012 TE change | ✓ | — | — |
| Cross-band transfer 81–97% (Sec. 5.1.1) | — | ✓ | ∼ |
| Majority-shared core ($k{\geq}3$) ≥99% recovery (Sec. 5.1.2) | — | ✓ | ∼ |
| IIA ≥0.95 across all band pairs (App. F.9.1) | — | ✓ | ∼ |
| Boundless DAS: 0.6–6.1% of $d_{\mathrm{model}}$ (App. F.9.2) | — | ✓ | ∼ |
| EAP-IG: no band specificity at any size (App. C.9) | — | ✓ | ∼ |
| Per-example agreement 83–93%, no same-band advantage | ✓ | ✓ | — |

level mechanisms but necessary for understanding multi-scale systems; the causal structure one observes depends on the scale at which one analyzes it. Our results provide an empirical instance of this principle in neural networks: at source-level granularity, circuits appear faithful and specialization appears genuine; at edge-level granularity, the phantom is revealed. The 0.22–0.57-point inflation we measure (Section 5.2.1) quantifies how much apparent causal coherence is gained by coarsening the description. The mechanism is structurally analogous to causal emergence (Hoel et al., 2013): source-level evaluation collapses many functionally equivalent edge configurations (degeneracy in Hoel et al.'s terminology) into a single "faithful" macro-state, raising effectiveness at the cost of state-space resolution. In Hoel et al.'s framework such coarse-graining is desirable when the goal is to identify the level at which the system's causal interactions are most effective; in circuit evaluation, however, the goal is precisely to detect the degeneracy that coarsening abstracts away, because it reveals whether the extracted circuit is the unique mechanistic explanation or merely one of many equivalent graphs. This is, to our knowledge, the first quantitative demonstration of granularity-dependent causal structure in neural network circuits. For circuit discovery, this means that edge-level evaluation is not merely a stricter test but a fundamentally different window onto the model's causal structure, one that reveals redundancy and degeneracy that coarser analyses systematically obscure. The practical consequence is that treating a single discovered circuit as *the* mechanistic explanation overstates what the evidence supports. Methods that aggregate over circuit distributions may be more robust than point estimates.

### 6.2 Implications for Evaluating Circuits Across Conditions

Our results yield four practical recommendations for the circuit discovery community.

**Cross-condition transfer tests are necessary.** Structural differences between circuits should not be taken as evidence of functional specialization without cross-condition transfer tests. In our study, circuits that differ by up to 75% of their edges (Section 5.1.1) nonetheless produce equivalent outputs across all frequency bands (Section 5.1.1). Any future comparison of circuits across input conditions, model variants, or training stages should verify whether observed structural differences translate to functional differences before interpreting them as evidence of distinct mechanisms.

**Edge-level evaluation should be the primary metric.** Source-level evaluation inflates apparent circuit accuracy by 0.22–0.57 points (Section 5.2.1) and can make even incomplete circuits appear faithful. We

Table 10: Positive control: cross-condition transfer for Pythia-160m (zero-distractor vs. standard LSC). Each cell shows edge-level circuit accuracy. Transfer efficiency 96.7%; Jaccard 0.539.

|  | Zero-dist. test | Standard test |
|---|---|---|
| Zero-dist. circuit | 82.7% | 80.4% |
| Standard circuit | 88.9% | 92.4% |
| Base model | 88.4% | 98.7% |

recommend reporting edge-level metrics as the primary faithfulness measure and treating source-level results as an upper bound that reflects the contribution of unselected pathways. This confound is not specific to our task or perturbation axis; it applies to any study comparing circuits extracted under different conditions.

**Multiple extractions are needed to identify stable structure.** Even within the same frequency band, 29–73% of edges appear in only one or two of three extractions (Section 5.1.3). A single ACDC run therefore conflates stable structure with extraction noise. Running at least three independent extractions and retaining only majority-vote edges ($k{\geq}2$ of 3 draws, analogous to our $k{\geq}3$ of 5 bands) substantially reduces noise and yields a more reliable circuit estimate.

**Structural metrics require careful controls.** Within-model correlation between Jaccard overlap and functional transfer is near zero (Section 5.2.2); even our largest gaps (0.032, $d = 1.22$–$1.45$) are functionally silent. Evidence for genuine specialization should be sought in cross-condition transfer advantages, not structural divergence magnitude. More broadly, model scale dominates variance on every metric, producing pervasive Simpson's paradox (Section 5.2.2). Studies pooling observations across model sizes (e.g., Tigges et al., 2024; Mondorf et al., 2025; Ferrando & Costa-Jussà, 2024) risk creating spurious associations; we recommend reporting both pooled and per-model statistics.

### 6.3   Limitations

Our findings are established in a controlled setting: within-model, within-task variation along a single in-put axis (token frequency) in a non-semantic copying task, using one model family. ACDC is the primary extraction algorithm, but a complementary EAP-IG analysis (Hanna et al., 2024) finds no band specificity (subsection C.9), and causal interventions (Section 5.3.2) provide method-independent confirmation. Phantom specialization may not extend to semantic tasks or architectures with different redundancy profiles. Replication with differentiable mask-optimization methods (Bhaskar et al., 2024) remains an important future direction.

A further limitation is that a within-task positive control could not be constructed. A reverse-copy variant (target before prefix, requiring offset $-5$ instead of $+1$) achieves 0% accuracy: we could not elicit a distinct reverse-copy mechanism in this setup (subsection I.1). A zero-distractor variant (removing the 10 distraction tokens) uses the same mechanism: cross-condition transfer efficiency is 96.7%, exceeding the cross-frequency-band efficiency of 92.6%, despite lower structural overlap (Jaccard 0.539 vs. 0.557; Table 10; subsection I.2). Appendix I extends phantom specialization to a second perturbation axis (sequence length). The pipeline detects the residual same-band advantage of 0.016–0.029 points with $p < 0.03$, confirming adequate sensitivity (subsection E.8), but this advantage vanishes under zero ablation (subsection H.12). The absence of a positive control demonstrating detection of *genuine* specialization remains the most important limitation.

Three draws per frequency band suffice for functional conclusions (cross-draw transfer 0.987–1.004) but limit structural precision; the computational cost of ACDC extraction ($\sim$736 GPU-hours for 75 circuits) makes additional draws prohibitive. We recommend five or more draws for models $\geq$1B parameters (subsection E.7). Pythia-70m is a boundary case with low base accuracy (30–67%; subsection D.10). All primary conclusions hold within each model individually. We do not claim that all structural variation is phantom; we show that structural divergence alone is insufficient evidence for specialization.

# 7    Conclusion

We showed that circuit structure maps many-to-one to circuit function, and that this degeneracy is visible only at edge-level evaluation granularity. We term the resulting pattern *phantom specialization*: structural divergence that does not correspond to functional specialization. In Literal Sequence Copying across four frequency bands and five Pythia scales, ACDC finds structurally distinct circuits, but band-specific edges act as generic boosters (81–97% cross-band transfer efficiency), edges shared by $\geq 3$ bands recover $\geq 99\%$ of full circuit accuracy, and the universal core contains the canonical copy mechanism.

Our methodological recommendations, cross-condition transfer tests, edge-level evaluation, and multiple extractions with majority-vote aggregation, address recurring challenges in circuit discovery and can be adopted by any study comparing circuits across conditions. More broadly, our experiments establish a many-to-one mapping from extracted circuit graph to measured function, indicating that a single discovered circuit should not be treated as uniquely identifying the underlying mechanism.

Promising future directions include Bayesian circuit discovery (posterior inclusion probabilities), joint multi-condition extraction, feature-level circuit discovery via sparse autoencoders (Marks et al., 2025) or transcoders (Dunefsky et al., 2024) (testing whether phantom specialization persists when circuits are defined over monosemantic features), and developmental analysis using Pythia's training checkpoints. What is phantom is not the structural variation itself, which is real and input-driven, but the inference from structural divergence to functional specialization.

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

# Appendix

**Contents**

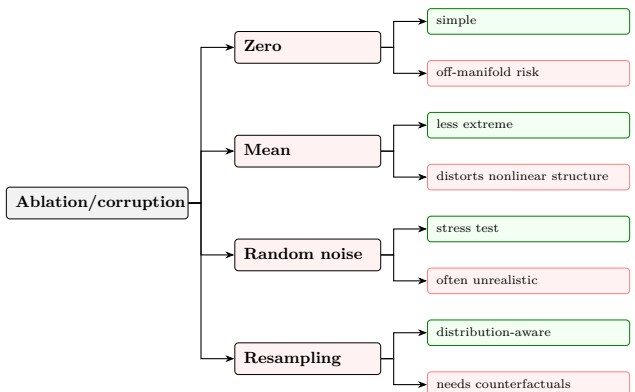

Figure 10: Common ways to construct the corrupt run for activation patching. Simpler ablations are easy to apply but can push activations off-manifold or distort the representation, whereas resampling-based corruption is usually more realistic but requires suitable counterfactual examples.

# A    Extended Background: Circuit Discovery and Evaluation

This appendix provides an extended review of circuit discovery methods, evaluation practices, and open challenges in mechanistic interpretability, complementing the overview in Section 2.1. For readers familiar with the field, the main-text treatment is self-contained; this appendix offers additional depth on method families, metrics, and structural challenges.

## A.1    Circuit Discovery and Evaluation

### A.1.1    Defining Circuits

Mechanistic interpretability seeks to reverse-engineer neural networks, analogous to recovering interpretable source code from a compiled program (Olah, 2022; Elhage et al., 2021). Three organizing concepts structure this enterprise: *features*, individual directions in activation space that correspond to interpretable concepts (Elhage et al., 2022); *circuits*, sparse subgraphs of the model's computational graph that explain how it implements a specific behavior (Olah et al., 2020; Conmy et al., 2023; Rai et al., 2024); and *universality*, the hypothesis that different models learn the same features and circuits for the same tasks (Olah et al., 2020; Chughtai et al., 2023).

Throughout this paper, we use *circuit* as an operational term: a sparse, task-relevant subset of model components or connections whose preservation is sufficient to recover substantial task performance under a specified intervention and evaluation procedure. This definition is intentionally method-relative: discovered circuits depend on how the computational graph is defined, which variables are intervened on, and how faithfulness is measured.

In a transformer, the residual stream serves as a shared communication channel through which attention heads and MLP blocks read and write, inducing a directed acyclic graph (DAG) of information flow. Nodes in this graph correspond to attention heads and MLP blocks; edges represent pathways of influence between components mediated by the residual stream. Due to the additive nature of the residual stream, even components in non-adjacent layers are effectively connected by writing to and reading from the same stream (Elhage et al., 2021). A *circuit* is a task-relevant subgraph of this DAG: it specifies which components matter for a given behavior and how they interact (Conmy et al., 2023; Rai et al., 2024). Although circuits were originally defined as connections between features (Olah et al., 2020), subsequent work has generalized them to connections between activation outputs of transformer components (Olsson et al., 2022; Wang et al., 2023). More recently, sparse autoencoders (SAEs) (O'Neill & Bui, 2024) have enabled a finer decomposition by mapping activations into higher-dimensional sparse representations with more monosemantic units, and initial work has begun defining circuits over SAE features rather than architectural components (O'Neill & Bui, 2024).

However, to our knowledge, whether feature-level circuits exhibit the same non-uniqueness as component-level circuits remains an open question.

A key challenge is that the naive decomposition into architectural components (individual neurons, attention heads, or layers) does not cleanly carve the network at its functional joints (Sharkey et al., 2025). Neurons and attention heads are often *polysemantic*, responding to multiple unrelated features (Elhage et al., 2022), which confounds circuit boundaries and motivates caution when interpreting circuits defined at this level of granularity.

The formal foundations for this approach trace to the causal abstraction framework of Geiger et al. (2025), which grounds interpretability claims in interchange interventions: replacing a component's activation with one from a different input and observing whether the model's output changes in the predicted way. This idea underpins *activation patching* (Vig et al., 2020; Meng et al., 2022), the causal tool used by most circuit discovery methods. An activation-patching experiment involves three forward passes: (1) a *clean run* on the original prompt, caching all intermediate activations; (2) a *corrupt run* on a modified prompt that disrupts the target behavior; and (3) a *patch run* in which the activation of a specific component is swapped between runs while the rest of the computation proceeds normally. Two complementary directions of patching are used (Rai et al., 2024; Heimersheim & Nanda, 2024). In *noising* (clean-to-corrupt) patching, the model runs on the clean input but one component's activation is replaced with its value from the corrupt run; if performance drops, the component is necessary. In *denoising* (corrupt-to-clean) patching, the model runs on the corrupt input but one component is restored to its clean value; if performance recovers, the component is sufficient.

The choice of corruption technique for the corrupt run is consequential. *Zero ablation* (Olsson et al., 2022) replaces the component's output with a zero vector, but this can push the model far out of distribution, causing effects unrelated to the component's actual role. *Mean ablation* (Wang et al., 2023) replaces the output with its mean over a reference distribution, partially mitigating the out-of-distribution problem but failing when activation distributions are non-linear (e.g., the mean of points on a circle lies at the center, not on the circle). *Resampling ablation* (Chan et al., 2022) replaces the output with the activation from a different, counterfactual input drawn from the same task distribution, avoiding the out-of-distribution issue at the cost of requiring paired counterfactual examples. Each choice can yield different circuits from the same model and task (Heimersheim & Nanda, 2024; Méloux et al., 2025b).

### A.1.2 Manual Circuit Discovery

The earliest circuits were identified through a hypothesis-driven, iterative process (Figure 2a) that follows a common workflow (Rai et al., 2024): (1) select a behavior for which the model performs well, ensuring the mechanism is reliably present; (2) define the computational graph, choosing the granularity of nodes (e.g., attention heads and MLP blocks) and the edge structure; (3) *localize* important nodes and edges via intervention; (4) *interpret* each component by generating and validating hypotheses about its functional role; and (5) *evaluate* the resulting circuit. In practice, steps 3–4 are iterative: researchers inspect activations, hypothesize which components are task-relevant, intervene via ablation or patching, measure the behavioral effect, and refine the hypothesized circuit until a stable, minimal subgraph emerges.

This manual approach has produced detailed mechanistic accounts of several core transformer behaviors: in-context copying via induction heads (Olsson et al., 2022), indirect object identification (IOI) with 26 attention heads grouped into 7 classes in GPT-2 Small (Wang et al., 2023), greater-than comparison (Hanna et al., 2023), copy suppression across the full training distribution (McDougall et al., 2024), successor operations (Gould et al., 2024), docstring completion (Heimersheim & Janiak, 2023), three-letter acronym prediction (García-Carrasco et al., 2024), subject–verb agreement across languages (Finlayson et al., 2021; Ferrando & Costa-Jussà, 2024), extractive question-answering (Basu et al., 2025), and arithmetic via a "bag of heuristics" (Nikankin et al., 2025). These studies have revealed specialized component roles: previous-token heads, duplicate-token heads, induction heads, negative heads that suppress already-appeared tokens, successor heads, and multi-function heads that implement different algorithms depending on context (Heimersheim & Janiak, 2023). The compositional requirements for induction-head formation (a core building block of many circuits) have also been studied (Singh et al., 2024; Crosbie & Shutova, 2025). The iterative

description–validation loop is rarely made explicit in published work; only the final interpretation is presented (Sharkey et al., 2025). However, the manual approach is labor-intensive, relies on subjective choices about which components to test, and does not scale to the thousands of edges present even in moderately sized models.

### A.1.3 Automated Circuit Discovery

To overcome the scalability limitations of manual analysis, automated methods algorithmically search for sparse, task-relevant subgraphs (Figure 2b). These methods generally follow a common pipeline: start from the full computational graph, score or mask each edge according to a relevance criterion, select a sparse subgraph, and validate the result via patching or ablation. Existing methods fall into several families:

**Iterative pruning.** Conmy et al. (2023) introduced ACDC, which systematizes the manual workflow into three steps (select behavior, define computational graph, patch activations) and automates the third step. ACDC greedily prunes edges in reverse topological order: for each candidate edge, it removes the edge, patches all pruned edges with corrupted activations, and measures the change in a divergence metric (e.g., KL divergence). If the change falls below a threshold $\tau$, the edge is permanently removed. ACDC validated its approach by rediscovering known circuits (e.g., all 5 component types in the Greater-Than circuit, selecting 68 of 32,000 edges in GPT-2 Small). However, ACDC automates only localization, not the subsequent interpretation of component roles (Sharkey et al., 2025). It is greedy and threshold-sensitive (the order of parent iteration can affect results) and can under-recover negative (suppressive) components when optimizing a single metric (Conmy et al., 2023). It also requires an independent forward pass for every edge tested, making it computationally expensive for larger models (Rai et al., 2024).

**Attribution-based methods.** Attribution patching approximates edge importance through gradients rather than testing each edge individually. EAP (Syed et al., 2024) computes first-order approximations of the effect of patching each edge, requiring only two forward passes and one backward pass to score all edges simultaneously, orders of magnitude faster than ACDC. Although the correlation between attribution and activation patching scores is modest ($R^2 \approx 0.27$ on the Docstring task (Syed et al., 2024)), EAP outperforms ACDC on circuit recovery because edge *ranking* matters more than magnitude accuracy. EAP-IG (Hanna et al., 2024) refines these estimates using integrated gradients to mitigate zero gradients at the linearization point, and AtP* (Kramár et al., 2024) further addresses failure modes from attention saturation and cancellation between direct and indirect effects. On the MIB benchmark, EAP-IG generally achieves the strongest performance among automated methods (Mueller et al., 2025). These methods trade some faithfulness for much greater scalability but rely on a first-order approximation of the true patching effect, and it remains unclear whether this is adequate for all settings (Sharkey et al., 2025).

**Differentiable masking.** Edge pruning methods formulate circuit discovery as differentiable optimization over continuous edge masks. Bhaskar et al. (2024) learn a mask for each edge that is jointly optimized to minimize a faithfulness loss subject to a sparsity penalty, using a disentangled residual stream that retains all previous activations for per-edge masking. Their method finds circuits in GPT-2 with less than half the edges of ACDC/EAP circuits at equal faithfulness, and scales to CodeLlama-13B (Rozière et al., 2024), $100\times$ larger than models typically tackled by automated methods. Yu et al. (2025) propose DiscoGP, which jointly prunes both edges and weight parameters at neuron-level granularity, and demonstrate that "canonical circuits" identified by prior work have very low functional faithfulness when evaluated in isolation. These approaches avoid the greedy ordering dependence of ACDC but introduce their own hyperparameters (sparsity weight, learning rate, mask initialization).

**Single-forward-pass methods.** Franco et al. (2026) propose ACC++, which traces information flow through the QK attention matrices via SVD decomposition in a single forward pass, without requiring activation patching or multiple model evaluations. This yields per-prompt circuits at low computational cost but is limited to attention-mediated pathways.

**Subspace-level methods.** Standard activation patching replaces entire hidden representations, implicitly assuming a localist mapping between causal variables and disjoint sets of neurons. Distributed Interchange Interventions (DII) instead intervene in rotated subspaces of the representation, enabling more fine-grained analysis when features are distributed across neurons (Geiger et al., 2024). Distributed Alignment Search (DAS) extends this by learning the rotation matrix and the $k$-dimensional subspaces that best align with high-level causal variables in a supervised fashion.

Despite their differences in search procedure, these methods share a common assumption: the discovered circuit is often treated as if it were the unique explanation for the target behavior (Sharkey et al., 2025; Méloux et al., 2025a). The output of each method depends on numerous choices: threshold, metric, patching direction (clean-to-corrupt vs. corrupt-to-clean), sparsity penalty, and data distribution; yet the resulting circuit is typically reported as a singular, fixed object.

### A.1.4 Metrics and Evaluation

Evaluating a discovered circuit requires assessing whether it faithfully captures the model's computation for the target behavior. Four properties are commonly considered (Wang et al., 2023; Shi et al., 2024):

- **Faithfulness**: does the circuit reproduce the full model's behavior on the target task?

- **Completeness**: does the circuit capture all components the model uses for the task?

- **Sufficiency**: does the circuit alone produce correct outputs when all other components are ablated?

- **Minimality**: is the circuit as sparse as possible while remaining faithful?

**Metric choice.** After performing an intervention, the resulting change in model output must be quantified. Three metrics are common (Rai et al., 2024; Heimersheim & Nanda, 2024). *Raw probability* or *logit* measures the change in the model's confidence on the correct token before and after patching. However, this can fail to detect *negative* components (those that suppress incorrect answers rather than promoting correct ones) because a component that boosts both the correct and incorrect token equally will show no change in raw probability (Zhang & Nanda, 2024; Heimersheim & Nanda, 2024). *Logit difference* (Wang et al., 2023) measures the change in the gap between the correct and incorrect logits, and is generally recommended because it controls for components that promote both targets and is linear in the residual stream. *KL divergence* (Conmy et al., 2023; Zhang & Nanda, 2024) compares the full output distribution before and after intervention, capturing changes beyond a single token pair. *Path patching* (Goldowsky-Dill et al., 2023) extends activation patching to isolate the effect of specific computational pathways rather than individual components.

Critically, Méloux et al. (2025b) show that varying the corruption method and evaluation metric leads to *disparate interpretability results*: for the IOI task, different method–metric combinations detect different, incomplete subsets of attention heads, with no single configuration recovering the full known circuit. Miller et al. (2024) further show that faithfulness scores are highly sensitive to seemingly insignificant changes in ablation methodology, concluding that such scores reflect the methodological choices of researchers as much as the actual components of the circuit.

**Evaluation granularity.** Most studies evaluate at the *source level*, preserving all outgoing connections from selected components, a weaker test than *edge-level* evaluation, which preserves only the specific connections identified by the discovery algorithm. Source-level evaluation can inflate apparent circuit quality by allowing components to exploit connections that the discovery method did not identify (Miller et al., 2024).

**Overlap vs. faithfulness.** Hanna et al. (2024) demonstrate that high component *overlap* between two circuits does not imply high *faithfulness*: when overlap is moderate, it does not predict faithfulness, leading them to recommend that "when comparing circuits, measuring overlap is no substitute for measuring faithfulness." Relatedly, Goldowsky-Dill et al. (2023) note that path patching measures *sufficiency* but not *completeness*: with redundant components, a compact subset can achieve zero unexplained effect while missing contributors.

**Aggregate vs. per-input performance.** Aggregate faithfulness can mask per-input failures: Miller et al. (2024) show that within-task variance of circuit performance is large, and uit de Bos & Garriga-Alonso (2024) find that circuits generalize poorly to adversarial evaluation examples. This observation led Sharkey et al. (2025) to question whether first selecting a human-defined task and then discovering its circuit is an effective approach, since the task definition may not align with the model's internal computational decomposition. Many published circuits do not pass strict sufficiency-and-necessity tests (Shi et al., 2024; uit de Bos & Garriga-Alonso, 2024). More broadly, seemingly convincing interpretations can prove false (*interpretability illusions* (Bolukbasi et al., 2021; Makelov et al., 2024)), underscoring the need for rigorous validation.

Standardized benchmarks are beginning to address these concerns: Tracr (Lindner et al., 2023) provides synthetic transformers with known ground-truth circuits, and MIB (Mueller et al., 2025) evaluates on both synthetic and pre-trained models. However, concerns remain that results on synthetic benchmarks may not transfer to naturally trained transformers (Rai et al., 2024; Sharkey et al., 2025). For the operational definitions used in this study, see Appendix C.5.

### A.1.5 Shortcomings and Open Challenges

Beyond evaluation, circuit discovery faces several structural challenges that suggest recovered circuits may not uniquely identify the underlying mechanism.

**Non-uniqueness and instability.** Sharkey et al. (2025) identify non-uniqueness as a key open problem. Multiple lines of evidence support this concern. At the method level, ACDC's output depends on threshold, metric, and the order of edge iteration (Conmy et al., 2023), and gradient-based attribution scores exhibit high variance across inputs (Méloux et al., 2025b). At a more fundamental level, Méloux et al. (2025a) exhaustively enumerate all valid explanations in small MLPs trained on Boolean functions and find substantial non-identifiability: multiple circuits with zero error, multiple interpretations per circuit, and multiple causally aligned algorithms per network, with the number of valid explanations growing dramatically with model size. Empirically, Franco et al. (2026) show that even within a single task (IOI), different prompt templates induce systematically different circuits that cluster into "prompt families," with the dominant source of variation being model-dependent. Haklay et al. (2025) further show that circuits are position-specific: assuming position-invariance, as most methods do, leads to low precision and recall when edge importance is aggregated across positions. Even on a single task (modular addition), small changes to hyperparameters and initialization can induce qualitatively different algorithms (e.g., the "Clock" vs. "Pizza" algorithms (Zhong et al., 2023)), and models sometimes implement multiple imperfect copies in parallel.

However, the picture is not entirely negative. Tigges et al. (2024) track circuits across 300 billion tokens of training in the Pythia suite (70M–2.8B parameters) and find that although individual components may change over time, the overarching *algorithm* remains stable and tends to replicate across model scale. Chughtai et al. (2023) reach a similar conclusion on group composition tasks: networks consistently implement the same representation-theoretic algorithm (weak universality), but the specific representations learned vary across random seeds (against strong universality).

**Redundancy and backup behavior.** McGrath et al. (2023) described the *Hydra effect*: when important attention heads are ablated, backup heads compensate by increasing their contribution, indicating built-in redundancy that complicates the notion of a unique circuit. Wang et al. (2023) observed similar backup behavior in the IOI circuit, where backup name-mover heads use a qualitatively different mechanism from the negative heads they compensate for (McDougall et al., 2024). Ortu et al. (2024) showed that factual recall and counterfactual reasoning engage competing mechanisms that dynamically trade off depending on context, and Dutta et al. (2024) documented functionally interchangeable parallel pathways at scale in Llama-2 7B, where multiple circuits simultaneously write the answer token during chain-of-thought reasoning. More broadly, models can implement multiple algorithms in tandem for the same task (Zhong et al., 2023; Nanda et al., 2023), and superficially identical behavior can arise from qualitatively different internal processes (Mahaut & Franzon, 2025). Noising and denoising interventions interact differently with redundant structure: in AND-like serial circuits, noising finds all components while denoising finds only the output; in OR-like parallel (backup) circuits, the reverse holds (Heimersheim & Nanda, 2024).

These empirical observations have theoretical grounding. Rohweder et al. (2026) show that under hierarchical data generation, gradient descent's implicit bias toward symmetric solutions distributes predictive power across multiple components, guaranteeing that ablation of one can be compensated by others. At the parameter level, Bushnaq et al. (2024) connect this redundancy to degeneracy in the loss landscape, showing that mechanistically distinct parameter configurations can achieve equivalent loss, paralleling the biological concept of *degeneracy*, in which structurally different elements perform the same function (Edelman & Gally, 2001).

**Component reuse across tasks.** Despite the non-uniqueness of circuits *within* a task, there is growing evidence that models reuse circuit components *across* tasks. Merullo et al. (2024) demonstrate that the IOI circuit and a Colored Objects circuit in GPT-2 Medium share approximately 78% of their most important attention heads, despite having no linguistic overlap, suggesting task-general algorithmic building blocks (detect duplication → inhibit/gather → copy). Lan et al. (2024) find that semantically related sequence continuation tasks rely on shared circuit subgraphs in both GPT-2 Small and Llama-2-7B. Mondorf et al. (2025) confirm both shared and task-specific substructure across string-edit operations, and show that circuits can be composed via set operations to explain more complex behaviors. Ferrando & Costa-Jussà (2024) show that the SVA circuit in Gemma 2B is highly consistent across English and Spanish, driven by a language-independent "subject number" direction. Sun (2025) show that a model's ability to apply consistent circuitry across input variants (circuit stability) predicts length, structural, and compositional generalization. Nainani et al. (2024) further show that the IOI circuit generalizes to prompt variants where its hypothesized algorithm should fail, reusing 100% of nodes with only additional input edges, though they also uncover an artifact of mean-ablation knockout ("S2 Hacking") that can inflate circuit performance relative to the full model.

**Scalability and scope.** Most circuit studies have been conducted on small models (GPT-2 Small, Pythia-70M, toy models), with only limited demonstrations on larger architectures (Lieberum et al., 2023). The heavy reliance on human interpretation for hypothesis generation and validation poses a scalability bottleneck (Rai et al., 2024). Furthermore, the tasks studied have been deliberately simple and amenable to mechanistic analysis; Sharkey et al. (2025) caution that this "streetlight interpretability" risks developing methods and insights that do not transfer to more complex, safety-relevant settings.

**The missing test.** Prior work has documented that circuits vary with extraction *method* (Conmy et al., 2023; Syed et al., 2024; Bhaskar et al., 2024), with *hyperparameters* such as metric and corruption method (Méloux et al., 2025b), with *prompt template* (Franco et al., 2026), and with *token position* (Haklay et al., 2025). However, to our knowledge, whether circuits for the same task within a single model vary when the *input distribution* changes (holding the task and method fixed) remains largely unexamined. Token frequency is a natural perturbation axis for this question: it is governed by Zipfian distributions (Zipf, 1949), influences embedding geometry (Zhou et al., 2021; Merullo et al., 2025) and few-shot reasoning accuracy (Razeghi et al., 2022), and affects task performance even when the task logic is frequency-invariant (Niu et al., 2025). At the neuron level, Liu et al. (2025) show that rare-token processing in Pythia and GPT-2 emerges through *distributed specialization*: coordinated but spatially scattered MLP neurons that operate within the shared architecture rather than forming discrete modules. Whether this distributed-specialization finding extends to the circuit level, i.e., whether structurally distinct circuits emerge for different frequency bands of the same task, remains an open question.

## A.2 Activation Patching

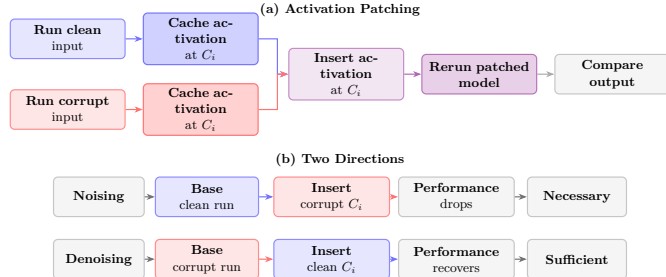

Figure 11: Activation patching and its two main directions. **(a)** Run the model on a clean input and on a corrupted input, cache the activation of a chosen component $C_i$ from each run, insert one into the other, rerun the model with that patched activation, and compare the output. **(b)** In *noising*, a corrupt activation is inserted into a clean base run; if performance drops, the component is necessary. In *denoising*, a clean activation is inserted into a corrupt base run; if performance recovers, the component is sufficient (Rai et al., 2024; Heimersheim & Nanda, 2024).

# B  Data Construction

Figure 12 provides an overview of the full data construction pipeline. The remainder of this section describes each stage in detail.

## B.1  Corpus and Token Frequency Extraction

Our frequency analysis requires exact token-level counts over the Pythia training corpus. We use the standard (undeduped) Pythia models (Biderman et al., 2023), which performed better in preliminary evaluations than the deduped variants. Accordingly, we use the undeduped Pile in its pre-shuffled form (`EleutherAI/pile-standard-pythia-preshuffled` on HuggingFace), the exact tokenized corpus used for pretraining.[8] The corpus contains approximately 300 B tokens (GPT-NeoX-20B (Black et al., 2022) tokenizer, vocabulary size 50,277; 50,063 token IDs have non-zero counts).

**Token identity preservation.** BPE encodes word boundaries through a leading-space prefix (Ġ), making Ġthe and the distinct entries. We preserve this distinction by operating on token IDs throughout and using the Ġ prefix for word-boundary classification (Sections B.2 and B.4).

## B.2  Vocabulary Filtering and Categorization

We transform raw counts into log-frequency space and categorize the vocabulary to isolate a usable word-level token pool.

**Frequency transformation.** Raw counts are converted to $f_{\log}(t) = \log_{10}(\text{count}(t)/N \times 10^6)$ ($N = 300{,}039{,}168{,}000$), where 0 corresponds to one occurrence per million tokens. The resulting distribution is smooth and unimodal with no natural clusters, motivating equal-width bands in log-frequency space (Section B.3).

**Token categorization.** The raw vocabulary includes punctuation, digits, code fragments, and subword pieces that cluster at particular frequencies. To ensure frequency is the primary axis of variation, we classify each token into eight categories using Unicode character analysis and the BPE leading-space prefix Ġ (Sennrich et al., 2016) (Table 11).

Table 12 summarizes the resulting category distribution; Figure 13 shows the frequency and length profiles across categories.

---

[8]https://huggingface.co/datasets/EleutherAI/pile-standard-pythia-preshuffled

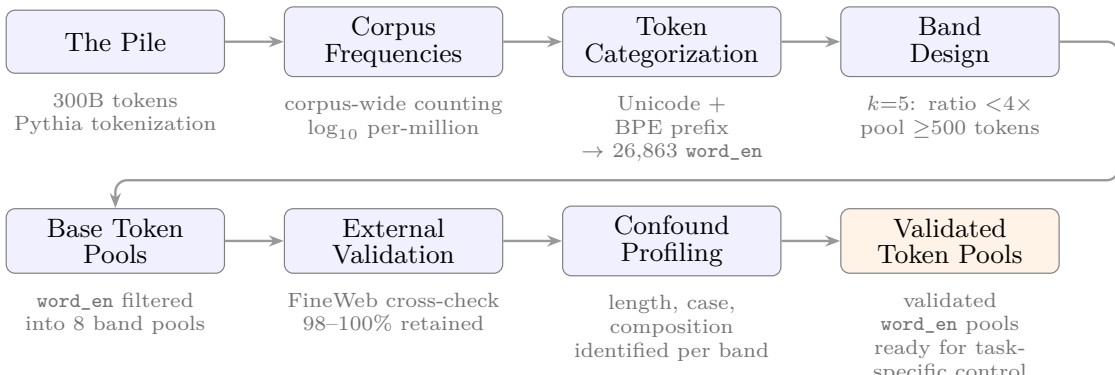

Figure 12: Data construction pipeline. Each stage (blue) transforms the output of the previous step into increasingly refined token pools; the final output (orange) is a set of frequency-controlled, validated token pools with identified confounds, ready for task-specific control (e.g., lowercase filtering and length-matching for LSC).

| Category | Rule |
|----------|------|
| **word_en** | Space prefix + all letters + all Latin script + all ASCII |
| **subword_en** | No space prefix + all letters + all Latin + all ASCII |
| word_other | Space prefix + all letters + non-Latin script or non-ASCII Latin (e.g., accented characters) |
| subword_other | No space prefix + all letters + non-Latin or non-ASCII Latin |
| numeric | All content characters are digits |
| punctuation | All content characters are punctuation or symbols |
| whitespace | Pure whitespace, control characters, or BPE whitespace markers |
| mixed | Multiple character groups present (e.g., letters + digits) |

Table 11: Token categorization rules. Each token is classified based on its BPE space prefix and the Unicode properties of its content characters (after stripping the prefix). **word_en** tokens form the primary experimental pool; the distinction between word_en and subword_en (and between word_other and subword_other) rests entirely on the presence of the space prefix, which marks word-initial position in BPE. Note that this is a heuristic filter: word_en admits loanwords, proper names, and tokens shared across Latin-script languages, not only English words.

## B.3 Frequency Band Design

Although token frequency is a continuous variable, comparing circuits across the spectrum requires discrete bands. Since the log-frequency distribution is unimodal and lacks clear natural clusters (Section B.2), we use equal-width bands in log-frequency space, so each band spans the same multiplicative frequency ratio.

**Core range.** We define the core experimental range as the 1st to 99th percentile of the word_en log-frequency distribution: $[-0.43, 2.27]$, spanning 2.69 log-units (a $493\times$ frequency ratio end-to-end). The tails beyond this range contain qualitatively different populations: the bottom 1% scatters 269 tokens across 2.6 log-units (too sparse for controlled experimentation), while the top 1% contains 269 ultra-common function words. Both tails are retained as separate exploratory conditions.

**Selecting $k$.** We sweep $k = 3, \ldots, 8$ and select the largest value that keeps the within-band frequency ratio below $4\times$ while leaving at least 500 word_en tokens per band. $k = 5$ is the largest value satisfying both constraints. Each band spans 0.54 log-units ($3.46\times$ within-band ratio), and the smallest core band (very_high) contains 821 word_en tokens. At $k = 4$, bands are too wide ($4.7\times$ ratio); at $k = 6$, the smallest pool becomes too small (513 tokens).

| Category | Count | % | Med. log-freq | Med. length |
|---|---|---|---|---|
| **word_en** | 26,863 | 53.7 | 0.53 | 6 |
| subword_en | 16,431 | 32.8 | 0.46 | 4 |
| mixed | 2,201 | 4.4 | 0.45 | 3 |
| numeric | 2,042 | 4.1 | 0.52 | 3 |
| punctuation | 1,418 | 2.8 | 0.65 | 3 |
| subword_other | 829 | 1.7 | 0.47 | 2 |
| word_other | 187 | 0.4 | 0.43 | 2 |
| whitespace | 92 | 0.2 | 1.22 | 3 |
| Total | 50,063 | 100 | | |

Table 12: Token vocabulary categories. Each of the 50,063 tokens with non-zero training frequency is classified into one of eight categories based on Unicode character content. **word_en** tokens (space-prefixed, all-Latin, ASCII-only) form the majority and constitute the experimental pool from which frequency bands are populated.

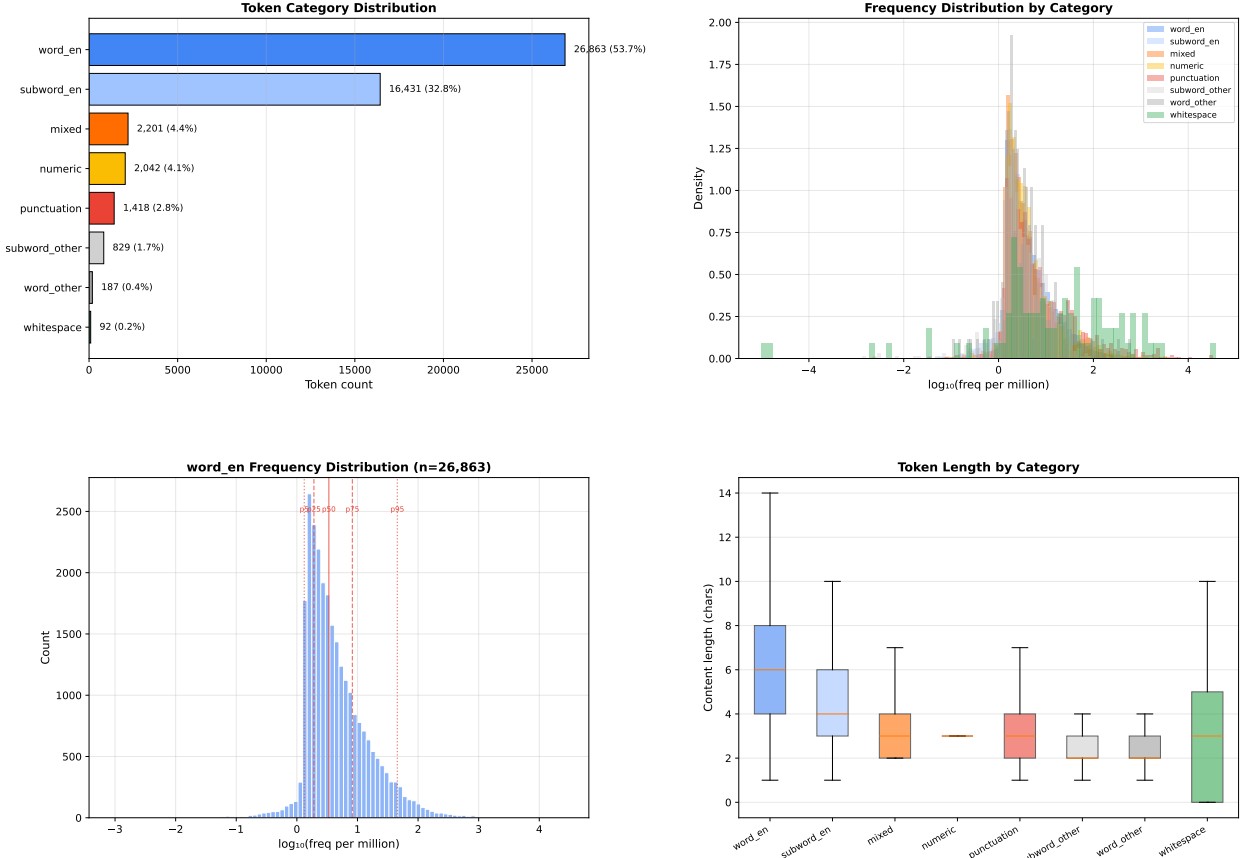

Figure 13: Token vocabulary overview. **Top left:** category distribution; word_en tokens dominate the vocabulary. **Top right:** log-frequency distributions by category. **Bottom left:** word_en frequency distribution with percentile markers (p1, p25, p75, p99) defining the core range for band construction. **Bottom right:** character length distributions by category.

**Final scheme: eight conditions.** The five core bands span the gradient from very_low to very_high. Two exploratory tail bands and one frequency-weighted control complete the design (Table 13).

| Band | Type | Log-freq range | Center | word_en tokens |
|------|------|----------------|--------|----------------|
| bottom_tail | exploratory | $[-3.05, -0.43)$ | | 269 |
| very_low | core | $[-0.43, 0.11)$ | $-0.16$ | 935 |
| low | core | $[0.11, 0.65)$ | 0.38 | 14,768 |
| medium | core | $[0.65, 1.19)$ | 0.92 | 6,917 |
| high | core | $[1.19, 1.73)$ | 1.46 | 2,884 |
| very_high | core | $[1.73, 2.27]$ | 2.00 | 821 |
| top_tail | exploratory | $(2.27, 4.44]$ | | 269 |
| control | baseline | $[-3.05, 4.44]$ | | 26,863 |

Table 13: Frequency band definitions. Core bands use equal-width intervals of 0.54 log-units (3.46× ratio). The control band samples from the full word_en pool weighted by pretraining frequency, representing the uncontrolled baseline.

Across the five core bands, center frequencies span a ~145× range, from about 0.7 per million to about 100 per million; the 493× ratio above is the band-edge span across the full core range.

The control condition samples from the entire word_en pool weighted by pretraining frequency. It reflects the natural frequency mix encountered during training and serves as an uncontrolled baseline against which frequency-specific circuits can be compared.

## B.4 Token Pool Construction and Validation

Each band's word_en tokens are exported to pool files. Before use, we validate these pools against an external corpus and profile potential confounds.

**External validation.** We validate word_en tokens against FineWeb (Penedo et al., 2024) by comparing standalone word frequency with substring occurrence; tokens used mainly as substrings are treated as likely subword fragments. Core bands retain 98.4%–99.7% of tokens (85.1%–100% across all bands), confirming that the vast majority are genuine English words.

**Confound profiling.** We profile three task-agnostic confounds at the token-pool level (Figure 14): (i) **character length**: lower-frequency tokens tend to be longer; addressed by length-matching during dataset generation (Section B.5); (ii) **capitalization**: controlled in LSC by restricting to lowercase tokens; (iii) **word vs. subword composition**: already controlled by the word_en filter. The validated pools serve as input for all downstream task generators.

## B.5 LSC Task and Dataset Generation

**Task definition.** Literal Sequence Copying (LSC), introduced by Niu et al. (2025), tests a language model's ability to perform induction: recognizing a repeated token pattern and copying what followed it. Niu et al. (2025) showed that induction-head performance degrades for rare tokens when tokenizer index is used as a frequency proxy. We adopt their task and extend it with the controlled frequency bands and confound controls described in Sections B.3–B.4. Each sequence has the structure:

$$\underbrace{S_1, S_2, S_3, S_4, S_5}_{\text{source}}, \; T, \; \underbrace{R_1, R_2, \ldots, R_{10}}_{\text{distraction}}, \; \underbrace{S_1, S_2, S_3, S_4, S_5}_{\text{repetition}}$$

yielding 21 tokens per sequence. At position 20 (the second occurrence of $S_5$), the model should predict $T$, the token that followed $S_5$ in the source segment. The 10-token distraction segment forces the model to use long-range attention (induction heads) rather than local bigram statistics. All 16 unique tokens in each sequence ($S_{1-5}$, $T$, $R_{1-10}$) are drawn from the same frequency band, sampled without replacement.

**Token pool preparation.** From the validated pools (Section B.4), we prepare LSC-specific token pools in two variants. The *matched* variant applies two additional filters: (i) restrict to lowercase tokens only (elimi-

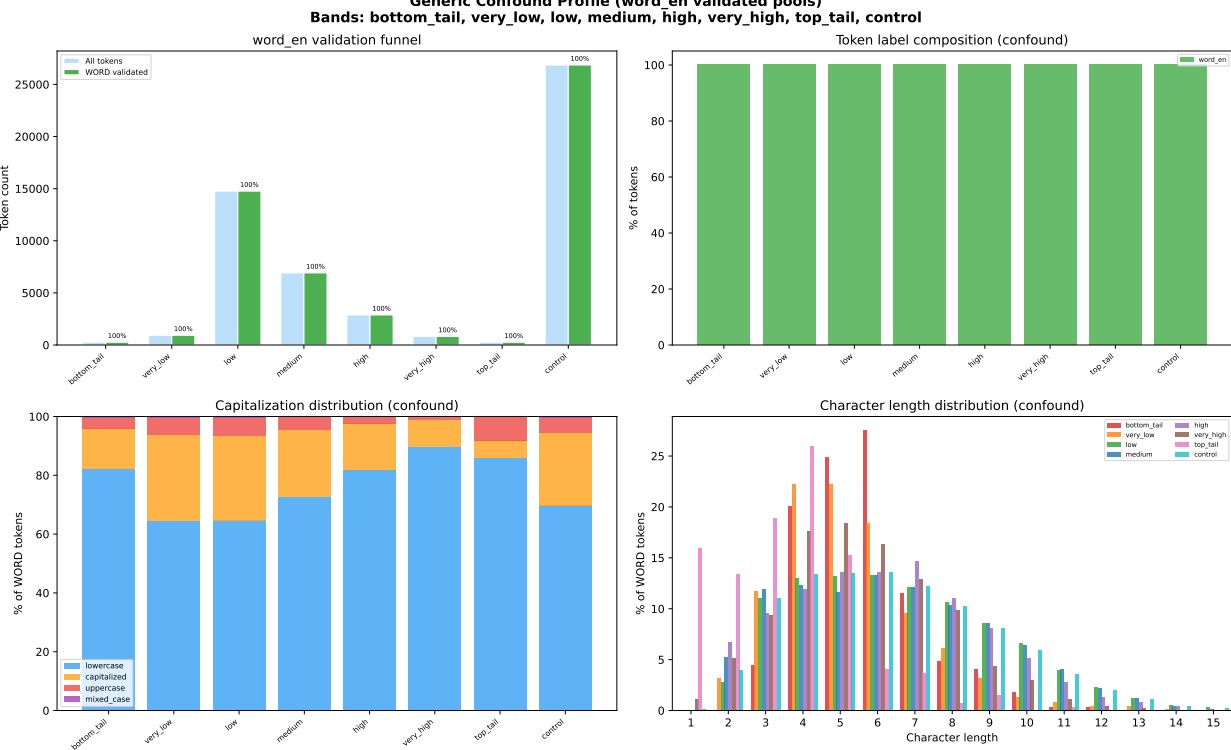

Figure 14: Confound profile across frequency bands for word-validated pools. **Top left:** FineWeb validation retention by band. **Top right:** word-status composition after validation. **Bottom left:** capitalization distribution. **Bottom right:** character length distribution. The latter two motivate the lowercase-only filter and length-matching used in LSC.

nating capitalization confounds), and (ii) match character length distributions across bands by subsampling to the intersection of lengths available in every participating band. Four of the five core bands (low through very_high) satisfy this requirement, yielding 703 tokens per band. The very_low band contains only 97 lowercase word_en tokens after filtering, too few for reliable length-matching, and is therefore excluded from the matched pools; it remains available in the unmatched variant. The matched *control* pool is the union of the four matched core bands (2,812 tokens), sampled with pretraining-frequency weights to reflect the model's natural frequency mix. The *unmatched* variant applies only the lowercase filter to the full validated pools, retaining all eight bands (pool sizes range from 97 for very_low to 15,269 for control) and serves as a robustness check.

**Dataset generation.**   For each band, we generate 1,500 sequences split 70/15/15 into train (1,050), validation (225), and test (225) sets. We produce three independent draws, the minimum needed to measure circuit extraction stability, using deterministic seeds to ensure exact reproducibility. Comparing results across draws quantifies within-band sampling noise and tests whether recovered circuits are robust to the specific tokens selected from each pool. The matched variant generates datasets for five conditions (the four matched core bands plus control); the unmatched variant generates seven conditions (all five core bands, top_tail, and control; bottom_tail is excluded due to its pool size of only 11 tokens).

**BOS handling.**   The raw dataset stores 21-token sequences without a beginning-of-sequence token. BOS is prepended at inference time by the modeling framework (TransformerLens Nanda & Bloom, 2022 uses `prepend_bos=True`), shifting all position indices by +1. With BOS, the prediction target moves from position 20 to position 21.

# C Circuit Discovery Details

This appendix details the circuit discovery pipeline summarized in Section 3.3, which produces 75 circuits across five Pythia models, five conditions (four frequency bands plus control), and three draws per frequency band in three phases:

1. **Threshold selection** (Section C.3): a Pareto sweep on the control band identifies candidate ACDC thresholds, from which one threshold $\tau^*$ is selected per model.

2. **Circuit extraction** (Section C.4): ACDC runs at $\tau^*$ across all five conditions and three draws per model, with evaluation on the held-out test split.

3. **Post-hoc validation** (Section C.6): random baselines test whether the discovered circuits outperform matched arbitrary edge subsets.

## C.1 Corruption Procedure

Each clean sequence has the form [BOS] $S_1 \ldots S_5\ T\ R_1 \ldots R_{10}\ S_1 \ldots S_5$ (22 tokens including BOS). The corrupted sequence is identical through position 16 and replaces positions 17–21 (the repeated prefix that triggers induction) with five tokens sampled without replacement from the same-band pool, excluding tokens in the clean sequence. The diverge index is set to 17 so that AutoCircuit restricts activation patching to later positions. Each example is paired with a wrong-answer token from the band pool, as required by the AutoCircuit `PromptDataset` API.

## C.2 ACDC Configuration

**Pythia's parallel architecture.** Pythia models use the GPTNeoX architecture (Biderman et al., 2023), which employs *parallel* transformer blocks: attention and MLP both receive the same layer-normalized input and their outputs are summed into the residual stream,

$$\mathbf{x}_{l+1} = \mathbf{x}_l + \mathrm{Attn}(\mathrm{LN}(\mathbf{x}_l)) + \mathrm{MLP}(\mathrm{LN}(\mathbf{x}_l)).$$

Attention and MLP within the same layer therefore cannot interact directly; the computational graph used for circuit discovery must respect this constraint.

**Algorithm and library choice.** In pilot experiments, EAP (Syed et al., 2024) and EAP-IG (Hanna et al., 2024) produced unstable circuits on our LSC task, consistent with the high intrinsic variance reported by Méloux et al. (2025b). ACDC (Conmy et al., 2023) produces deterministic circuits for a given input set and threshold, making it the appropriate choice for multi-condition comparison. We nevertheless include a cross-method analysis as a robustness check (Appendix C.9). We run ACDC through AutoCircuit (Miller et al., 2024) rather than the original codebase, which produced metric collapse on Pythia models. AutoCircuit correctly handles parallel blocks via the `parallel_attn_mlp` flag and also makes threshold sweeps practical through faster edge patching (Section C.3).

**Model loading and ACDC settings.** Models are loaded through TransformerLens with `fold_ln=True`, `center_writing_weights=True`, and `center_unembed=True` (folding layer-norm into adjacent weights), with hook flags `use_attn_result`, `use_attn_in`, and `use_hook_mlp_in` enabled. ACDC uses factorized tree patching with resample ablation, joint QKV treatment, and output slicing at the last sequence position; non-circuit edges carry activations resampled from the corrupted distribution (Section C.1). All computations use FP32; BF16 produced numerical discrepancies on Pythia.

## C.3 Threshold Selection via Pareto Analysis

**Motivation.** ACDC's pruning threshold $\tau$ controls the sparsity/faithfulness trade-off. No reference circuit for LSC on Pythia exists, so we select $\tau$ from sweep results using four criteria: *minimality* (few edges), *faithfulness* (low KL to the base model), *sufficiency* (task performance), and *necessity* (performance degrades when the circuit is removed).

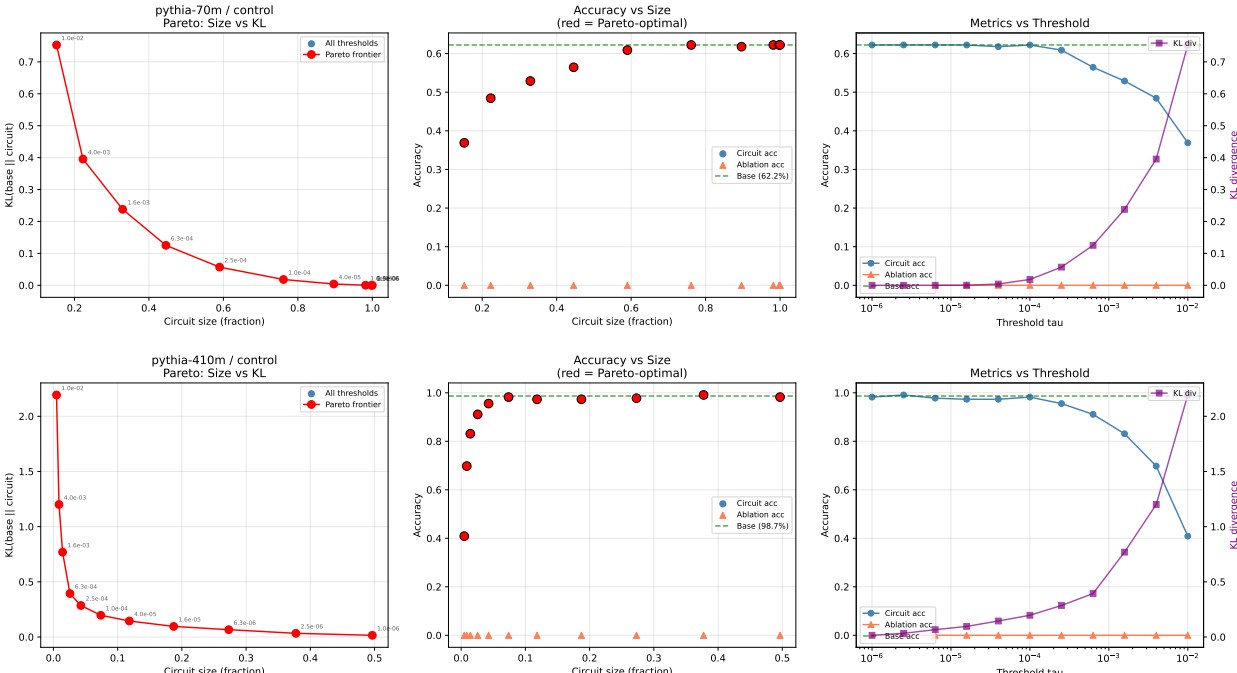

Figure 15: Pareto sweep results for two representative Pythia models on the control band (top: 70m; bottom: 410m). Results for 160m, 1b, and 1.4b are qualitatively similar. **Left:** Pareto frontier over circuit size (edge fraction) and $\mathrm{KL}(p_{\mathrm{base}}\|p_{\mathrm{circuit}})$; threshold values annotated. **Center:** Circuit and ablation accuracy vs. circuit size; red points are Pareto-optimal. **Right:** Accuracy and KL vs. threshold $\tau$. Pythia-70m has substantially lower base accuracy; see the *Pythia-70m* note below Table 14.

**Sweep design.** We sweep 11 log-uniformly spaced thresholds from $10^{-2}$ to $10^{-6}$ on the *control* band using draw 1. For each threshold, ACDC trains on 256 examples sampled from the training split (seed 42) and the resulting circuit is evaluated on the full validation split (225 examples, seed 123). We use the control band because it mixes frequency-weighted tokens from all four core bands (Section B.5).

**Pareto frontier.** For each model, we compute the two-objective Pareto frontier over edge fraction and $\mathrm{KL}(p_{\mathrm{base}}\|p_{\mathrm{circuit}})$; frontier points are non-dominated in both objectives.

**Semi-automated threshold selection.** Automated heuristics filter Pareto-optimal points by minimum quality criteria ($\geq 80\%$ accuracy retention, $\mathrm{KL} < 0.5$) and an expected size range for each model. A human reviewer then selects one threshold $\tau^*$ per model from the filtered candidates (Figure 15; Table 14).

Table 14: Selected ACDC thresholds per model. All points lie on the Pareto frontier and achieve 0% ablation accuracy. Retention is circuit accuracy divided by base model accuracy on the control band validation split.

| Model | $\tau^*$ | Edge % | $n_{\mathrm{edges}}$ | KL | Circ. acc. (%) | Retention (%) | Abl. acc. (%) |
|---|---|---|---|---|---|---|---|
| Pythia-70m | $1.58 \times 10^{-3}$ | 32.9 | 436 | 0.238 | 52.9 | 85.0 | 0.0 |
| Pythia-160m | $6.31 \times 10^{-4}$ | 12.2 | 1,396 | 0.284 | 93.3 | 96.3 | 0.0 |
| Pythia-410m | $2.51 \times 10^{-4}$ | 4.3 | 3,444 | 0.285 | 95.6 | 96.8 | 0.0 |
| Pythia-1b | $1.58 \times 10^{-3}$ | 9.4 | 939 | 0.485 | 92.0 | 94.1 | 0.0 |
| Pythia-1.4b | $6.31 \times 10^{-4}$ | 2.6 | 2,097 | 0.499 | 92.4 | 94.1 | 0.0 |

**Pythia-70m.** Pythia-70m is an outlier: base accuracy is only 62.2% (vs. $\geq 95\%$ for larger models), yielding a denser circuit (32.9% of edges, 85.0% retention). Differences from larger models may reflect capacity limitations rather than qualitative differences in circuit organization.

## C.4 Circuit Extraction

Using the model-specific thresholds from Table 14, we run ACDC across all five conditions (four frequency bands plus control) and three independent draws for each of the five models, yielding 75 circuits in total. Each circuit trains on 256 examples from the band-specific training split (seed 42) and is evaluated on the test split (225 examples, seed 123); the validation split is reserved for threshold selection (Section C.3), preventing information leakage. Each circuit is also evaluated on the test splits of the other four bands, yielding a $5 \times 5$ cross-band transfer matrix for each model and draw (Section 4.2).

## C.5 Evaluation Metrics

We assess circuit quality along four dimensions (Wang et al., 2023):

**Faithfulness.** We measure faithfulness as $\mathrm{KL}(p_{\mathrm{base}} \| p_{\mathrm{circuit}})$ at the final sequence position (Conmy et al., 2023).

**Sufficiency.** Circuit accuracy (top-1, top-5, top-10) and mean correct-token probability when non-circuit edges carry resample-ablated activations. Resample ablation upper-bounds standalone performance (Yu et al., 2025).

**Necessity (completeness).** Ablation accuracy: task performance when only the circuit's complement is retained; a necessary circuit yields near-zero ablation accuracy.

**Minimality.** Edge fraction $n_{\mathrm{edges}}/n_{\mathrm{total}}$. Base-model metrics are precomputed with HuggingFace Transformers and cached across phases.

## C.6 Random Baselines

Each circuit is compared against $K = 100$ random edge sets of identical size, reporting the $z$-score and percentile rank. All 75 circuits achieve a percentile rank of 100%.

## C.7 Implementation Details

All experiments use fixed random seeds (seed 42 for training, seed 123 for evaluation) and deterministic CUDA settings on NVIDIA A100 80GB GPUs. Batch sizes are adapted to GPU memory (256 for 70m/160m, 128 for 410m, 96 for 1b). The full pipeline requires approximately 736 GPU-hours ($\sim$289 sweep + $\sim$447 discovery), with individual ACDC runs ranging from $\sim$80 s (70m) to $\sim$3.5 h (1b), parallelized across four GPUs.

## C.8 Threshold Robustness

To confirm that the selected $\tau^*$ is not a cherry-picked operating point, we evaluate control-band circuits at three thresholds: $\tau_{\mathrm{low}}$ (one step looser than $\tau^*$ in the sweep), $\tau^*$ (the selected threshold), and $\tau_{\mathrm{high}}$ (one step stricter), spanning a 2–3$\times$ range in circuit size. For each threshold, the circuit is evaluated on the test splits of all five frequency bands, and cross-band transfer efficiency is computed as the ratio of mean off-control accuracy to on-control accuracy.

Transfer efficiency varies by at most 2.4 pp (looser) and 5.9 pp (stricter) relative to $\tau^*$ across all models. Pythia-70m shows lower absolute efficiency (73–77%), consistent with its capacity limitations (Section 5.1.2); all other models remain above 88% even at $\tau_{\mathrm{high}}$, confirming that phantom-specialization conclusions hold across plausible threshold choices.

Table 15: Cross-band transfer efficiency of control-band circuits at three threshold levels. $\tau_{\text{low}}$ retains more edges than $\tau^*$; $\tau_{\text{high}}$ retains fewer. Transfer efficiency is the ratio of mean off-control accuracy to on-control accuracy.

| Model | $\tau_{\text{low}}$ | | $\tau^*$ | | $\tau_{\text{high}}$ | |
|---|---|---|---|---|---|---|
| | Edges | Transf. Eff. | Edges | Transf. Eff. | Edges | Transf. Eff. |
| Pythia-70m | 590 | 0.770 | 436 | 0.755 | 295 | 0.731 |
| Pythia-160m | 2138 | 0.996 | 1396 | 0.993 | 904 | 0.965 |
| Pythia-410m | 5949 | 0.993 | 3444 | 0.984 | 2073 | 0.968 |
| Pythia-1b | 1425 | 0.967 | 939 | 0.954 | 551 | 0.919 |
| Pythia-1.4b | 3770 | 0.913 | 2097 | 0.935 | 1197 | 0.880 |

## C.9 Cross-Method Comparison: ACDC, EAP, and EAP-IG

As an independent check on whether ACDC's iterative search is responsible for the phantom specialization pattern, we compute EAP-IG (Hanna et al., 2024) edge-importance scores for each of the 75 extraction conditions (5 models × 5 bands × 3 draws), using integrated gradients over ten interpolation steps. For each condition we threshold the continuous scores to produce binary circuits at ten sizes ranging from 0.1× to 5.0× the corresponding ACDC edge count (7,500 evaluations total).

**Methodological caveat.** Top-$k$ thresholding selects individually important edges but does not guarantee a coherent subgraph; at the ACDC-matched size (1.0×), EAP-IG circuits achieve only 2–24% of base accuracy for the four larger models (Table 16), vs. 79–99% for ACDC. We therefore compare methods via *same-band advantage* (same-band minus cross-band accuracy) rather than transfer-efficiency ratios.

**No same-band advantage at any circuit size.** Across all five models and ten size multipliers, the same-band advantage never exceeds 4.6 pp, with inconsistent sign across models (Table 16). Small negative advantages for ACDC at 1.0× in Pythia-160m ($\Delta = -0.05$) and 410m ($\Delta = -0.04$) reflect the asymmetric transfer pattern (Section 4.5) and are within measurement noise. This pattern holds from 1% to 100% of edges, ruling out threshold-choice artifacts.

**Faithful-point comparison.** At the smallest size where EAP-IG achieves $\geq$50% of base accuracy, same-band advantage is +0.8 to +4.6 pp across all five models, consistent with ACDC. Larger models require 3.0–5.0× edges to reach this threshold, at which point 14–27% of all edges are included.

**Structural overlap with ACDC.** Jaccard overlap ranges from 0.28 to 0.60 at the 1.0× point, decreasing with model size (Spearman $r \approx -0.9$, $p < 0.001$). Two methods selecting largely different edges yet both finding no band specificity strengthens the many-to-one mapping conclusion.

**Plain EAP comparison.** We also evaluate plain EAP (Syed et al., 2024) (single gradient at zero mask value, without integrated gradients) under the same protocol. Table 17 and Figure 17 compare transfer efficiency and Jaccard overlap across all three methods. Plain EAP shows erratic transfer efficiency on larger models (61.7% for Pythia-410m, 16.7% for Pythia-1.4b), reflecting the instability of single-gradient estimates in deeper networks. EAP-IG (with integrated gradients over ten steps) is substantially more stable, with transfer efficiency ranging from 72.8% to 106.8% for models $\geq$160M. Despite these differences in circuit quality, neither gradient-based method shows a consistent same-band advantage at any circuit size, corroborating the ACDC finding.

**Summary.** Both EAP and EAP-IG corroborate the ACDC findings: gradient-based methods selecting largely different edges (Jaccard 0.25–0.60) find no band specificity. EAP-IG is the more reliable of the two gradient-based methods; plain EAP is unstable for models $\geq$410M. Three caveats apply: (i) top-$k$ thresholding does not produce coherent circuits, especially for deeper models; (ii) all three methods rely on activation-patching scores; replication with mask-optimization methods (e.g., Bhaskar et al., 2024) would

Table 16: EAP-IG circuit accuracy (same-band vs. cross-band) at representative sizes, compared to ACDC at 1.0×. Same-band accuracy is averaged over the four frequency bands and three draws where the circuit is evaluated on its extraction band; cross-band accuracy averages over all other-band evaluations. Δ is same-band minus cross-band accuracy (positive would indicate specialization). EAP-IG circuits require 3–5× more edges than ACDC to achieve comparable faithfulness.

| Method | Model | Size | Same Acc. | Cross Acc. | Δ | Jaccard |
|---|---|---|---|---|---|---|
| ACDC | 70M | 1.0× | 0.40 | 0.38 | +0.02 | — |
| ACDC | 160M | 1.0× | 0.81 | 0.86 | −0.05 | — |
| ACDC | 410M | 1.0× | 0.83 | 0.87 | −0.04 | — |
| ACDC | 1B | 1.0× | 0.83 | 0.82 | +0.01 | — |
| ACDC | 1.4B | 1.0× | 0.85 | 0.84 | +0.01 | — |
| *EAP-IG at ACDC-matched size* | | | | | | |
| EAP-IG | 70M | 1.0× | 0.24 | 0.23 | +0.01 | 0.60 |
| EAP-IG | 160M | 1.0× | 0.15 | 0.14 | +0.01 | 0.42 |
| EAP-IG | 410M | 1.0× | 0.09 | 0.10 | −0.01 | 0.32 |
| EAP-IG | 1B | 1.0× | 0.06 | 0.05 | +0.01 | 0.36 |
| EAP-IG | 1.4B | 1.0× | 0.02 | 0.05 | −0.03 | 0.28 |
| *EAP-IG at faithful size (≥50% base accuracy)* | | | | | | |
| EAP-IG | 70M | 1.5× | 0.36 | 0.35 | +0.01 | — |
| EAP-IG | 160M | 2.0× | 0.49 | 0.45 | +0.05 | — |
| EAP-IG | 410M | 3.0× | 0.58 | 0.55 | +0.03 | — |
| EAP-IG | 1B | 3.0× | 0.58 | 0.54 | +0.04 | — |
| EAP-IG | 1.4B | 5.0× | 0.60 | 0.58 | +0.02 | — |

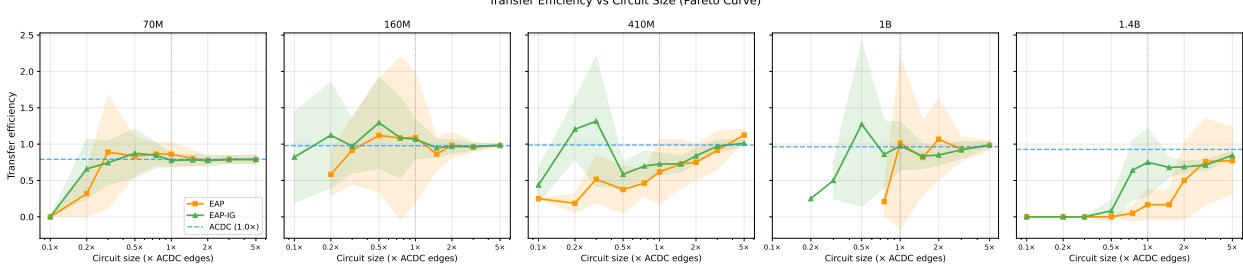

Figure 16: Same-band and cross-band accuracy of EAP-IG circuits as a function of circuit size (as a multiple of the ACDC edge count), averaged over draws and circuit bands. Each panel shows one model; the vertical dotted line marks the ACDC-matched point (1.0×). The two curves overlap closely at all sizes, confirming that the absence of band specificity is not sensitive to the threshold choice. Note that EAP-IG circuits require substantially more edges than ACDC to achieve comparable faithfulness (see text).

provide stronger resolution; (iii) faithful EAP-IG circuits include a large fraction of model edges. The causal interventions in Section 5.3.2 bypass circuit discovery entirely, directly confirming band interchangeability (IIA ≥ 0.95; Boundless DAS ≤2.5% of dimensions).

Table 17: Cross-method comparison of circuit discovery algorithms on LSC. Transfer efficiency is cross-band accuracy divided by same-band accuracy, averaged over all circuit bands and draws at the ACDC-matched size (1.0×). Jaccard is the edge overlap between EAP/EAP-IG and ACDC circuits.

| Method | Model | Transfer Eff. (%) | Jaccard vs. ACDC |
|--------|-------|-------------------|------------------|
| ACDC | 70M | 79.1 | — |
| ACDC | 160M | 97.8 | — |
| ACDC | 410M | 98.8 | — |
| ACDC | 1B | 96.4 | — |
| ACDC | 1.4B | 92.7 | — |
| EAP | 70M | 86.2 | 0.594 |
| EAP | 160M | 109.1 | 0.379 |
| EAP | 410M | 61.7 | 0.307 |
| EAP | 1B | 101.4 | 0.315 |
| EAP | 1.4B | 16.7 | 0.252 |
| EAP-IG | 70M | 77.6 | 0.598 |
| EAP-IG | 160M | 106.8 | 0.416 |
| EAP-IG | 410M | 72.8 | 0.323 |
| EAP-IG | 1B | 97.5 | 0.361 |
| EAP-IG | 1.4B | 75.0 | 0.281 |

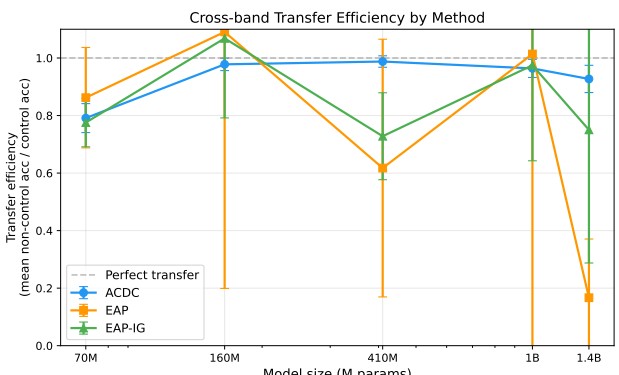

Figure 17: Transfer efficiency across all three circuit discovery methods (ACDC, EAP, EAP-IG) at the ACDC-matched circuit size (1.0×). Despite selecting largely different edges (Jaccard 0.25–0.60), all methods converge on the absence of band specificity, though plain EAP is unstable on larger models.

# D    Additional Functional Results

This appendix reports the full functional evaluation of all 75 circuits (5 models × 5 bands × 3 draws).

## D.1    Base Model Performance Across Frequency Bands

Pythia-70m shows a pronounced frequency gradient (30–67% top-1), whereas all larger models achieve $\geq 93\%$ with near-ceiling top-5 (Figure 18, Table 18).

The frequency effect is significant only for Pythia-70m (Kruskal-Wallis $H = 13.5$, $\eta^2 = 0.95$, $p_{\mathrm{BH}} = 0.036$; larger models $p_{\mathrm{BH}} \geq 0.054$).

## D.2    Circuit Size and Minimality

Circuit sparsity increases with scale: Pythia-70m retains 29–32% of edges, decreasing to 2.5–3.2% for Pythia-1.4b (Figure 19, Table 19).

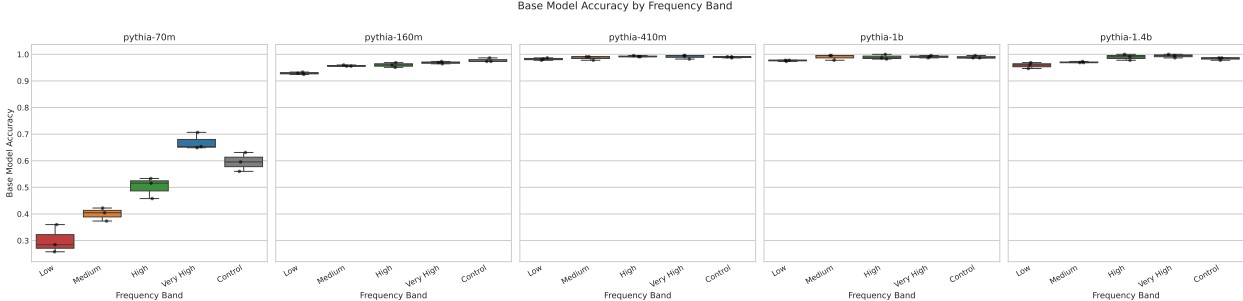

Figure 18: Base model top-1 accuracy across frequency bands. Pythia-70m shows a clear frequency gradient; larger models are near ceiling.

Table 18: Base model performance on LSC by model and frequency band. Values are top-1 accuracy (%) ± std over three draws. Top-5 accuracy is ≥99% for all models ≥160M and is omitted.

|  | Pythia-70m | | Pythia-160m | | Pythia-410m | | Pythia-1b | | Pythia-1.4b | |
|---|---|---|---|---|---|---|---|---|---|---|
| Band | Acc | ± | Acc | ± | Acc | ± | Acc | ± | Acc | ± |
| low | 30.1 | 5.3 | 92.9 | 0.4 | 98.2 | 0.4 | 97.6 | 0.3 | 95.9 | 1.1 |
| medium | 40.0 | 2.5 | 95.7 | 0.3 | 98.7 | 0.8 | 99.0 | 1.0 | 97.0 | 0.3 |
| high | 50.2 | 4.0 | 96.0 | 0.9 | 99.3 | 0.3 | 99.0 | 0.9 | 99.0 | 1.1 |
| very_high | 67.0 | 3.2 | 96.9 | 0.4 | 99.1 | 0.8 | 99.1 | 0.4 | 99.4 | 0.7 |
| control | 59.6 | 3.6 | 97.8 | 0.8 | 99.0 | 0.3 | 99.0 | 0.5 | 98.4 | 0.5 |

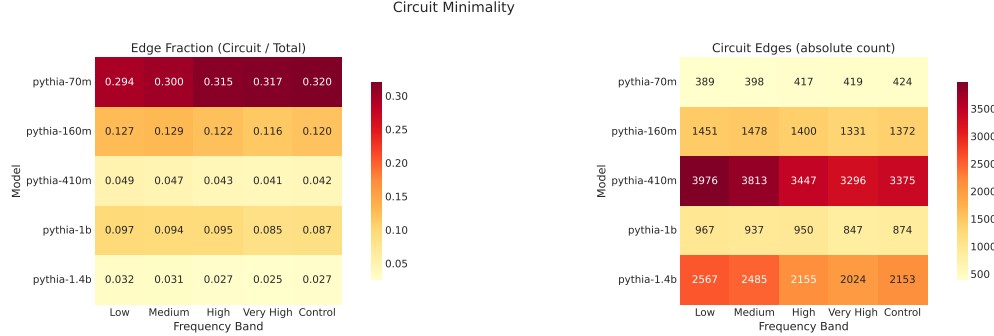

Figure 19: Circuit size (edge fraction) across models and frequency bands. Within-model variation across bands is not significant after FDR correction.

Circuit size does not vary significantly across bands after FDR correction (all $p_{\mathrm{BH}} \geq 0.064$).

### D.3 Same-Band Faithfulness and Sufficiency

Same-band circuit accuracy ranges from 24–55% (Pythia-70m, retention 80–89%) to ≥89% for larger models (retention ≥93%; Pythia-1.4b retains 83–94%).

KL divergence increases with model size (0.23–0.27 for Pythia-70m; 0.51–0.65 for Pythia-1.4b). A band effect on circuit accuracy is significant only for Pythia-70m ($H = 12.7$, $\eta^2 = 0.87$, $p_{\mathrm{BH}} = 0.047$).

Table 19: Circuit size by model and frequency band. Values are $n_{\text{edges}} \pm$ std over three draws; edge fraction (% of full graph) in the rightmost column per model.

| | Pythia-70m | | | Pythia-160m | | | Pythia-410m | | | Pythia-1b | | | Pythia-1.4b | | |
|---|---|---|---|---|---|---|---|---|---|---|---|---|---|---|---|
| Band | $n$ | $\pm$ | % | $n$ | $\pm$ | % | $n$ | $\pm$ | % | $n$ | $\pm$ | % | $n$ | $\pm$ | % |
| low | 389 | 8 | 29.4 | 1451 | 39 | 12.7 | 3976 | 138 | 4.9 | 967 | 14 | 9.7 | 2567 | 52 | 3.2 |
| medium | 398 | 7 | 30.0 | 1478 | 34 | 12.9 | 3813 | 106 | 4.7 | 937 | 61 | 9.4 | 2485 | 57 | 3.1 |
| high | 417 | 13 | 31.5 | 1400 | 49 | 12.2 | 3447 | 59 | 4.3 | 950 | 50 | 9.5 | 2155 | 31 | 2.7 |
| very_high | 419 | 4 | 31.7 | 1331 | 26 | 11.6 | 3296 | 173 | 4.1 | 847 | 56 | 8.5 | 2024 | 84 | 2.5 |
| control | 424 | 13 | 32.0 | 1372 | 24 | 12.0 | 3375 | 54 | 4.2 | 874 | 53 | 8.7 | 2153 | 128 | 2.7 |

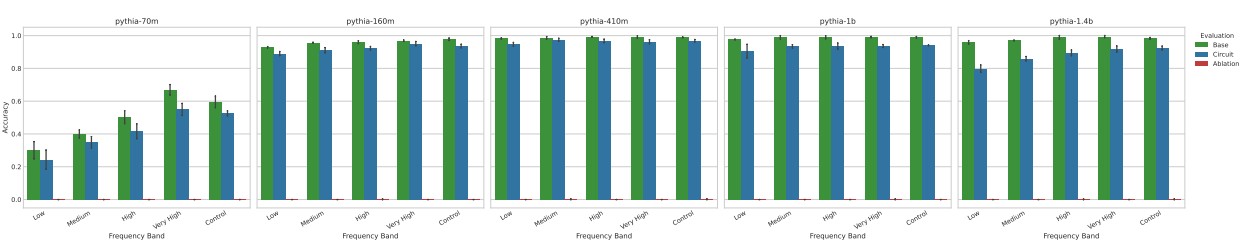

Figure 20: Base model, circuit, and ablation accuracy across models and bands. Ablation accuracy is at or near zero in all conditions.

Table 20: Cross-band generalization gap by model and training band. Same/Cross are accuracy (%); Gap $=$ Same $-$ Cross (positive $=$ better same-band).

| | Pythia-70m | | | Pythia-160m | | | Pythia-410m | | | Pythia-1b | | | Pythia-1.4b | | |
|---|---|---|---|---|---|---|---|---|---|---|---|---|---|---|---|
| Band | S | C | Gap | S | C | Gap | S | C | Gap | S | C | Gap | S | C | Gap |
| low | 24.3 | 39.6 | $-.15$ | 89.0 | 91.4 | $-.02$ | 94.7 | 96.7 | $-.02$ | 90.5 | 93.0 | $-.03$ | 79.9 | 91.3 | $-.11$ |
| medium | 34.8 | 40.1 | $-.05$ | 91.1 | 90.1 | $+.01$ | 97.5 | 95.6 | $+.02$ | 93.5 | 92.3 | $+.01$ | 85.9 | 89.1 | $-.03$ |
| high | 41.6 | 41.1 | $+.01$ | 92.3 | 89.0 | $+.03$ | 96.7 | 94.4 | $+.02$ | 93.6 | 91.7 | $+.02$ | 89.5 | 84.6 | $+.05$ |
| very_high | 55.0 | 36.0 | $+.19$ | 95.1 | 89.5 | $+.06$ | 96.3 | 93.1 | $+.03$ | 93.6 | 84.8 | $+.09$ | 91.9 | 77.5 | $+.14$ |
| control | 52.6 | 40.2 | $+.12$ | 93.6 | 89.9 | $+.04$ | 96.9 | 94.2 | $+.03$ | 94.1 | 89.2 | $+.05$ | 92.4 | 85.6 | $+.07$ |

## D.4 Circuit Completeness (Necessity)

Ablation accuracy is 0.0% in 69/75 circuits (Figure 20). The remaining six circuits have ablation accuracy of 0.44% (completeness 0.998): Pythia-160m high (draw 3), Pythia-410m medium and control (draw 2), Pythia-1b very_high (draw 2), and Pythia-1.4b high (draw 2) and control (draw 1).

Wilcoxon signed-rank confirms necessity for every model ($p_{\text{BH}} < 0.003$; $d = 3.4$–73.9). Completeness does not vary by band ($p > 0.40$).

## D.5 Cross-Band Generalization

Generalization gaps are small and often negative (Figure 21, Table 20); for the causal decomposition via targeted ablation, see Appendix H.2.

The same-vs-cross difference is significant only for Pythia-1b (Mann-Whitney $r = 0.41$, $p_{\text{BH}} = 0.035$); other models do not survive FDR correction. The gap does not differ across models ($p_{\text{BH}} = 0.92$).

## D.6 Asymmetric Transfer

LF→HF exceeds HF→LF by 4.7–20.4 pp (Figure 22, Table 21); the largest ratio is Pythia-70m (1.47), the largest absolute gap Pythia-1.4b (+20.4 pp), the smallest asymmetry Pythia-410m (1.05).

Table 21: Transfer asymmetry by model. Asymmetry = LF→HF − HF→LF accuracy; Ratio = LF→HF / HF→LF.

| Model | LF→HF (%) | HF→LF (%) | Asymmetry | Ratio |
|---|---|---|---|---|
| Pythia-70m | 43.1 | 29.3 | 0.138 | 1.47 |
| Pythia-160m | 91.8 | 86.1 | 0.057 | 1.07 |
| Pythia-410m | 96.5 | 91.7 | 0.047 | 1.05 |
| Pythia-1b | 92.5 | 83.4 | 0.091 | 1.11 |
| Pythia-1.4b | 92.1 | 71.7 | 0.204 | 1.28 |

Table 22: Circuit properties by model size (means over all bands and draws).

| Size (M) | Base (%) | Circ. (%) | ± | Edge % | Retention | KL |
|---|---|---|---|---|---|---|
| 70 | 49.4 | 41.7 | 12.3 | 30.9 | 0.841 | 0.251 |
| 160 | 95.9 | 92.2 | 2.4 | 12.3 | 0.962 | 0.292 |
| 410 | 98.8 | 96.4 | 1.3 | 4.4 | 0.975 | 0.320 |
| 1000 | 98.7 | 93.1 | 2.3 | 9.1 | 0.943 | 0.523 |
| 1400 | 97.9 | 87.9 | 5.0 | 2.8 | 0.897 | 0.559 |

Table 23: Variance decomposition of circuit accuracy.

| Factor | $\eta^2$ | Variance (%) |
|---|---|---|
| Model | 0.931 | 93.1 |
| Band | 0.026 | 2.6 |
| Model × Band | 0.036 | 3.6 |
| Draw | 0.001 | 0.1 |
| Residual | 0.007 | 0.7 |

The asymmetry is significant for all models ($p_{\mathrm{BH}} < 0.04$; $r = 0.65$–$0.88$) and increases with frequency distance ($\rho = 0.93$, $p_{\mathrm{BH}} = 0.036$).

### D.7 Model Scaling

Circuit accuracy peaks at Pythia-410m (96.4%), while edge fraction decreases from 30.9% to 2.8% and retention reaches 97.5% (Table 22). Pythia-1b/1.4b deviate: accuracy (93.1%, 87.9%) and retention (94.3%, 89.7%) fall below Pythia-410m, while KL continues to increase (0.52, 0.56).

Circuit accuracy varies strongly across models ($H = 47.0$, $\eta^2 = 0.79$, $p_{\mathrm{BH}} < 10^{-8}$). Edge fraction decreases with size ($\rho = -0.77$, $p < 10^{-11}$) and retention improves ($\rho = 0.41$, $p = 0.006$). The generalization gap does not scale with model size ($p_{\mathrm{BH}} = 0.92$).

### D.8 Variance Decomposition

Table 23 decomposes circuit accuracy variance including interaction terms; for the unified cross-perspective view, see Appendix G.4.

### D.9 Control Band Analysis

The control circuit does not differ from frequency-specific circuits in accuracy or size ($p_{\mathrm{BH}} > 0.17$; Figure 25). For Pythia-70m, it achieves the best cross-band transfer on 4/5 test bands (binomial test (Arbuthnot, 1710), $p_{\mathrm{BH}} = 0.033$); for larger models it is not significantly better. Jaccard similarity to band-specific circuits is nearly uniform (CV 1–3%).

### D.10 Per-Example Failure Analysis

Bimodality is rare: only 22/300 conditions are bimodal by Hartigan's dip test (2 same-band, 20 cross-band); most conditions show uniform leftward shift rather than bimodal splitting. Degradation is uniform across examples. Cross-band transfer correlates with same-band difficulty (Spearman $\rho = 0.37\text{--}0.77$): examples solved confidently on their own band also transfer well. Pythia-70m has 168 always-wrong examples; larger models have zero always-wrong examples (579–810 always-correct).

### D.11 Statistical Testing Summary

All hypothesis tests across the three analytical perspectives use BH-FDR correction at $\alpha = 0.05$.

**Functional (Phase 1).** Across 137 tests in 11 domains (Kruskal-Wallis, Mann-Whitney U, Wilcoxon signed-rank, Jonckheere-Terpstra, Spearman), 33 are significant (24%). The strongest effects are in random baselines (D11: 6/6), completeness (D8: 4/4), and asymmetric transfer (D4: 9/43); the weakest are control vs. frequency (D7: 0/8) and generalization gap (D3: 1/10).

**Structural (Phase 2).** Across 371 structural hypothesis tests (229 basic, 142 deep), the strongest effects correspond to containment asymmetry, draw stability, and model-level variance decomposition. Band effects are significant only for Jaccard gap and component-level Jaccard (Sections E.6–E.9), with small absolute magnitudes.

**Representational (Phase 3).** All 28 representational hypotheses (88 individual tests) were tested; verdicts are reported in the relevant subsections of Appendix F.1–F.9.



Figure 21: Cross-band transfer matrices for all five models (averaged over draws). Rows = training band, columns = test band. Matrices are largely uniform with strong low→high transfer.

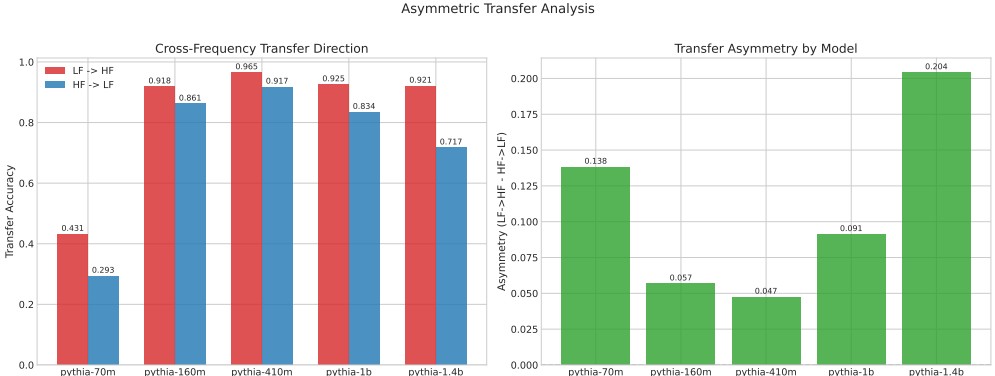

Figure 22: Asymmetric transfer: low-frequency (LF) circuits evaluated on high-frequency (HF) data consistently exceed the reverse direction.

Circuit Properties vs Model Size

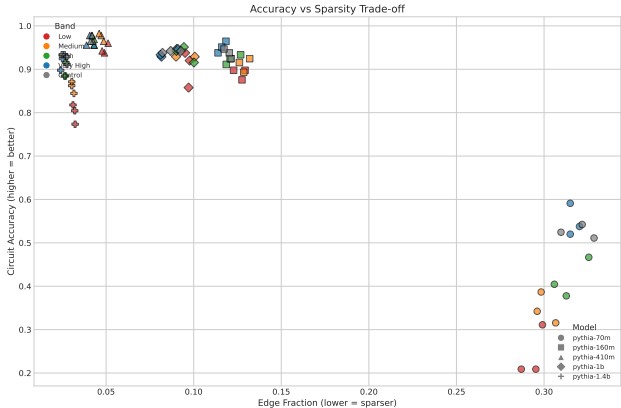

Figure 23: Circuit properties as a function of model size. Larger models yield more faithful (higher accuracy, higher retention) and more minimal (lower edge fraction) circuits, though KL divergence increases because larger models produce sharper output distributions. Pythia-1b deviates slightly from the trend.

Figure 24: Circuit accuracy versus circuit size (edge fraction) for all 75 circuits. Larger models achieve higher accuracy with smaller circuits, occupying the high-accuracy, low-edge-fraction region of the plot. Within each model, circuits from different frequency bands cluster together, reflecting the absence of a band effect on circuit size or accuracy.

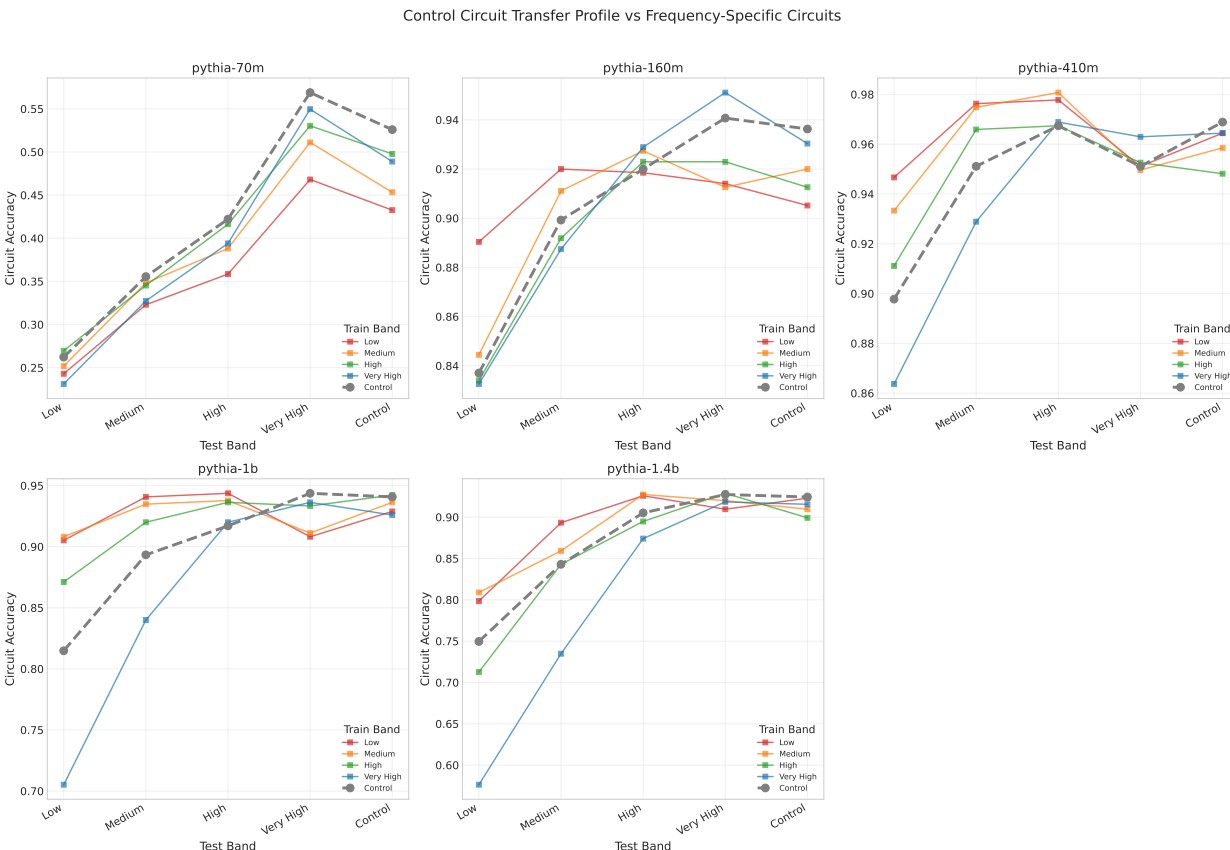

Figure 25: Control circuit matches the accuracy and transfer of frequency-specific circuits.

# E Additional Structural Results

This appendix provides structural analyses over all 75 circuits (5 models $\times$ 5 bands $\times$ 3 draws). Statistical tests use the non-parametric framework described in Section D.11; model effects dominate variance (Section E.15).

## E.1 Edge Statistics and Circuit Size

Edge fraction decreases from 30.9% (Pythia-70m) to 4.4% (Pythia-410m), with Pythia-1b intermediate at 9.1% (Table 24).

## E.2 Component Composition

Attention edges dominate all circuits (54.8–61.9%), followed by MLP (29.2–38.4%) and residual (6.9–12.9%; Table 25).

## E.3 Edge Categories

Skip edges dominate (62.3% in Pythia-70m to 88.7% in Pythia-410m); forward edges show the inverse pattern. Local edges <2%; input 0.4–4.5%; output 6.9–12.9%.

## E.4 Head Participation

Head participation decreases with model size: 85.0% (Pythia-70m), 68.2% (160m), 55.9% (410m), 62.0% (1b; Figure 26).

## E.5 Universal vs. Band-Specific Edges

Universal fraction decreases from 65.5% (Pythia-70m) to 15.4% (Pythia-1.4b; Figure 27, Table 26). Band-specific edges are predominantly attentional (75–93%); universal edges include larger MLP (37–50%) and residual (14–17%) contributions. The universal subgraph forms a single connected component covering 64–91% of nodes.

## E.6 Circuit Overlap (Jaccard Similarity)

Within-band Jaccard exceeds between-band in all models (gaps 0.013–0.032; Table 27).

## E.7 Power Analysis for Jaccard Gap Detection

Effect sizes are large for 70m–410m ($d \geq 1.0$, power $\geq 0.998$); Pythia-1b is underpowered ($d = 0.53$, power $= 0.69$; five draws needed). CLES values (0.65–0.85) confirm multiple extractions are necessary (Table 28).

## E.8 Power Analysis for Cross-Band Transfer Test

The Jaccard power analysis above addresses *structural* detection; we now bound the sensitivity of the *functional* cross-band transfer test. Each model contributes $n_{\text{same}}$=20 same-band observations and $n_{\text{cross}}$=80 cross-band observations (4 bands $\times$ 5 test conditions $\times$ 3 draws, partitioned by same vs. cross evaluation). A two-sample $t$-test at $\alpha$=0.05 achieves power $\geq 0.80$ to detect a same-band advantage of $\sim$0.02–0.03 accuracy points (depending on model-specific variance). The pipeline detects the residual same-band advantage of 0.016–0.029 accuracy points with $p < 0.03$ for all five models (Table 2), confirming adequate sensitivity at this effect size. The detected advantage constitutes only 2–4% of the total generic boost, and vanishes entirely under zero ablation (Cohen's $d \leq 0.18$; Appendix H.12). The pipeline's limitation is therefore not insufficient sensitivity but the absence of a positive control demonstrating that it can detect *genuine* specialization when present (Section 6.3). Note that this sensitivity bound applies to the functional transfer test;

Table 24: Circuit edge statistics by model (range over bands, draw-averaged).

| Model | $n_{\text{edges}}$ | Edge % |
|---|---|---|
| Pythia-70m | 389–424 | 30.9 |
| Pythia-160m | 1,331–1,478 | 12.3 |
| Pythia-410m | 3,296–3,976 | 4.4 |
| Pythia-1b | 847–967 | 9.1 |
| Pythia-1.4b | 2,024–2,567 | 2.8 |

Table 25: Component composition (%) by model (ranges over bands).

| Model | Attn | MLP | Resid |
|---|---|---|---|
| Pythia-70m | 54.8–58.4 | 29.2–32.2 | 12.3–12.9 |
| Pythia-160m | 54.9–57.9 | 33.6–35.9 | 8.5–9.3 |
| Pythia-410m | 53.6–57.5 | 35.6–38.4 | 6.9–8.1 |
| Pythia-1b | 59.6–61.9 | 30.9–32.7 | 6.9–8.1 |
| Pythia-1.4b | 61.0–62.3 | 30.6–31.5 | 6.8–7.6 |

Fraction of Active Attention Heads

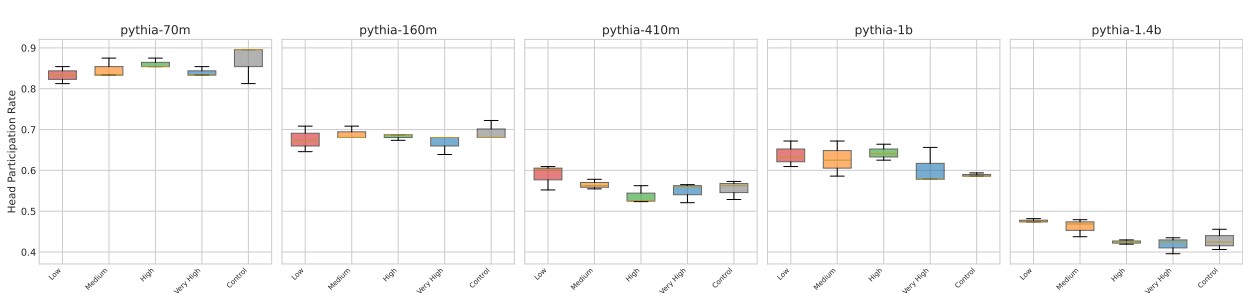

Figure 26: Head participation rate by model and band.

Universal vs Band-Specific Edges

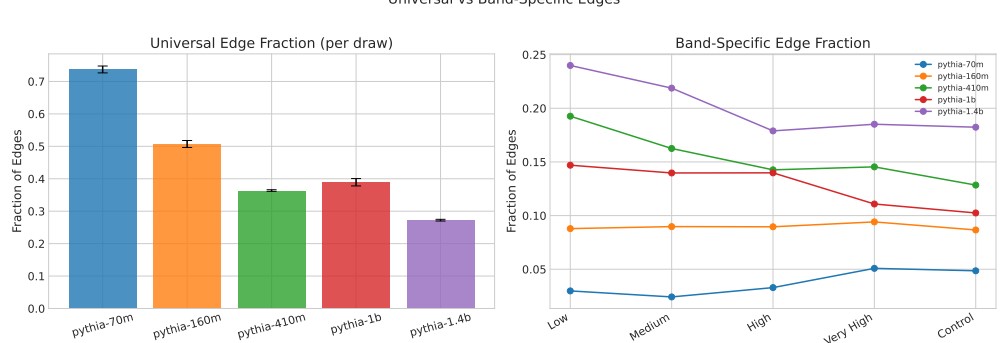

Figure 27: Universal vs. band-specific edge counts per model.

Table 26: Universal edge counts and fractions per model.

| Model | Universal | Mean edges | Univ. % |
|---|---|---|---|
| Pythia-70m | 268 | 409 | 65.5 |
| Pythia-160m | 531 | 1,407 | 37.8 |
| Pythia-410m | 872 | 3,581 | 24.3 |
| Pythia-1b | 223 | 915 | 24.4 |
| Pythia-1.4b | 350 | 2,277 | 15.4 |

the structural Jaccard gap analysis rests on fewer pairwise comparisons and has weaker power (Table 28 above).

Table 27: Within- vs. between-band Jaccard per model (95% bootstrap CIs).

| Model | $J_{\text{within}}$ [95% CI] | $J_{\text{between}}$ [95% CI] | Gap [95% CI] |
|---|---|---|---|
| Pythia-70m | 0.795 [.787, .803] | 0.763 [.756, .769] | 0.032 [.023, .042] |
| Pythia-160m | 0.589 [.582, .596] | 0.557 [.552, .562] | 0.032 [.024, .041] |
| Pythia-410m | 0.446 [.442, .452] | 0.430 [.427, .434] | 0.016 [.010, .022] |
| Pythia-1b | 0.478 [.470, .487] | 0.465 [.460, .470] | 0.013 [.003, .023] |
| Pythia-1.4b | 0.385 [.375, .395] | 0.366 [.361, .371] | 0.019 [.008, .030] |

Table 28: Power analysis for detecting the Jaccard gap ($n_{\min}$: draws for 80% power at $\alpha$=0.05; CLES: Common Language Effect Size).

| Model | Gap | Pooled SD | $d$ | CLES | $n_{\min}$ | Power (3) |
|---|---|---|---|---|---|---|
| Pythia-70m | 0.032 | 0.026 | 1.22 | 0.81 | 3 | 1.00 |
| Pythia-160m | 0.032 | 0.022 | 1.45 | 0.85 | 2 | 1.00 |
| Pythia-410m | 0.016 | 0.016 | 1.02 | 0.76 | 3 | 1.00 |
| Pythia-1b | 0.013 | 0.024 | 0.53 | 0.65 | 5 | 0.69 |
| Pythia-1.4b | 0.019 | 0.026 | 0.72 | 0.69 | 4 | 0.89 |

Table 29: Component-level Jaccard gaps. *Significant after BH-FDR.

| Model | Component | $J_{\text{within}}$ | $J_{\text{between}}$ | Gap | $d$ |
|---|---|---|---|---|---|
| | Attn* | 0.699 | 0.657 | 0.042 | 0.99 |
| Pythia-70m | MLP* | 0.925 | 0.910 | 0.014 | 0.74 |
| | Resid* | 0.985 | 0.971 | 0.013 | 0.77 |
| | Attn* | 0.478 | 0.440 | 0.038 | 1.30 |
| Pythia-160m | MLP* | 0.715 | 0.695 | 0.019 | 0.96 |
| | Resid* | 0.957 | 0.928 | 0.029 | 1.24 |
| | Attn* | 0.344 | 0.328 | 0.017 | 0.74 |
| Pythia-410m | MLP* | 0.552 | 0.540 | 0.012 | 0.98 |
| | Resid* | 0.869 | 0.844 | 0.025 | 1.03 |
| | Attn* | 0.421 | 0.407 | 0.014 | 0.51 |
| Pythia-1b | MLP | 0.536 | 0.526 | 0.010 | 0.38 |
| | Resid | 0.786 | 0.777 | 0.009 | 0.20 |
| | Attn* | 0.321 | 0.303 | 0.018 | 0.64 |
| Pythia-1.4b | MLP* | 0.456 | 0.437 | 0.019 | 0.66 |
| | Resid* | 0.744 | 0.714 | 0.029 | 0.76 |

### E.9 Component-Level Jaccard

The within>between gap holds per component, with attention showing the largest gap (0.014–0.042, significant in all models). MLP significant in three models; residual in two (Table 29).

### E.10 Band Affinity and Containment

Shared non-universal structure decreases with band distance, significantly for Pythia-70m ($\rho = -0.926$, $p_{\text{BH}} = 0.035$; Figure 28).

Low-frequency circuits contain more high-frequency edges than the reverse (significant for 160m, 410m, 1b; Pythia-70m inverted), paralleling functional transfer asymmetry (Appendix D.6).

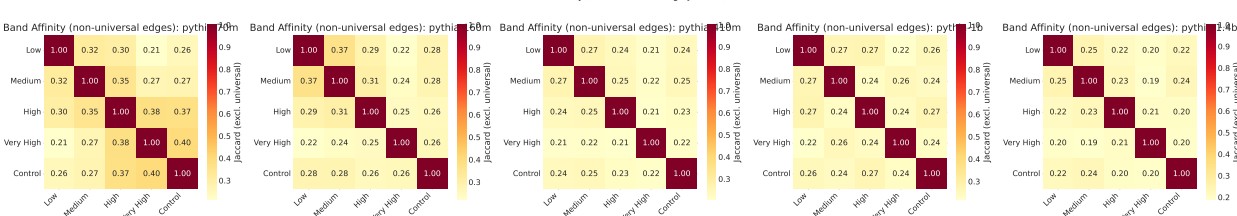

Figure 28: Band affinity heatmaps per model.

Table 30: Graph topology metrics per model (averaged over bands and draws).

| Model | Density | Diameter | Clustering | Path length |
|---|---|---|---|---|
| Pythia-70m | 0.150 | 2.6 | 0.714 | 1.71 |
| Pythia-160m | 0.079 | 3.5 | 0.703 | 1.89 |
| Pythia-410m | 0.041 | 3.9 | 0.685 | 2.02 |
| Pythia-1b | 0.065 | 3.9 | 0.564 | 2.12 |
| Pythia-1.4b | 0.036 | 4.1 | 0.543 | 2.22 |

Table 31: Draw stability by sharing level (% of edges appearing in 1, 2, or 3 draws).

| | Band-specific (1-band) | | | Universal (5-band) | | |
|---|---|---|---|---|---|---|
| Model | 1 draw | 2 draws | 3 draws | 1 draw | 2 draws | 3 draws |
| Pythia-70m | 87 | 12 | 1 | 8 | 10 | 82 |
| Pythia-160m | 87 | 12 | 1 | 12 | 17 | 71 |
| Pythia-410m | 89 | 10 | 1 | 17 | 22 | 60 |
| Pythia-1b | 93 | 6 | 1 | 16 | 25 | 59 |
| Pythia-1.4b | 90 | 10 | 1 | 19 | 24 | 57 |

### E.11 Layer Sensitivity

Neither layer sensitivity nor per-layer universal fraction correlates with depth (all $|\rho| \leq 0.38$, $p_{\mathrm{BH}} > 0.08$).

### E.12 Head Discrimination Entropy

Normalized entropy of head participation decreases with model size (0.907 to 0.684), indicating larger models develop more band-specialized heads.

### E.13 Graph Topology

Graph topology varies primarily by model (density $\eta^2 = 0.93$, clustering 0.68, diameter 0.64; all $p_{\mathrm{BH}} < 10^{-7}$), with no significant band effects (Table 30).

### E.14 Draw Stability

Universal edges are nearly 100% draw-stable; band-specific edges appear in 1–2 of 3 draws (Cramér's $V = 0.53$–0.58; rank-biserial $r = 0.91$–0.93; Table 31).

### E.15 Scaling and Variance Decomposition

Table 32 summarizes model-size trends.

Table 32: Structural metrics by model size (averaged over bands and draws).

| Size (M) | Edge % | Skip % | Head part. % | Attn % |
|---------:|-------:|-------:|-------------:|-------:|
| 70 | 30.9 | 62.2 | 85.0 | 57.2 |
| 160 | 12.3 | 80.2 | 68.2 | 56.2 |
| 410 | 4.4 | 88.7 | 55.9 | 55.6 |
| 1000 | 9.1 | 79.2 | 62.0 | 61.1 |
| 1400 | 2.8 | 84.4 | 44.2 | 61.7 |

Table 33: Variance decomposition ($\eta^2$) for structural metrics.

| Metric | Factor | $\eta^2$ | Var. (%) |
|--------|--------|---------:|---------:|
| | Model | 0.995 | 99.5 |
| | Band | 0.000 | 0.0 |
| Edge fraction | Model $\times$ Band | 0.004 | 0.4 |
| | Residual | 0.001 | 0.1 |
| | Model | 0.993 | 99.3 |
| | Band | 0.002 | 0.2 |
| Skip fraction | Model $\times$ Band | 0.002 | 0.2 |
| | Residual | 0.003 | 0.3 |
| | Model | 0.943 | 94.3 |
| | Band | 0.004 | 0.4 |
| Head part. rate | Model $\times$ Band | 0.016 | 1.6 |
| | Residual | 0.037 | 3.7 |

**Variance decomposition.** Table 33 decomposes structural metric variance; for the unified cross-perspective view, see Appendix G.4.

## F  Additional Representational Results

This appendix details representational analyses of all 75 circuits (5 models $\times$ 5 bands $\times$ 3 draws). Statistical tests use the non-parametric framework described in Section D.11.

### F.1  Embedding Space Geometry

Embedding-layer KNN purity (0.308–0.336) and probe accuracy (0.462–0.527) exceed chance (0.214 and 0.2; all $p = 0.0$; Table 34, Figure 29). Separation ratio scales monotonically (3.89 in Pythia-70m to 7.42 in Pythia-1b). CKA between band subspaces ranges 0.48–0.66, with no adjacency advantage. Intrinsic dimensionality decreases with scale (64–115% of ambient in Pythia-70m vs. 26–49% in Pythia-1b).

### F.2  Residual Stream Trajectories

Probe accuracy peaks at depth 0.50 (Pythia-70m) to 0.94 (Pythia-1b); peak separation ratio scales from 12.1 to 16.2 (Table 35). RSA yields $\rho = 0.77$–0.93. A single linear frequency direction explains up to $R^2 = 0.79$, peaking at early-to-mid layers. Circuit extraction preserves residual-stream geometry (CKA $> 0.97$).

### F.3  Logit Lens Convergence

Output distributions converge at fractional depths 0.68–0.88, with low-frequency bands converging later in all models (Table 36, Figure 31). Final $P$(correct) scales from 0.10–0.30 (Pythia-70m) to 0.58–0.71 (Pythia-1b; Figure 35a).

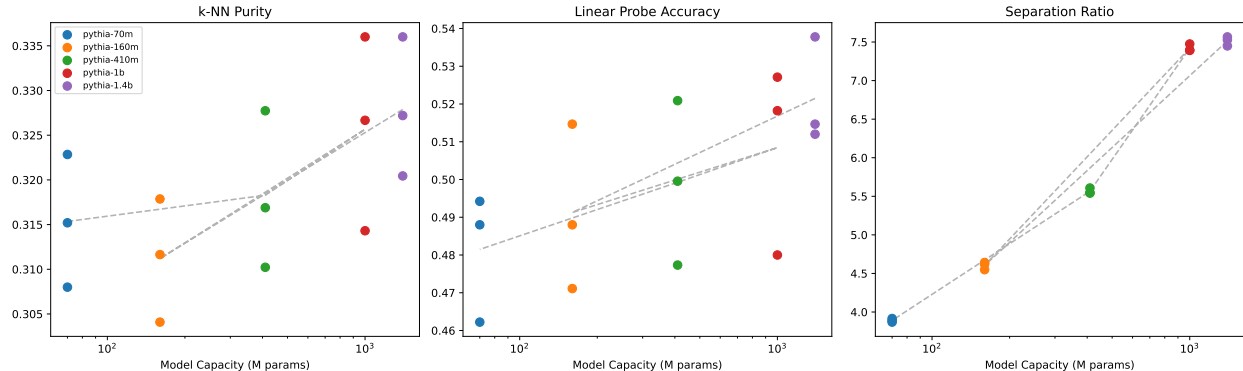

Figure 29: Embedding-layer representational metrics by model size.

Table 34: Embedding-layer metrics by model (draw-averaged).

| Model | KNN Pur. | Probe | Sep. R. |
|---|---|---|---|
| Pythia-70m | 0.315 | 0.481 | 3.89 |
| Pythia-160m | 0.311 | 0.491 | 4.60 |
| Pythia-410m | 0.318 | 0.499 | 5.57 |
| Pythia-1b | 0.326 | 0.509 | 7.42 |
| Pythia-1.4b | 0.328 | 0.521 | 7.51 |

Table 35: Residual-stream peak metrics (draw-averaged).

| Model | Peak Probe | Depth (frac) | Peak Sep. R. |
|---|---|---|---|
| Pythia-70m | 0.59 | 0.50 | 12.1 |
| Pythia-160m | 0.65 | 0.56 | 12.8 |
| Pythia-410m | 0.69 | 0.94 | 13.7 |
| Pythia-1b | 0.69 | 0.94 | 16.2 |
| Pythia-1.4b | 0.74 | 0.96 | 19.6 |

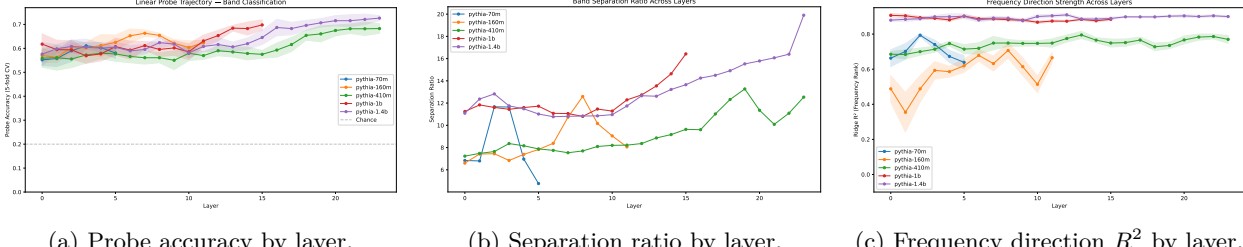

(a) Probe accuracy by layer.    (b) Separation ratio by layer.    (c) Frequency direction $R^2$ by layer.

Figure 30: Residual-stream representational trajectories across layers for all five Pythia models.

Component attribution varies by only 1.0–2.5 pp across bands ($\geq$160M); embedding norms do not predict convergence ($\rho = -0.40$ to $+0.40$, $p > 0.6$). The convergence gap scales from $+0.4$ layers (Pythia-70m) to $+7.0$ (Pythia-1.4b).

**Tuned lens validation.** To test whether the convergence delay is an artifact of logit-lens misalignment, we trained tuned lenses (Belrose et al., 2023) for all five Pythia models. Each tuned lens consists of per-layer affine translators ($h \mapsto h + Wh + b$, $W \in \mathbb{R}^{d \times d}$) trained to minimize $\mathrm{KL}(p_{\text{final}} \| p_{\text{lens}})$ over 500 steps of random-token sequences, then applied to the same LSC test data used in the standard logit-lens analysis. Because translators are trained on random tokens rather than LSC sequences, a distribution mismatch may affect calibration; however, the relative ordering across bands (low-frequency converging later) is preserved in all five models, and the gap ratio (tuned/logit) ranges from 0.73 to 1.25 (Table 37).

Circuit extraction preserves convergence timing (shift $< 0.05$ fractional layers).

## F.4 Attention Patterns

Induction scores are uniformly weak (0.031–0.054) and copy scores negligible ($\leq 0.12$; Tables 38–39, Figure 35b). Although LSC is an induction task, the low scores are expected: the standard induction-score

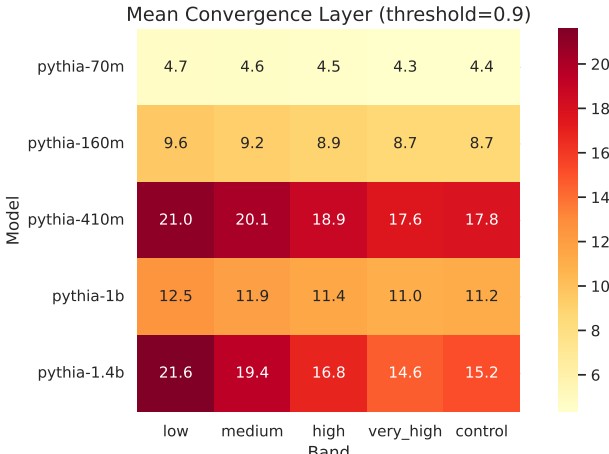

Figure 31: Logit lens convergence layer by model and frequency band.

Table 36: Logit lens convergence (ranges over bands, draw-averaged).

| Model | Conv. depth | $P$(correct) | Attn frac |
|---|---|---|---|
| Pythia-70m | .72–.79 | .10–.30 | .44–.54 |
| Pythia-160m | .71–.80 | .41–.56 | .39–.41 |
| Pythia-410m | .73–.88 | .51–.65 | .52–.54 |
| Pythia-1b | .68–.78 | .58–.71 | .59–.60 |
| Pythia-1.4b | .61–.90 | .55–.56 | .54–.57 |

Table 37: Convergence gap (low − very_high, frac. depth) under logit and tuned lens.

| Model | Logit gap | Tuned gap | Ratio |
|---|---|---|---|
| Pythia-70m | +.088 | +.073 | 0.83 |
| Pythia-160m | +.076 | +.055 | 0.73 |
| Pythia-410m | +.151 | +.152 | 1.01 |
| Pythia-1b | +.111 | +.138 | 1.25 |
| Pythia-1.4b | +.277 | +.209 | 0.76 |

Table 38: Attention metrics by model (ranges over bands).

| Model | Ind. score | BOS frac | Entropy |
|---|---|---|---|
| Pythia-70m | .050–.055 | .285–.299 | 1.81–1.93 |
| Pythia-160m | .031–.034 | .363–.384 | 1.65–1.67 |
| Pythia-410m | .032–.034 | .456–.476 | 2.05–2.14 |
| Pythia-1b | .036–.044 | .353–.360 | 2.53–2.61 |
| Pythia-1.4b | .027–.030 | .490–.495 | 1.94–1.96 |

Table 39: Head role distribution (%) by model.

| Model | BOS | Diff. | Prev | Ind. |
|---|---|---|---|---|
| Pythia-70m | 35.4 | 35.4 | 22.9 | 6.2 |
| Pythia-160m | 56.9 | 25.7 | 13.9 | 3.5 |
| Pythia-410m | 62.8 | 28.9 | 6.2 | 2.1 |
| Pythia-1b | 56.2 | 32.0 | 8.6 | 3.1 |
| Pythia-1.4b | 67.2 | 26.6 | 4.4 | 1.8 |

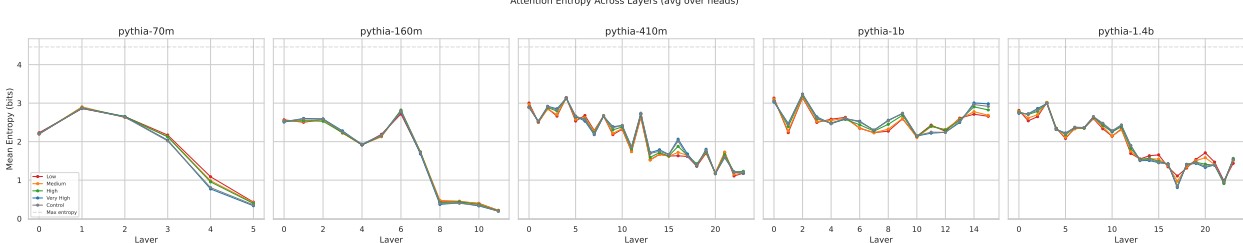

Figure 32: Attention entropy by layer and model.

metric (Olsson et al., 2022) measures attention to the token following a *bigram* repetition, whereas LSC uses a five-token prefix before the repeated segment begins. The longer prefix means that the two-token pattern-matching heuristic does not fire, even though the heads perform the same underlying copy computation. BOS-sink heads dominate (35–63%); induction heads are rare (2–6%); roles are draw-stable (77–87%) and band-invariant. Circuit extraction preserves attention entropy.

Table 40: MLP metrics by model. Sep. ratio ranges span draws.

| Model | Peak Probe | Sep. Ratio | MLP Frac | Selectivity % |
|---|---|---|---|---|
| Pythia-70m | 0.55 | 12.1–18.9 | 0.57 | 66.9 |
| Pythia-160m | 0.60 | 15.5–16.4 | 0.63 | 67.6 |
| Pythia-410m | 0.66 | 25.6–28.4 | 0.62 | 63.8 |
| Pythia-1b | 0.64 | 20.3–20.8 | 0.55 | 62.4 |
| Pythia-1.4b | 0.68 | 29.5–29.8 | 0.57 | 65.0 |

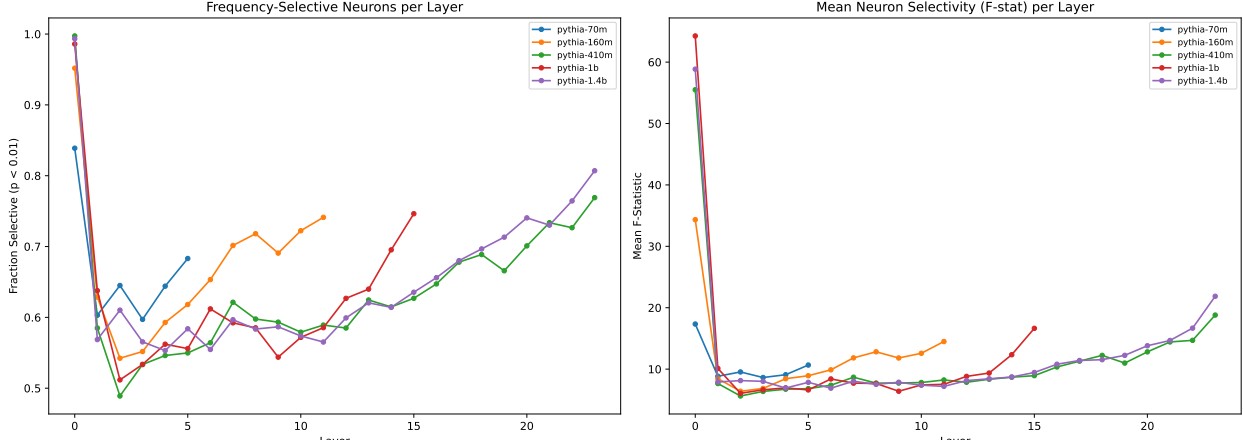

Figure 33: Fraction of frequency-selective MLP neurons by layer and model.

Table 41: Information-theoretic metrics by model (ranges over draws).

| Model | Peak MI | Peak Layer | Coding Eff. | Info Frac |
|---|---|---|---|---|
| Pythia-70m | 0.69–0.76 | 0 | 0.0013–0.0015 | 0.30–0.33 |
| Pythia-160m | 0.91 | 6 | 0.0012 | 0.39 |
| Pythia-410m | 0.97–1.01 | 1–3 | 0.0010 | 0.42–0.44 |
| Pythia-1b | 1.15–1.17 | 1–14 | 0.0006 | 0.50 |
| Pythia-1.4b | 1.11–1.16 | 1–22 | 0.0005–0.0006 | 0.48–0.50 |

## F.5   MLP Contributions

MLP outputs carry frequency information (peak probe 0.55–0.66; Table 40, Figure 35c), with neuron selectivity 62–68% and Gini sparsity 0.37–0.43 peaking at mid-depth. Circuit extraction preserves selectivity (Spearman $\rho = 0.71$–0.98).

## F.6   Information-Theoretic Measures

Peak MI scales from 0.73 (Pythia-70m) to 1.16 (Pythia-1b); coding efficiency decreases (0.0014 to 0.0006; Table 41, Figure 35d). Attention carries more band MI than MLP (0.70–1.07 vs. 0.46–0.74 nats); the embedding layer contributes the largest $\Delta$MI (0.69–1.08 nats). Circuit extraction preserves the MI trajectory (loss $< 10\%$).

## F.7   Cross-Band Transfer

Low-frequency circuits transfer to high-frequency bands more than the reverse (asymmetry up to 0.237 in Pythia-70m, $< 0.087$ for $\geq$160M). Early-layer residual CKA is the strongest predictor ($r = 0.931$); a combined

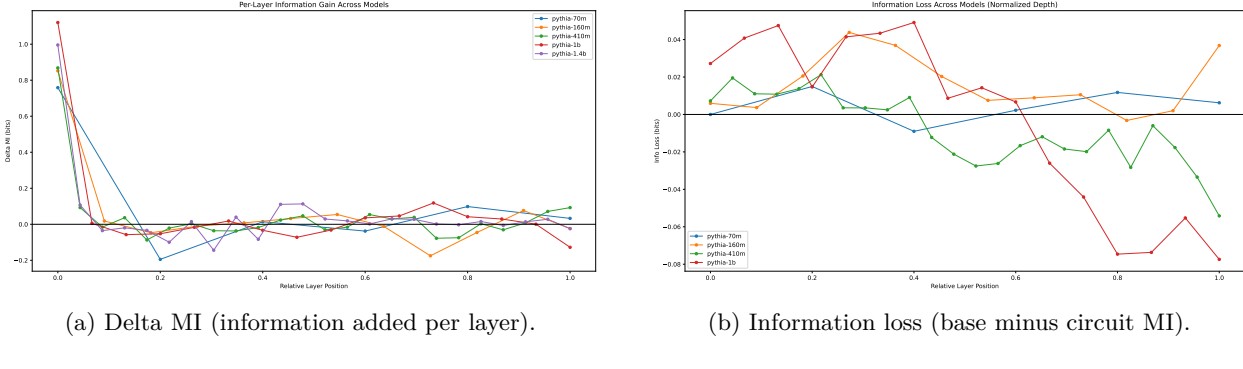

(a) Delta MI (information added per layer).

(b) Information loss (base minus circuit MI).

Figure 34: Information-theoretic layer-wise analyses across all five Pythia models.

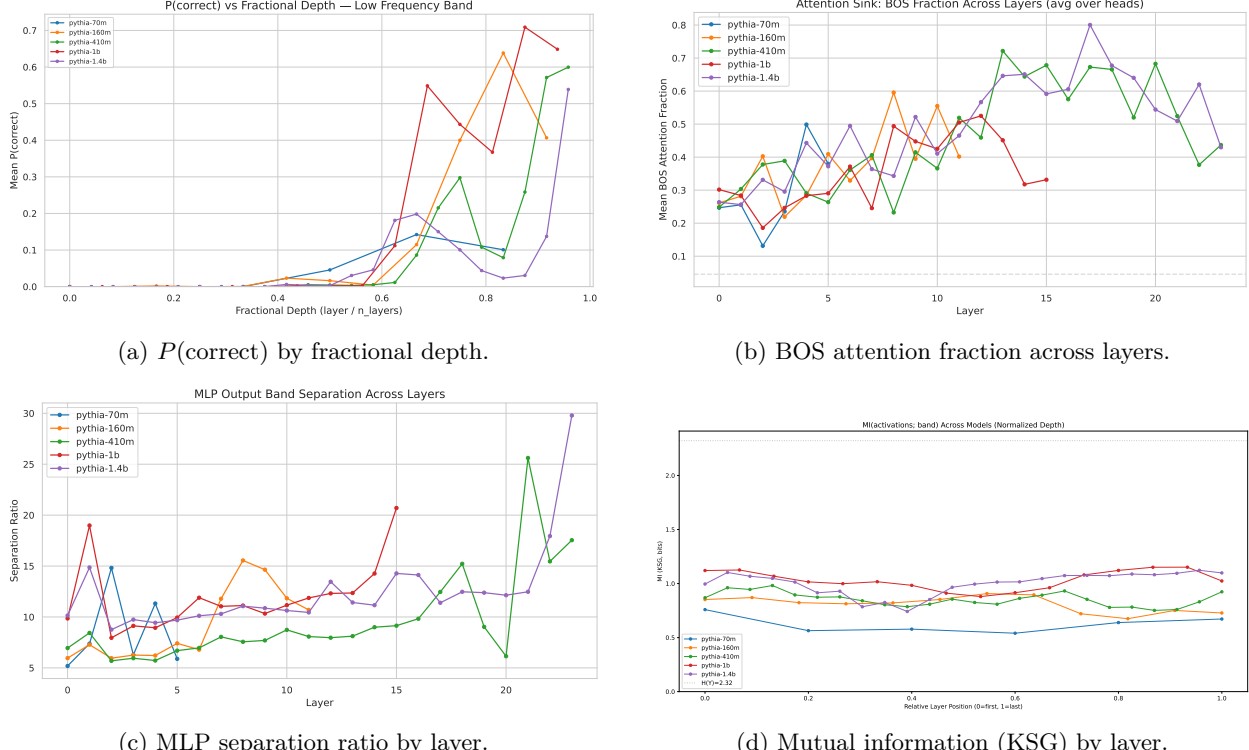

(a) $P$(correct) by fractional depth.

(b) BOS attention fraction across layers.

(c) MLP separation ratio by layer.

(d) Mutual information (KSG) by layer.

Figure 35: Layer-wise representational metrics across all five Pythia models. (a) Logit lens convergence by fractional depth; (b) BOS attention fraction; (c) MLP band-separation ratio; (d) mutual information between activations and band identity.

regression achieves $R^2 = 0.990$ (CV 0.978), with representational features adding $\Delta R^2 = 0.130$ ($F = 35.9$, $p < 10^{-8}$; Table 42).

## F.8 Scaling Summary

Table 43 consolidates model-size trends.

## F.9 Causal Interventions: Interchange Patching and Boundless DAS

The preceding correlational analyses show that structurally distinct circuits produce equivalent outputs (Sections 5.1.1–5.1.3) and that the universal core preserves base-model representational geometry (Section 5.3.1).

Table 42: Representational–transfer correlations ($n = 40$).

| Metric | $r$ | $p_{\text{FDR}}$ | Sig. |
|---|---|---|---|
| Resid CKA (early) | 0.931 | $< 10^{-16}$ | ✓ |
| Jaccard | −0.919 | $< 10^{-15}$ | ✓ |
| Resid CKA (mid) | 0.828 | $< 10^{-10}$ | ✓ |
| Embedding CKA | 0.790 | $< 10^{-8}$ | ✓ |
| Resid CKA (final) | 0.712 | $< 10^{-6}$ | ✓ |
| Convergence diff | 0.413 | 0.009 | ✓ |
| Freq distance | −0.084 | 0.70 | |

Table 43: Representational metrics by model size (draw-averaged).

| Size (M) | Emb Sep | Peak Probe | Peak Sep | Conv. Depth | Peak MI | Coding Eff. |
|---|---|---|---|---|---|---|
| 70 | 3.89 | 0.59 | 12.1 | 0.75 | 0.73 | 0.0014 |
| 160 | 4.60 | 0.65 | 12.8 | 0.75 | 0.91 | 0.0012 |
| 410 | 5.57 | 0.69 | 13.7 | 0.80 | 0.99 | 0.0010 |
| 1000 | 7.42 | 0.69 | 16.2 | 0.73 | 1.16 | 0.0006 |
| 1400 | 7.51 | 0.74 | 19.6 | 0.73 | 1.13 | 0.0006 |

This subsection provides causal evidence via interchange patching and Boundless DAS (Wu et al., 2023), applied at the residual stream of the full (unpruned) model.

### F.9.1 Interchange Intervention (Activation Patching)

**Method.** For each band pair (base, source), we construct 100 interchange pairs from the two bands. We cache source activations at the residual stream (`hook_resid_post`) and run the base input with source activations patched in at a single position. Interchange Intervention Accuracy (IIA) measures the fraction of examples where the patched model's prediction matches the source target. High cross-band IIA indicates a shared representational format; low cross-band IIA would indicate band-specific encoding.

**Layer sweep and full IIA matrix.** Sweeping across all layers at the prediction position (position 21, with BOS) for four representative band pairs, IIA rises sharply in upper layers and peaks at model-specific layers: L5 (70m), L11 (160m), L23 (410m), L13 (1b), L21 (1.4b) (Figure 36). At each model's peak layer, we evaluate all 25 band pairs (Figure 37).

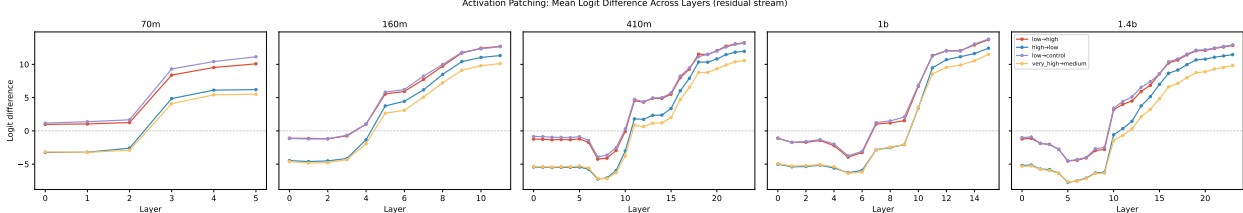

Figure 36: Interchange intervention logit difference by layer (residual stream, prediction position). The patched model's logit for the source target rises sharply in upper layers, with all band pairs converging to similar peak values.

For models ≥160M, cross-band IIA is uniformly high (Table 7): mean IIA ranges from 0.96 (Pythia-160m) to 0.99 (Pythia-1b). The same-band vs. cross-band difference is at most 0.004, with no systematic advantage for same-band pairs. Pythia-70m achieves lower IIA overall (mean 0.47), consistent with its limited task capacity.



Figure 37: Full 5×5 IIA matrices at peak layer. Rows: base band; columns: source band. Near-uniform values indicate that the model uses a shared representational format across all bands.

**Position and component analysis.** At the peak layer, sweeping patching position across all token positions shows that IIA is zero everywhere except the final prediction position (position 21), confirming that band-distinguishing information is causally localized to the prediction position. Decomposing the intervention into residual stream, attention, and MLP components shows that attention heads contribute more than MLP at the peak layer, consistent with the attention-mediated copy mechanism.

**Draw robustness.** Repeating the peak-layer evaluation across all three draws yields virtually identical IIA (standard deviation 0.000–0.008 for all model–band-pair combinations).

**Positive control: layer-sweep sensitivity.** To verify that interchange patching has sufficient power to detect causal structure when it exists, we perform a layer sweep using within-band example pairs with distinct target tokens. Across all five models, source IIA is exactly 0 at layers 0–2 and rises in a sigmoid pattern to 0.92–0.98 at the peak layer (Figure 38). The transition is sharp: for models ≥160M, IIA jumps from ≤0.06 to ≥0.73 within two to three layers. Comparing within-band and cross-band patching at the peak layer, the IIA gap is ≤0.010 for all models ≥160M ($\Delta = +0.006, +0.010, -0.003, +0.004$ for 160M, 410M, 1B, 1.4B respectively). The framework reliably detects when a causal property is present (target identity) and when it is absent (early layers); it finds no evidence that band identity is a causal property. We note that this control validates the sensitivity of interchange patching specifically; it does not validate the full circuit-comparison pipeline, which would require a setting where ground-truth specialization is known *a priori*.

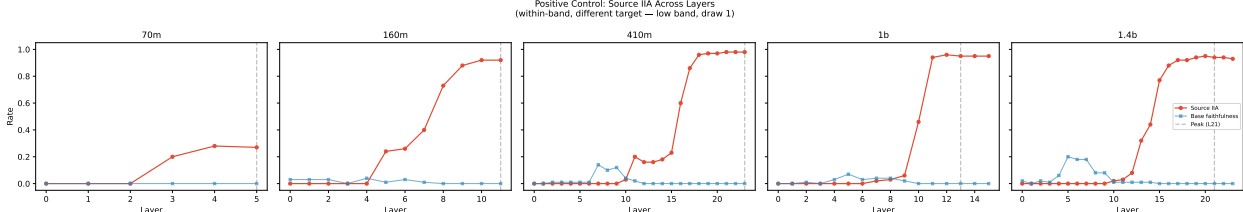

Figure 38: Positive control: source IIA across layers for within-band different-target interchange patching. IIA is 0 at early layers and rises to 0.92–0.98 at peak layers, demonstrating full dynamic range. Dashed lines mark the peak layer for each model.

### F.9.2 Boundless DAS: Minimal Subspace for Band Identity

**Method.** To test whether band-distinguishing information concentrates in a low-dimensional subspace, we apply Boundless DAS (Wu et al., 2023). Boundless DAS jointly learns a rotation matrix $R \in \mathbb{R}^{k \times d}$ and a boundary parameter $b \in [0, 1]$ determining the effective subspace dimension, with $L_1$ regularization encouraging the smallest sufficient subspace. We train at each model's peak layer on the low→high and high→low band pairs (500 steps, $\lambda_{\text{boundary}} = 0.05$, warmup 100 steps, max candidate dimension $\min(128, d_{\text{model}})$).

**Results.** Boundless DAS achieves IIA = 1.0 for all models and both directions. The effective dimension decreases with model scale (Table 8; Figure 39): from $\sim$31 dimensions in Pythia-70m (6.1% of $d_{\text{model}}$) to $\sim$12 in Pythia-1.4b (0.6%). The absolute number of band-encoding dimensions drops from 31 to 12 (2.6$\times$), while the ambient dimension grows from 512 to 2048 (4$\times$), compounding to a $\sim$10$\times$ decrease in the fractional subspace, suggesting genuine compression rather than merely proportional scaling. Both directions (low$\rightarrow$high vs. high$\rightarrow$low) agree to within 1–2 dimensions for every model, indicating a direction-symmetric subspace.

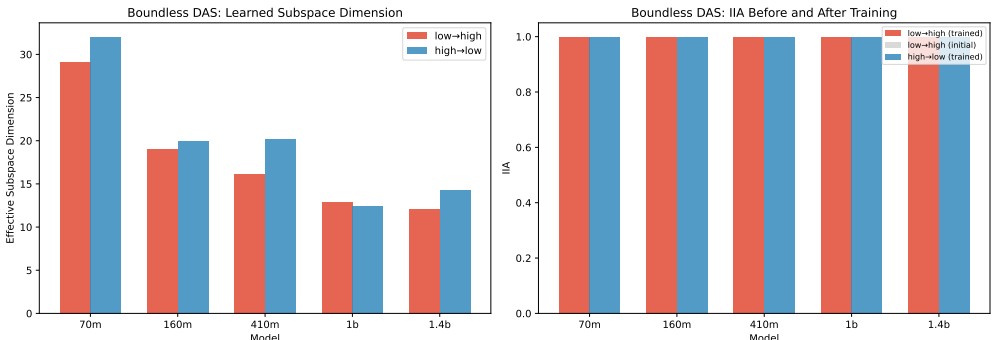

Figure 39: Boundless DAS effective subspace dimension across models. Left: absolute dimension; right: fraction of $d_{\text{model}}$. Both directions converge to nearly identical dimensions, confirming direction symmetry.

### F.9.3 Interpretation

The interchange patching and Boundless DAS results jointly provide causal confirmation of phantom specialization: **(1)** Band representations are interchangeable: replacing one band's residual-stream activations with another's redirects prediction with near-perfect accuracy ($\geq$0.95 for models $\geq$160M). **(2)** Band identity occupies a tiny subspace of 12–31 dimensions (0.6–6.1% of $d_{\text{model}}$), shrinking with scale. **(3)** The subspace is universal: both patching directions yield symmetric effective dimensions and the full 5×5 IIA matrix shows no pair-specific asymmetry. These causal findings complement the correlational evidence: not only do structurally distinct circuits produce equivalent outputs, but the underlying representations are causally interchangeable.

## G Integration Analysis

Key results are summarized in Section 5.2.2 of the main text.

### G.1 Structure–Function Correlations

Edge fraction correlates negatively with accuracy ($\rho = -0.876$, $p < 10^{-19}$) and retention ($\rho = -0.712$), whereas skip fraction correlates positively with accuracy ($\rho = 0.761$; Table 44). Within-model correlations are generally non-significant except for Pythia-70m and 160m edge fraction ($p < 0.003$).

### G.2 Structure–Representation Mantel Tests

Per-model Mantel tests (Mantel, 1967) comparing structural (Jaccard) and representational (CKA) distance matrices yield non-significant correlations ($r = -0.41$ to $+0.39$, all $p > 0.18$).

### G.3 Similarity Triangle

Figure 40 and Table 45 summarize pairwise Spearman correlations across the three triangle edges. The structure–function edge yields the strongest overall correlations, though 52/80 pairs exhibit Simpson's paradox. The structure–representation edge shows the highest reversal rate (68/80), with within-model correla-

Table 44: Structure–function Spearman correlations (all $p < 10^{-6}$).

| Structural | Functional | $\rho$ | $p$ |
|---|---|---|---|
| Edge fraction | Accuracy | $-0.876$ | $< 10^{-19}$ |
| Edge fraction | Retention | $-0.712$ | $< 10^{-9}$ |
| Edge fraction | KL div. | $-0.595$ | $< 10^{-6}$ |
| Skip fraction | Accuracy | $+0.761$ | $< 10^{-12}$ |
| Skip fraction | Retention | $+0.783$ | $< 10^{-13}$ |
| Head part. rate | Accuracy | $-0.842$ | $< 10^{-16}$ |
| Head part. rate | Retention | $-0.700$ | $< 10^{-9}$ |
| Total edges | Accuracy | $+0.754$ | $< 10^{-11}$ |
| Total edges | Retention | $+0.817$ | $< 10^{-15}$ |

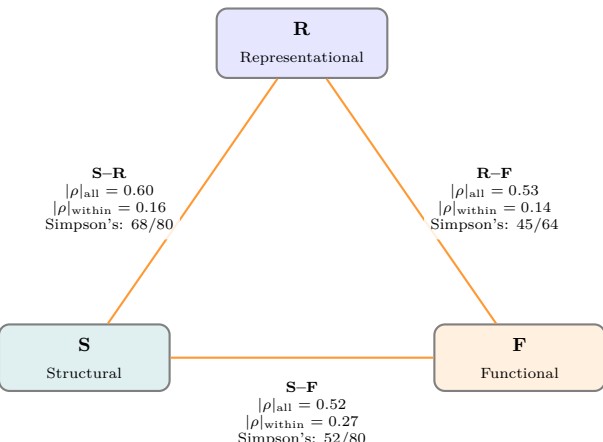

Figure 40: Quantitative similarity triangle. Mean absolute Spearman correlations computed over all metric pairs per edge; "within" averages per-model correlations. All three edges are confounded by model scale (orange), with high Simpson's paradox rates indicating sign reversals within individual models. Compare with the conceptual triangle in Figure 4.

Table 45: Strongest pairwise correlations per triangle edge. Within-model $\rho$ is the mean across five models; "Simp." indicates Simpson's paradox (sign reversal within models).

| Edge | Metric pair | Overall $\rho$ | Within $\rho$ | Simp. |
|---|---|---|---|---|
| S–F | Edge frac. $\leftrightarrow$ Size frac. | $+1.000$ | $+1.000$ | No |
| S–F | Edges/head $\leftrightarrow$ $n_{\text{edges}}$ | $+0.982$ | $+0.850$ | No |
| S–F | $n_{\text{univ.}} \leftrightarrow$ Total edges | $+0.975$ | n/a | No |
| S–R | Univ. frac. $\leftrightarrow$ Peak probe layer | $-0.958$ | $-0.122$ | Yes |
| S–R | Edge frac. $\leftrightarrow$ Peak probe layer | $-0.953$ | $-0.050$ | Yes |
| S–R | Univ. frac. $\leftrightarrow$ Peak MI layer | $-0.953$ | $-0.092$ | Yes |
| R–F | Peak probe layer $\leftrightarrow$ Size frac. | $-0.953$ | $-0.050$ | Yes |
| R–F | Peak MI layer $\leftrightarrow$ Size frac. | $-0.945$ | $+0.013$ | Yes |
| R–F | Convergence layer $\leftrightarrow$ Size frac. | $-0.915$ | $+0.368$ | Yes |

tions near zero even for strong overall pairs (e.g., universal fraction vs. peak probe layer: $\rho_{\text{overall}} = -0.958$, $\rho_{\text{within}} = -0.12$). The representation–function edge follows a similar pattern (45/64 reversals; Figure 9 in main text).

Table 46: Variance decomposition ($\eta^2$) for representative metrics from each perspective. All metrics are model-dominated except completeness.

| Metric | Persp. | $\eta^2_{\text{model}}$ | $\eta^2_{\text{cond.}}$ | $\eta^2_{\text{repl.}}$ | Resid. |
|---|---|---|---|---|---|
| Edge fraction | S | 0.995 | <0.001 | <0.001 | 0.005 |
| Skip fraction | S | 0.994 | 0.002 | <0.001 | 0.004 |
| Univ. fraction | S | 0.996 | 0.000 | 0.003 | 0.001 |
| Base accuracy | F | 0.907 | 0.029 | <0.001 | 0.064 |
| Circuit accuracy | F | 0.937 | 0.023 | 0.001 | 0.039 |
| Retention ratio | F | 0.796 | 0.020 | 0.007 | 0.177 |
| Completeness | F | 0.022 | 0.022 | 0.044 | 0.911 |
| Convergence layer | R | 0.982 | 0.011 | <0.001 | 0.007 |
| Peak MI probe | R | 0.936 | <0.001 | 0.042 | 0.021 |
| Peak efficiency | R | 0.991 | <0.001 | 0.003 | 0.006 |

## G.4    Unified Variance Decomposition

Extending the structural variance decomposition (Appendix E.15) to all 28 unified metrics, model identity remains dominant ($\eta^2_{\text{model}} \geq 0.69$ for all metrics except completeness; Table 46). Frequency band accounts for less than 4% of variance in every metric. Completeness is the sole exception: replication-driven ($\eta^2_{\text{repl.}} = 0.044$) with large residual (0.911).

## G.5    Cross-Perspective Concordance and Scaling

Hierarchical clustering of the $26 \times 26$ Spearman correlation matrix recovers three blocks aligning with structural, functional, and representational perspective labels (Figure 41). Within-perspective correlations are consistently stronger than between-perspective ones; the strongest cross-perspective pair is skip fraction with retention ratio ($\rho = 0.783$).

Cross-perspective agreement on condition difficulty is weak (mean Kendall's $W = 0.202$); no single frequency band consistently ranks highest or lowest across all three perspectives. Per-circuit perspective disagreement averages 0.443 (range 0.237–0.723), with Pythia-1b showing the highest mean disagreement (0.515) and Pythia-410m the lowest (0.375).

All three perspectives exhibit consistent scaling trends with model capacity (Figure 42). Functional metrics improve with scale (base accuracy $\rho = 0.864$, circuit accuracy $\rho = 0.639$), representational metrics deepen (peak MI probe $\rho = 0.928$, final probability correct $\rho = 0.940$), and structural metrics compress (edge fraction $\rho = -0.775$, universal fraction $\rho = -0.777$, peak efficiency $\rho = -0.972$). Only MLP fraction ($\rho = 0.171$, $p = 0.246$) and completeness ($\rho = -0.115$, $p = 0.434$) fail to reach significance.

## G.6    Incremental Prediction

In pooled leave-one-out cross-validated regressions ($N = 48$), skip fraction is the best single structural predictor of retention ratio ($R^2 = 0.712$), while final probability correct is the best representational predictor ($R^2 = 0.533$; Table 47). Adding representational metrics to a full structural model yields negative incremental $R^2$ ($\Delta R^2 = -0.154$). Per-model regressions ($N = 12$ each) produce strongly negative $R^2$ values.

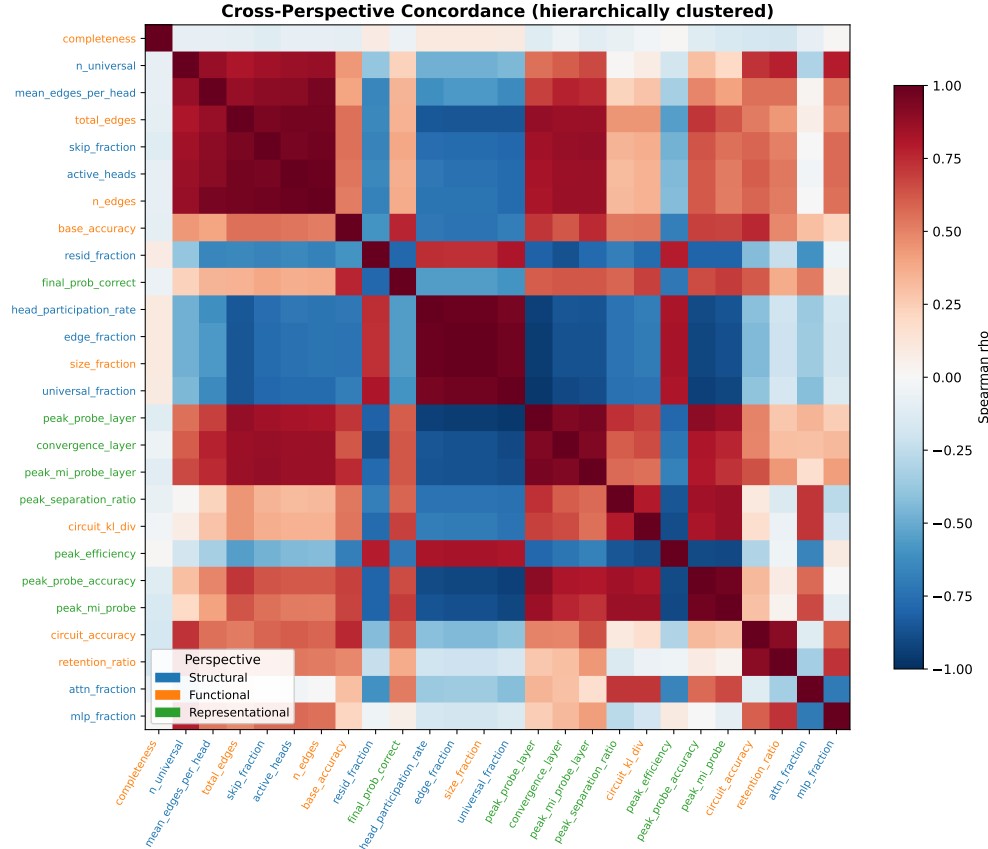

Figure 41: Hierarchically clustered correlation matrix of 26 unified metrics. Block structure aligns with perspective labels (blue: structural, orange: functional, green: representational).

Table 47: Incremental prediction of retention ratio (LOO-CV $R^2$). Per-model regressions overfit due to small $N$.

| Scope | $N$ | Best structural | $R^2_S$ | Best repr. | $R^2_R$ |
|---|---|---|---|---|---|
| Overall | 60 | Skip fraction | +0.412 | Final prob. correct | +0.364 |
| Pythia-1.4b | 12 | Edge fraction | +0.452 | Convergence layer | +0.650 |
| Pythia-160m | 12 | Skip fraction | +0.283 | Convergence layer | +0.070 |
| Pythia-1b | 12 | MLP fraction | −0.042 | Peak efficiency | −0.122 |
| Pythia-410m | 12 | MLP fraction | −0.200 | Peak probe layer | −0.288 |
| Pythia-70m | 12 | Resid. fraction | −0.093 | Peak sep. ratio | −0.258 |

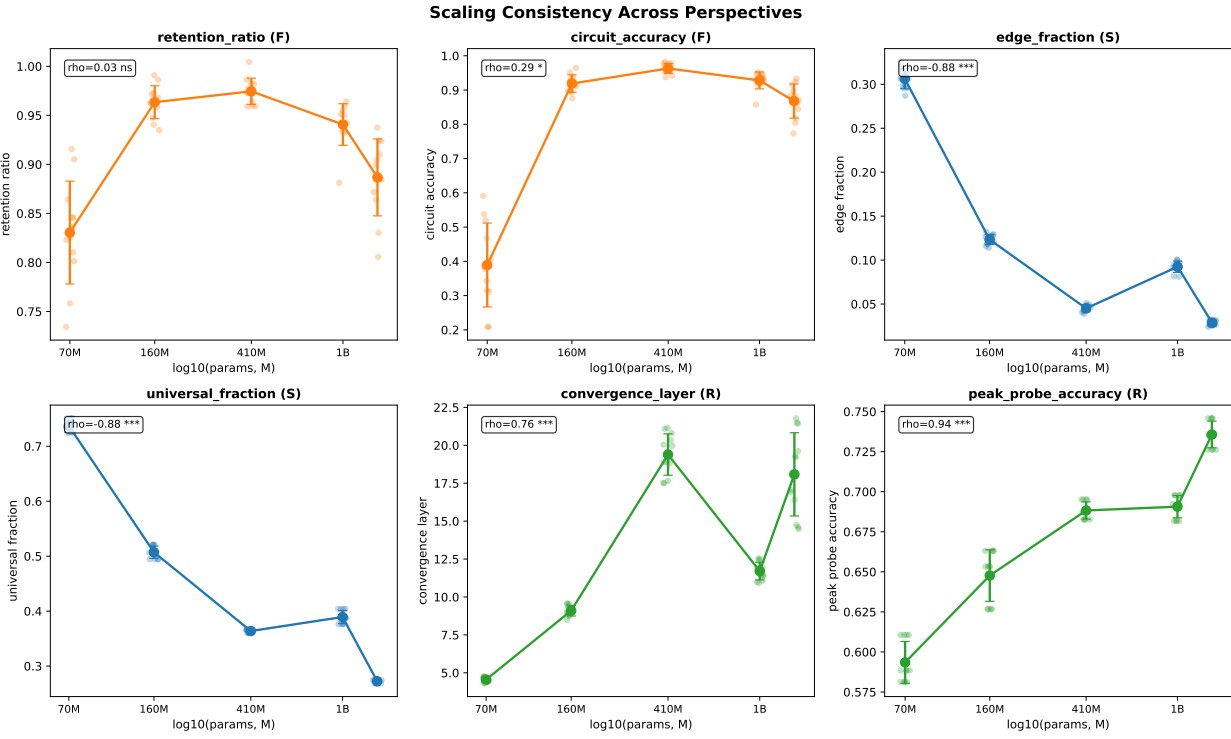

Figure 42: Scaling consistency across perspectives. Six representative metrics (two per perspective) plotted against $\log_{10}$ model capacity. Points are individual circuits; lines show model-level means ± one standard deviation.

# H    Targeted Ablation Studies

This appendix reports ten targeted ablation analyses testing the universal core hypothesis across all 75 circuits (5 models $\times$ 5 bands $\times$ 3 draws).

## H.1    Universal Core Sufficiency

The universal core recovers 71.4% (Pythia-70m) to 13.7% (Pythia-1.4b) of full circuit accuracy (Figure 43, Table 48). Band-specific edges in isolation achieve 0% accuracy across all models.

The universal core outperforms size-matched random edge sets by 868–4,970$\times$ (Wilcoxon $p < 3.3 \times 10^{-4}$).

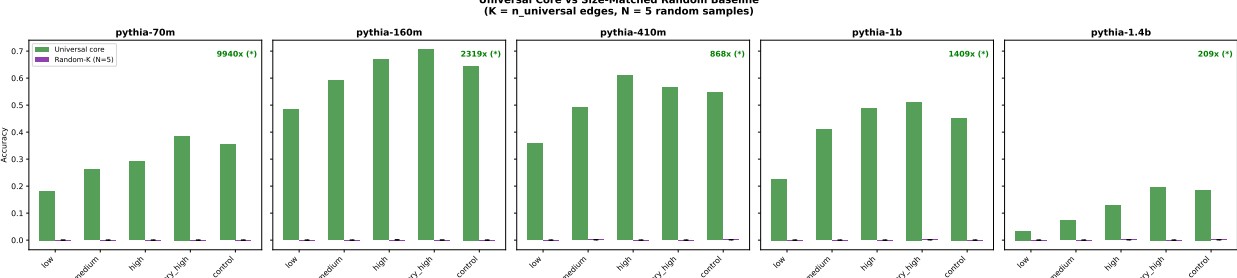

Figure 43: Universal core accuracy across models and frequency bands. Retention decreases with model scale.

Table 48: Universal core sufficiency by model (means over bands and draws).

| Model | Full Acc | Univ. Acc | Retention | Band-Spec. Acc | Random Acc |
|-------|----------|-----------|-----------|----------------|------------|
| Pythia-70m | 0.417 | 0.295 | 0.714 | 0.000 | <0.001 |
| Pythia-160m | 0.922 | 0.618 | 0.669 | 0.000 | <0.001 |
| Pythia-410m | 0.964 | 0.514 | 0.533 | 0.001 | <0.001 |
| Pythia-1b | 0.931 | 0.417 | 0.447 | 0.000 | <0.001 |
| Pythia-1.4b | 0.879 | 0.124 | 0.137 | 0.001 | <0.001 |

Figure 44: Universal core accuracy vs. size-matched random edge sets. The universal core advantage is 868–4,970× across models.

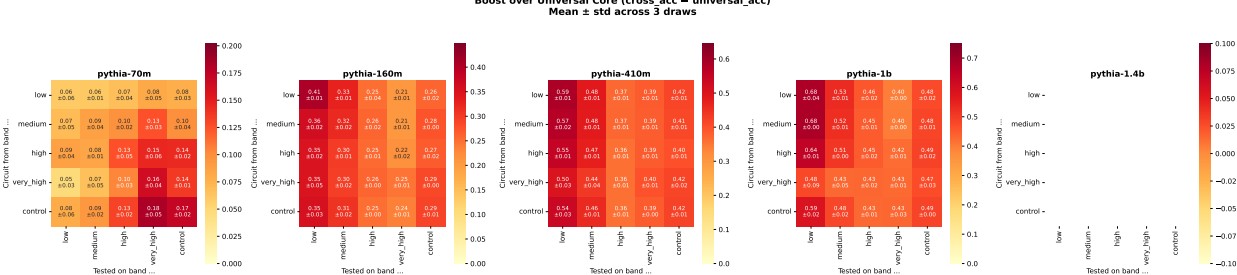

Figure 45: Cross-band accuracy boost from band-specific edges. Each panel shows one model; rows are source bands (whose specific edges are added to the universal core), columns are test bands. Near-uniform rows indicate that band-specific edges help all bands similarly, not just their source band.

Table 49: Cross-band transfer of band-specific edges; $d$: Cohen's $d$ for the same-vs-cross difference; $p$: permutation test. 95% bootstrap percentile CIs ($N$=10,000) in brackets.

| Model | Same Boost | Cross Boost | Random | Transfer Eff. [95% CI] | $d$ | $p$ |
|-------|-----------|-------------|--------|------------------------|-----|-----|
| Pythia-70m | $0.122 \pm 0.055$ | $0.099 \pm 0.035$ | 0.014 | 0.814 [.74, .91] | 0.49 | 0.004 |
| Pythia-160m | $0.304 \pm 0.059$ | $0.282 \pm 0.048$ | 0.028 | 0.926 [.89, .96] | 0.42 | <0.001 |
| Pythia-410m | $0.450 \pm 0.080$ | $0.434 \pm 0.064$ | 0.024 | 0.964 [.95, .98] | 0.22 | 0.006 |
| Pythia-1b | $0.513 \pm 0.091$ | $0.485 \pm 0.064$ | 0.027 | 0.944 [.92, .97] | 0.36 | 0.001 |
| Pythia-1.4b | $0.756 \pm 0.027$ | $0.733 \pm 0.037$ | 0.010 | 0.970 [.94, .99] | 0.71 | 0.028 |

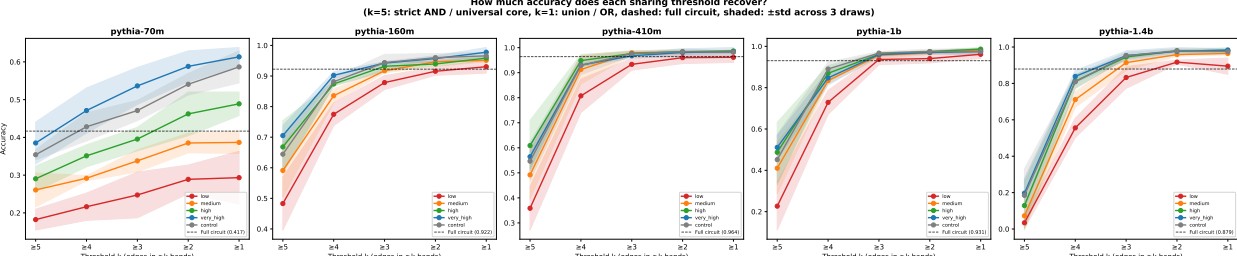

Figure 46: Accuracy as a function of sharing threshold $k$. Lower $k$ includes more edges and recovers more accuracy.

## H.2 Cross-Band Transfer

Band-specific edges transfer across bands at 81–97% efficiency (Table 49; Figure 45). Same-band boost significantly exceeds cross-band boost ($p < 0.03$, $d = 0.22$–$0.71$), but the absolute advantage is small (0.016–0.029 accuracy points), indicating a fixed structural offset rather than growing specialization. Random edges contribute only 1–3% boost.

## H.3 Sharing Threshold Sweep

The critical threshold where accuracy reaches 95% of the full circuit is predominantly $k = 3$ (Pythia-410m requires $k = 3$–4; Table 50).

## H.4 Majority-Shared Core: Per-Band Edge Composition

The $k{\geq}3$ edge sets are largely band-agnostic (Table 51). Pairwise Jaccard similarity exceeds the full-circuit value for every model, with the largest increase at Pythia-1.4b (+79%), where low universal-core retention

Table 50: Critical sharing threshold $k$ per model and band. Target accuracy is the value at the critical $k$; full circuit accuracy in parentheses.

| Band | Pythia-70m | | Pythia-160m | | Pythia-410m | | Pythia-1b | | Pythia-1.4b | |
|---|---|---|---|---|---|---|---|---|---|---|
| | $k$ | Acc (Full) | $k$ | Acc (Full) | $k$ | Acc (Full) | $k$ | Acc (Full) | $k$ | Acc (Full) |
| low | 3 | .231 (.243) | 3 | .846 (.890) | 3 | .899 (.947) | 3 | .860 (.905) | 3 | .759 (.799) |
| medium | 3 | .331 (.348) | 3 | .866 (.911) | 3 | .926 (.975) | 3 | .888 (.935) | 3 | .816 (.859) |
| high | 3 | .395 (.416) | 3 | .877 (.923) | 4 | .919 (.967) | 3 | .889 (.936) | 3 | .850 (.895) |
| very_high | 3 | .522 (.550) | 3 | .904 (.951) | 4 | .915 (.963) | 3 | .889 (.936) | 3 | .873 (.919) |
| control | 2 | .500 (.526) | 3 | .889 (.936) | 4 | .920 (.969) | 3 | .894 (.941) | 3 | .878 (.924) |

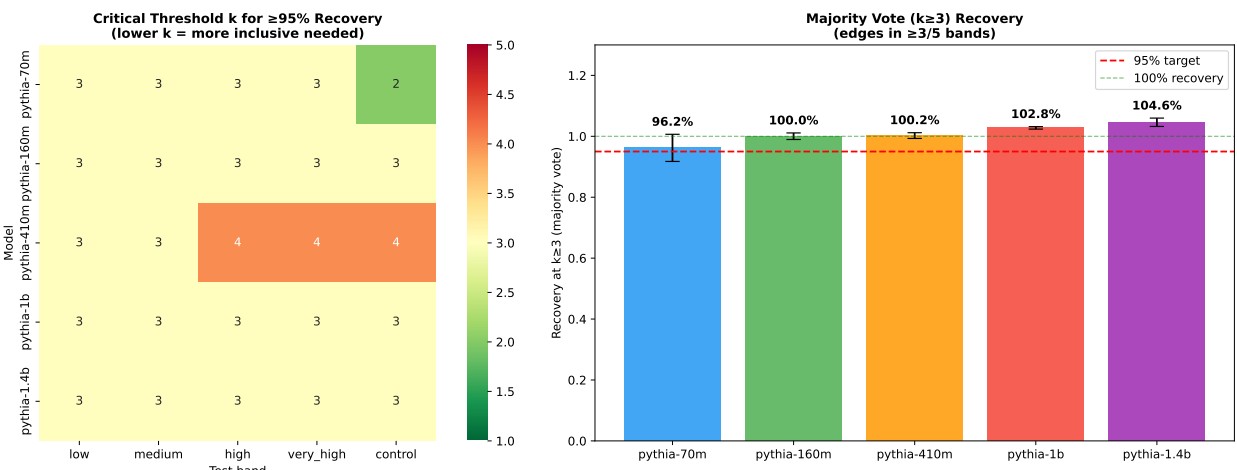

Figure 47: Critical sharing threshold per model and band. Most configurations reach 95% recovery at $k = 3$.

(13.7%) makes this most relevant. Per-band participation ratios range from 0.84 to 1.18, indicating near-uniform band membership.

Table 51: Per-band $k{\geq}3$ Jaccard similarity compared with full-circuit between-band Jaccard (means over three draws).

| Model | Full-circuit between-band | $k{\geq}3$ per-band | Relative increase |
|---|---|---|---|
| Pythia-70m | 0.763 | 0.876 | +14.8% |
| Pythia-160m | 0.557 | 0.770 | +38.2% |
| Pythia-410m | 0.430 | 0.700 | +62.8% |
| Pythia-1b | 0.465 | 0.704 | +51.4% |
| Pythia-1.4b | 0.366 | 0.656 | +79.2% |

### H.5 Draw Variability

Cross-draw transfer ratios range from 0.990 to 1.017, with all 95% CIs spanning 1.0 (Table 58).

Structurally, 68–73% of Pythia-70m edges appear in all three draws, decreasing to 29–31% for Pythia-410m; despite this structural variation, functional transfer remains near-perfect.

Figure 48: Edge sharing distribution by model. Smaller models have a higher fraction of universal (5-band) edges.

Table 52: Role enrichment in universal heads (Fisher's exact test (Fisher, 1922)). $^*$ denotes significance at $\alpha = 0.05$.

| Model | Role | OR | $p$ | Sig |
|---|---|---|---|---|
| Pythia-70m | previous_token$^*$ | $\infty$ | 0.044 | Yes |
| | bos_sink$^*$ | 0.10 | 0.002 | Yes |
| | induction | $\infty$ | 0.563 | No |
| Pythia-160m | previous_token$^*$ | 5.49 | 0.006 | Yes |
| | bos_sink$^*$ | 0.21 | <0.001 | Yes |
| | induction | $\infty$ | 0.066 | No |
| Pythia-410m | previous_token$^*$ | 3.73 | 0.004 | Yes |
| | bos_sink$^*$ | 0.38 | <0.001 | Yes |
| | induction$^*$ | 10.32 | 0.010 | Yes |
| Pythia-1b | previous_token$^*$ | 14.38 | 0.002 | Yes |
| | bos_sink$^*$ | 0.28 | <0.001 | Yes |
| | induction$^*$ | $\infty$ | 0.040 | Yes |
| Pythia-1.4b | induction$^*$ | $\infty$ | <0.001 | Yes |
| | previous_token$^*$ | 11.64 | <0.001 | Yes |
| | bos_sink$^*$ | 0.17 | <0.001 | Yes |

Table 53: Mechanism profiling summary by model.

| Model | Univ. Heads % | Dom. Role | Dom. Edge | Univ. Stable | Band-Spec. Stable |
|---|---|---|---|---|---|
| Pythia-70m | 75.0 | diffuse | attn→mlp | 97.9% | 1.2% |
| Pythia-160m | 55.6 | bos_sink | attn→mlp | 95.9% | 1.7% |
| Pythia-410m | 41.4 | bos_sink | attn→mlp | 94.8% | 1.6% |
| Pythia-1b | 45.3 | bos_sink | attn→attn | 93.9% | 1.1% |
| Pythia-1.4b | 31.0 | diffuse | attn→attn | 93.2% | 0.8% |

## H.6 Edge and Mechanism Profiling

Universal edges are 94–98% stable across draws; band-specific edges are only 1–2% stable (Figure 48).

The `previous_token` role is enriched in universal heads (OR = 3.7–14.4, $p < 0.05$; Table 52), while `bos_sink` is depleted (OR = 0.10–0.38, $p < 0.002$). Universal head fraction decreases from 75% (Pythia-70m) to 31% (Pythia-1.4b), and the dominant role shifts from `diffuse` to `bos_sink` at larger scales (Table 53, Figure 49).

## H.7 BOS-Sink Ablation

Removing all BOS-sink edges and comparing against equal-count random removal (Table 55) shows BOS-sink edges are distinguishable from noise: $-67$ pp vs. $-89$ pp (Pythia-160m), $-19$ pp vs. $-93$ pp (Pythia-410m).

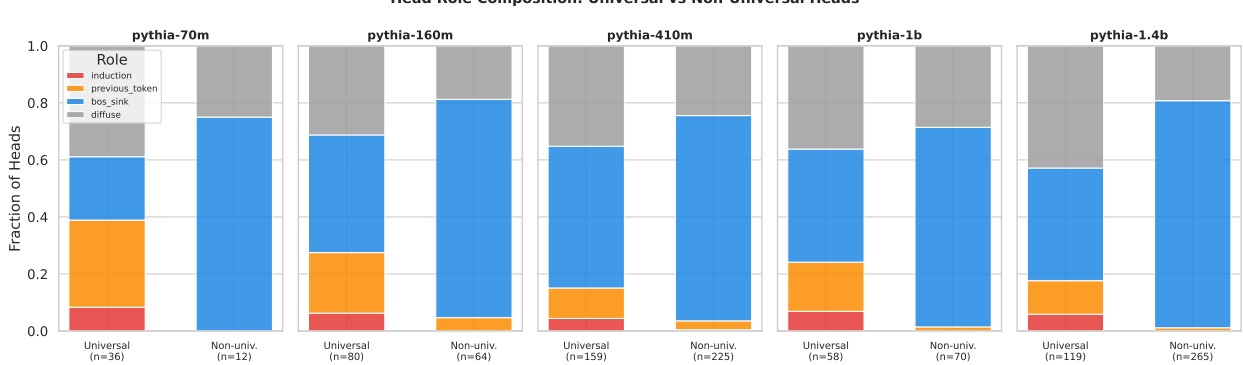

Figure 49: Attention head role distribution by universality class.

Table 54: Per-layer mechanism profiles binned into early, middle, and late layers. Universal edge fraction and universal head fraction are means within each bin; dominant role is the mode.

| Model | Position | Layers | Univ. Edge % | Univ. Head % | Dom. Role |
|---|---|---|---|---|---|
| Pythia-70m | early | 0–1 | 0.79 | 0.75 | bos_sink |
| | middle | 2–4 | 0.48 | 0.79 | diffuse |
| | late | 5 | 0.68 | 0.62 | bos_sink |
| Pythia-160m | early | 0–3 | 0.30 | 0.46 | bos_sink |
| | middle | 4–8 | 0.26 | 0.68 | bos_sink |
| | late | 9–11 | 0.40 | 0.47 | bos_sink |
| Pythia-410m | early | 0–7 | 0.13 | 0.21 | bos_sink |
| | middle | 8–16 | 0.15 | 0.65 | bos_sink |
| | late | 17–23 | 0.22 | 0.35 | bos_sink |
| Pythia-1b | early | 0–5 | 0.24 | 0.40 | bos_sink |
| | middle | 6–10 | 0.22 | 0.53 | bos_sink |
| | late | 11–15 | 0.20 | 0.45 | bos_sink |
| Pythia-1.4b | early | 0–7 | 0.15 | 0.30 | bos_sink |
| | middle | 8–16 | 0.11 | 0.43 | bos_sink |
| | late | 17–23 | 0.09 | 0.16 | bos_sink |

Table 55: BOS-sink ablation: accuracy after removing all BOS-sink edges vs. removing the same number of random edges (means across five bands).

| Model | Full circuit | −BOS-sink | $\Delta_{\mathrm{BOS}}$ | −Random | $\Delta_{\mathrm{rand}}$ |
|---|---|---|---|---|---|
| Pythia-160m | 89.9% | 22.4% | −67.5 pp | 0.9% | −89.0 pp |
| Pythia-410m | 96.0% | 76.7% | −19.3 pp | 3.4% | −92.6 pp |

The contribution is substantial but *generic*: accuracy drops are similar across all five bands (59–77 pp for 160m, 13–25 pp for 410m), with no band preference. The model-size dependence is consistent with smaller models having fewer redundant pathways. Memory constraints prevented evaluation on Pythia-1b and 1.4b; the decreasing trend suggests the effect continues to diminish with scale. BOS-sink edges likely suppress competing attention signals (Xiao et al., 2024) rather than propagating target information; their contribution is band-agnostic.

Table 56: Scaling synthesis of targeted ablation metrics. 95% bootstrap CIs in brackets.

| Model | Params | Univ. % | Retention | Transf. Eff. [CI] | Draw Transf. [CI] | Crit. $k$ |
|---|---|---|---|---|---|---|
| Pythia-70m | 70M | 73.7 | 0.714 | 0.814 [.74, .91] | 0.990 [.94, 1.05] | 2.8 |
| Pythia-160m | 160M | 50.7 | 0.669 | 0.926 [.89, .96] | 0.998 [.99, 1.01] | 3.0 |
| Pythia-410m | 410M | 36.4 | 0.533 | 0.964 [.95, .98] | 0.999 [.99, 1.01] | 3.6 |
| Pythia-1b | 1B | 38.9 | 0.447 | 0.944 [.92, .97] | 1.004 [.99, 1.01] | 3.0 |
| Pythia-1.4b | 1.4B | 27.2 | 0.137 | 0.970 [.94, .99] | 1.017 [1.00, 1.03] | 3.0 |

## H.8 Scaling Synthesis

Table 56 consolidates the preceding metrics across model scale.

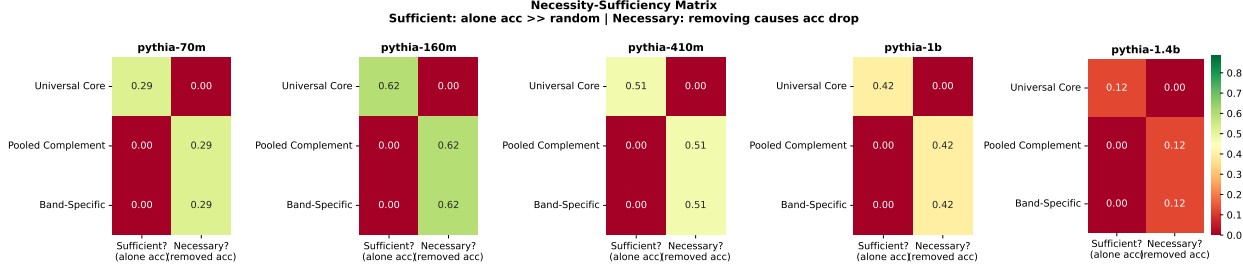

Figure 50: Necessity and sufficiency matrix. The universal core is both sufficient and necessary (complement accuracy is zero).

Table 57: Necessity and sufficiency by model (means over bands).

| Model | Univ. Acc | Full Acc | Comp. Acc | Univ. Ret. | Comp. Ret. |
|---|---|---|---|---|---|
| Pythia-70m | 0.295 | 0.417 | 0.000 | 0.714 | 0.000 |
| Pythia-160m | 0.618 | 0.922 | 0.000 | 0.669 | 0.000 |
| Pythia-410m | 0.514 | 0.964 | 0.001 | 0.533 | 0.001 |
| Pythia-1b | 0.417 | 0.931 | 0.000 | 0.447 | 0.000 |
| Pythia-1.4b | 0.124 | 0.879 | 0.001 | 0.137 | 0.001 |

## H.9 Complement Ablation

The complement circuit (all edges not in the universal core) achieves 0% accuracy across all models and bands, establishing necessity (Figure 50, Table 57). Random edge sets of equal size also yield 0%.

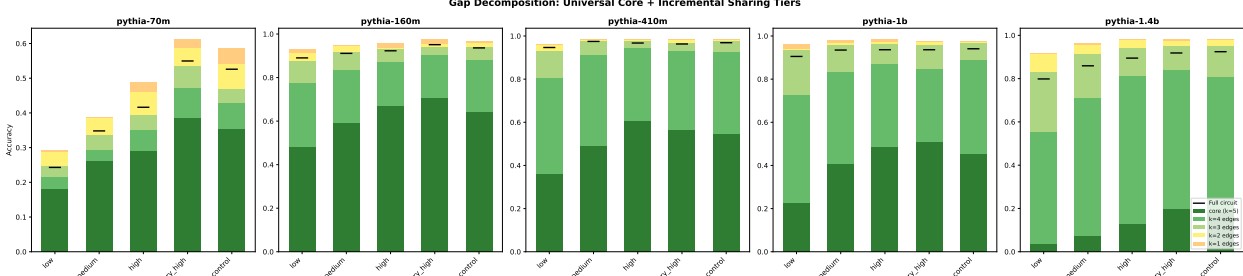

Figure 51: Logit lens trajectory overlay: universal core vs. full circuit. The universal core follows the same trajectory shape at reduced amplitude.

Table 58: Draw variability: same- vs. cross-draw accuracy (band means; 95% CIs).

| Model | Same | Cross | Ratio [CI] |
|---|---|---|---|
| Pythia-70m | .417 | .413 | .990 [.94, 1.05] |
| Pythia-160m | .922 | .921 | .998 [.99, 1.01] |
| Pythia-410m | .964 | .963 | .999 [.99, 1.01] |
| Pythia-1b | .931 | .934 | 1.00 [.99, 1.01] |
| Pythia-1.4b | .879 | .894 | 1.02 [1.0, 1.03] |

Table 59: Algorithm preservation (band and draw means). Conv. Shift: convergence delay of universal core vs. full circuit (layers); Amp. Ratio: amplitude ratio (core / full circuit).

| Model | Pearson $r$ | Conv. Shift | Amp. Ratio |
|---|---|---|---|
| Pythia-70m | 0.766 | 0.50 | 3.10 |
| Pythia-160m | 0.919 | 1.98 | 2.01 |
| Pythia-410m | 0.949 | 2.19 | 1.82 |
| Pythia-1b | 0.987 | 1.04 | 1.55 |
| Pythia-1.4b | 0.954 | 3.34 | 1.88 |

Figure 52: Gap decomposition across sharing tiers. Including $k \geq 3$ edges closes nearly all of the universal-to-full-circuit gap.

## H.10 Algorithm Preservation

Logit lens trajectories of the universal core correlate highly with the full circuit: $r = 0.77$–$0.99$ (Figure 51, Table 59).

The universal core converges 0.5–3.3 layers later at 1.6–3.1× amplitude ratio, consistent with an attenuated version of the same computation.

## H.11 Performance Gap Decomposition

The gap between universal core and full circuit grows with scale (0.12–0.76 accuracy points; Table 60). Including $k{\geq}3$ edges closes 99–105% of this gap; band-specific edges contribute negligibly. Source-level evaluation inflates apparent accuracy by 0.22–0.57 points relative to edge-level ablation.

Table 60: Performance gap decomposition by model (means over bands). Gap closed at $k \geq 3$ exceeds 99% for all models.

| Model | Full Acc | Univ. Acc | Gap | Closed ($k{\geq}3$) | Source Acc | Edge Acc |
|---|---|---|---|---|---|---|
| Pythia-70m | 0.417 | 0.295 | 0.122 | 99.3% | 0.513 | 0.295 |
| Pythia-160m | 0.922 | 0.618 | 0.304 | 100.9% | 0.954 | 0.618 |
| Pythia-410m | 0.964 | 0.514 | 0.450 | 100.9% | 0.989 | 0.514 |
| Pythia-1b | 0.931 | 0.417 | 0.513 | 105.3% | 0.991 | 0.417 |
| Pythia-1.4b | 0.879 | 0.124 | 0.756 | 105.5% | 0.974 | 0.124 |

Table 61: Cross-band transfer under resample vs. zero ablation. [†]Transfer efficiency undefined when same-band boost is negative or near zero.

| Model | Ablation | Same Boost | Cross Boost | Transfer Eff. | Cohen's $d$ |
|---|---|---|---|---|---|
| Pythia-70m | resample | +0.122 | +0.099 | 81.4% | 0.49 |
| | zero | −0.115 | −0.127 | —[†] | 0.18 |
| Pythia-160m | resample | +0.304 | +0.282 | 92.6% | 0.42 |
| | zero | +0.127 | +0.127 | 99.8% | 0.00 |
| Pythia-410m | resample | +0.450 | +0.434 | 96.4% | 0.22 |
| | zero | +0.001 | +0.002 | —[†] | −0.16 |
| Pythia-1b | resample | +0.513 | +0.485 | 94.4% | 0.36 |
| | zero | +0.001 | +0.001 | —[†] | −0.14 |
| Pythia-1.4b | resample | +0.756 | +0.733 | 97.0% | 0.71 |
| | zero | +0.004 | +0.003 | 73.1% | 0.15 |

### H.12 Zero-Ablation Robustness

Under zero ablation (Conmy et al., 2023), the same-band advantage vanishes: Cohen's $d \leq 0.18$ for all models, compared to 0.22–0.71 under resample ablation (Table 61, Figure 53). Absolute accuracy drops substantially: models $\geq$410M collapse to near-chance ($<0.4\%$) because zeroing 71–95% of edges pushes activations off the data manifold (Conmy et al., 2023; Yu et al., 2025). Despite this collapse, the same-vs-cross boost difference is $\leq$0.012 accuracy points ($\leq$0.001 for three models), confirming that the residual same-band advantage under resample ablation reflects ablated context rather than genuine specialization. Source-level zero ablation preserves 35–90% accuracy while edge-level collapses to $\leq$0.5% for $\geq$410M models, further confirming source-level inflation.

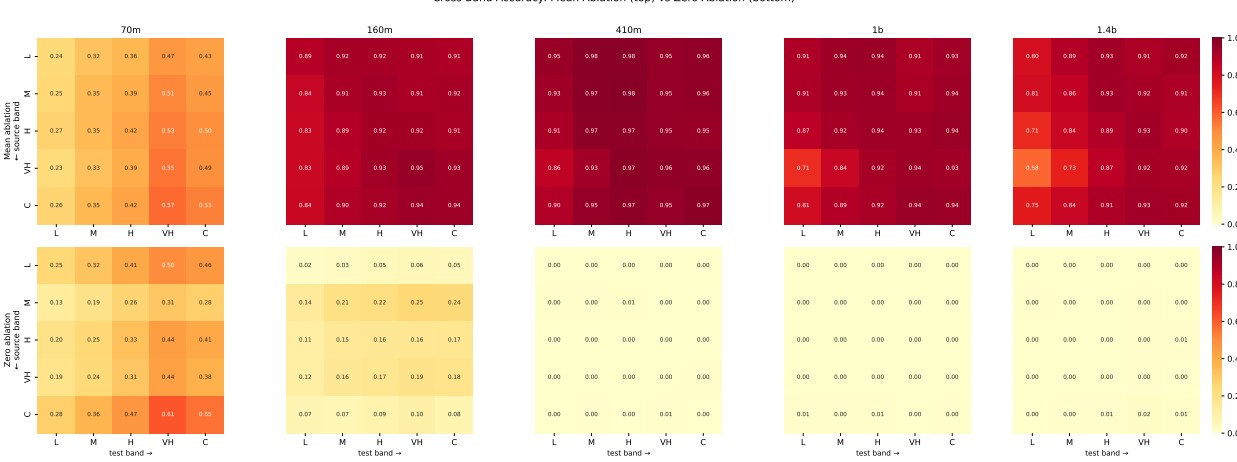

Figure 53: Cross-band accuracy under resample (top) vs. zero ablation (bottom). The same-band advantage vanishes; models ≥410M collapse uniformly.

# I Pipeline Positive Control: Within-Task Mechanistic Contrast

The layer-sweep positive control (Appendix F.9.1) validates interchange patching sensitivity but does not validate the full circuit-comparison pipeline: ACDC extraction → structural comparison → cross-condition transfer testing. To test whether the pipeline can detect genuine within-task mechanistic differences, we attempted two controlled variants of LSC on Pythia-160m, both preserving the task family while altering computational requirements.

## I.1 Attempt 1: Reverse-Copy LSC

We placed the target token $T$ *before* the source prefix:

$$\text{Standard:} \quad S_1\ S_2\ S_3\ S_4\ S_5\ \mathbf{T}\ R_1\ldots R_{10}\ S_1\ S_2\ S_3\ S_4\ S_5\ \rightarrow\ \text{predict } T$$
$$\text{Reverse:} \quad \mathbf{T}\ S_1\ S_2\ S_3\ S_4\ S_5\ R_1\ldots R_{10}\ S_1\ S_2\ S_3\ S_4\ S_5\ \rightarrow\ \text{predict } T$$

Standard induction copies from offset $+1$; reverse-copy requires offset $-5$. Pythia-160m achieves **0.0% top-1 accuracy** (standard LSC: 98.7%), predicting $R_1$ (the offset $+1$ token) in 94.2% of cases, confirming its induction heads are exclusively wired for forward copy. Since the model cannot solve this variant, no meaningful circuit can be extracted and it cannot serve as a positive control.

## I.2 Attempt 2: Zero-Distractor LSC

We removed all distractor tokens, reducing the sequence from 21 to 11 tokens:

$$\text{Standard (21 tokens):} \quad S_1\ S_2\ S_3\ S_4\ S_5\ \mathbf{T}\ R_1\ldots R_{10}\ S_1\ S_2\ S_3\ S_4\ S_5$$
$$\text{Zero-distractor (11 tokens):} \quad S_1\ S_2\ S_3\ S_4\ S_5\ \mathbf{T}\ S_1\ S_2\ S_3\ S_4\ S_5$$

The motivation is that standard LSC requires compositional induction to bridge the 15-position gap between $T$ and the prediction point, whereas zero-distractor places $T$ only 5 positions away, potentially enabling a single direct-copy head without previous-token composition. Pythia-160m solves zero-distractor LSC at 88.4% accuracy (vs. 98.7% standard).

We ran ACDC at the same threshold ($\tau^* = 6.31 \times 10^{-4}$) and computed a $2 \times 2$ cross-condition transfer matrix (Table 62). The transfer matrix is uniform rather than block-diagonal: cross-condition transfer efficiency is 96.7%, *exceeding* the cross-frequency-band efficiency of 92.6% (Table 2). Jaccard similarity between the two circuits is 0.539, *below* the between-band Jaccard for Pythia-160m (0.557; Table 27): structurally more

divergent, yet functionally more interchangeable, the defining signature of phantom specialization, now demonstrated on a second axis of variation. The standard circuit outperforms on both test conditions, consistent with the asymmetric transfer pattern (Section 4.5): the circuit extracted under a stronger signal (98.7% base accuracy) generalizes better than the reverse.

Table 62: Cross-condition transfer matrix for Pythia-160m. Each cell shows edge-level circuit accuracy. The standard circuit outperforms on both conditions because it was extracted under higher base accuracy (98.7% vs. 88.4%).

|  | Zero-distractor test | Standard test |
| --- | --- | --- |
| Zero-distractor circuit (1,498 edges) | 82.7% | 80.4% |
| Standard circuit (1,392 edges) | 88.9% | 92.4% |
| Base model | 88.4% | 98.7% |

### I.3 Interpretation

Both attempts confirm that LSC is a single-mechanism task on Pythia: forward copy via induction heads is the only available mechanism (reverse-copy fails completely) and is robust to input structure (zero-distractor uses the same circuit). This single-mechanism nature precludes constructing a within-task positive control for the full pipeline.

This does not undermine the paper's conclusions. Indirect evidence for pipeline sensitivity is substantial: the layer-sweep control demonstrates full dynamic range (Appendix F.9.1); EAP-IG converges on the same null result with largely different edge sets (Appendix C.9); zero ablation confirms the pattern under a stricter baseline (Appendix H.12); and per-example agreement shows no systematic divergence (Section 6.1). The zero-distractor experiment provides independent confirmation: phantom specialization replicates on a second perturbation axis, with structurally more divergent circuits (Jaccard 0.539 vs. 0.557) yet higher functional interchangeability (96.7% vs. 92.6%).

