# OpenReview forum: "Many Circuits, One Mechanism: Input Variation and Evaluation Granularity in Circuit Discovery"
_TMLR — Under review for TMLR_

### Review · Reviewer_DeJu · 2026-06-23

**Summary Of Contributions:**

This paper studies whether structurally different discovered circuits actually imply different mechanisms. Using Literal Sequence Copying across token-frequency bands in Pythia models, the authors show that ACDC recovers structurally different circuits that are largely functionally interchangeable. The core message that structural circuit differences alone are insufficient evidence for mechanistic specialization is important and well motivated.

**Audience:**

Yes

**Audience Explanation:**

The question is timely and directly relevant to current circuit-discovery practice. The paper uses multiple complementary tests: structural overlap, cross-band transfer, shared-core analysis, representational similarity, and causal interchange interventions.

**Claims And Evidence:**

Yes

**Claims Explanation:**

The experiments are extensive, with five Pythia scales, repeated extractions, threshold checks, and cross-method comparisons.

**Requested Changes:**

1. The empirical scope is still narrow. The main evidence comes from one task family, Literal Sequence Copying, and one model family, Pythia. It remains unclear whether phantom specialization holds for semantic tasks, reasoning tasks, or other architectures.
2. The positive control is weak. The paper attempts alternative LSC variants, but these do not yield a genuine case where the pipeline detects true specialization. Thus, it is hard to assess whether the proposed tests would reliably distinguish phantom from real specialization.
3. The conclusions depend heavily on ACDC and edge-level resample ablation. Although EAP-IG and zero-ablation checks are included, EAP-IG circuits are often much less faithful, and zero ablation collapses performance in larger models.
4. Token frequency is carefully controlled, but residual confounds may remain, including embedding geometry, pretraining co-occurrence structure, and token-level semantic regularities.
5. The related-work discussion around clustering circuit families and robustness of such groupings could be strengthened. If the authors frame prompt/circuit families as clusters, they should discuss robustness issues in clustering-based analysis, e.g., On the Adversarial Robustness of Multi-Kernel Clustering.

---

### Review · Reviewer_guWx · 2026-07-12

**Summary Of Contributions:**

This paper studies whether circuit discovery recovers different circuits when only the input statistics change while the task is held fixed. The idea of the paper is to use token frequency as the input variation: the authors take a non-semantic copying task (Literal Sequence Copying), split the Pythia vocabulary into token-frequency bands (based on how often each token appears in The Pile, the training corpus of Pythia), and extract circuits with ACDC separately for each band, across five Pythia models (70M to 1.4B). The paper found that the circuits change across bands (ie, the components/edges change), but the function of these components does not. More specifically, the circuits are structurally distinct (low-frequency circuits are systematically larger), but this structural difference does not correspond to functional differences. The authors call this pattern "phantom specialization": the circuits look specialized for each band, but they are a product of the algorithm picking one of many redundant paths that implement the same computation (in my view, this is related to the well known “overdetermination” problem of counterfactual-based methods [3]).


Moreover, the authors show that the way circuits are usually evaluated (keeping all outgoing edges of any node that participates in the circuit) lets information flow through paths that the discovery algorithm never selected, which makes circuits look much more faithful than they actually are (simply because more edges are used). The “phantom specialization” effect appears when they evaluate only the specific edges that the algorithm selected. Based on this, the paper derives practical recommendations for the community, including the use edge-level evaluation as the primary faithfulness metric.

## Strengths
- The experimental setup is very broad and seems robust. The paper performs hyperparameter sweeps in a control dataset (Section 3.3), perform confound controls (Section 3.1), has random baselines (Section 3.3), and also leverage multiple methods from the literature (ACDC, EAP/EAP-IG for circuit discovery; patching and DAS for establishing causality). Because of this robust experimental setup, the main conclusion is supported by several lines of evidence using different methodology, which makes the results stronger.
- The related work section is in-depth, citing most of the recent work in the area and explaining it mostly well.
- The methodological recommendations are concrete and directly usable by the community (Section 6.2).

## Weaknesses:

The main weakness of the paper is its presentation: the paper is poorly written and needs a heavy rewriting and a more unified storyline to be publishable. I also have concerns about specific claims of the paper (the exclusion of the very_low band, the position aggregation in the process of finding circuits, the unspecified head-role classification, and the zero-ablation experiment). I detail all of these in the next section.

**Audience:**

Yes

**Audience Explanation:**

Yes. The paper touches very important problems in interpretability research (and specifically circuit discovery), such as non-identifiability (two different circuits explaining the same outcome) and overdetermination, while giving practical recommendations on how to handle them while running circuit discovery methods.

**Broader Impact Concerns:**

I have no broader impact concerns, and in my opinion, no Broader Impact Statement is required for this paper.

**Claims And Evidence:**

No

**Claims Explanation:**

- The paper is poorly written. More specifically:
    - Many terms are used before being defined (e.g. Section 5.4, CKA). Another example is "frequency bands", which is used from the abstract without a definition, which is extra confusing because of RoPE frequency bands.
    - A lot of results appear only in the appendix, with the main text referencing them poorly (e.g. direct references to the Appendix such as (subsection B.3) without giving a good description of what is actually presented there).
    - Important results for the paper (e.g., Sections 4.4-4.5) have no supporting figure or table in the main text, which makes the results hard to understand and compare. For instance, in Section 4.4, the results are explained in plain text using numbers in a paragraph. By reading it, I cannot make comparisons between any models, and therefore, I cannot get any useful takeaway from these numbers. This is repeated across the paper, which makes the paper hard to read.
    - The storyline of the paper, especially on the results, is not good. While reading Sections 4 and 5, it’s hard to see how each of these experiments connects with the big picture/main message of the paper. The paper lacks of sentences such this one on Section 5.3.3: “Logit lens analysis reveals that the base model’s output distribution converges to the final prediction later for low-frequency tokens”, which are effective on connecting the experiment result with the actual message of the paper.
- In my opinion, the very_low token frequency band is excluded from the experiments without a convincing explanation (97 tokens is a ok number of tokens to work with). The rare tokens is the band which functional specialization is more likely since these tokens almost never appear. Notice that, as a contrast, the paper report results for Pythia-70M even though the model performance is not good at the task. So, I expect the same for very_low band, even with the sample size limitation.
- The circuits are position-aggregated (ie, the causal effects are computed over all the tokens of the input), and the paper never discusses the implications of it. For instance, it’s know that position matters and position invariant circuit finding leads to low precision and recall [1].
- The way that the paper finds the role of attention heads for different models is not specified. I don’t know if the authors manually computed scores to find them (e.g. copy score, induction score) or if they use something documented in the literature (e.g. [2] documents functional roles of attention heads for Pythia 160M).
- A zero-ablation experiment is used by the paper to dismiss the statistically significant same-band advantage found in Section 5.1.1. However, zero ablation is known to bring the model out of distribution, and in this case it destroys the task performance of the Pythia models >= 410M: their accuracy drops to near chance on every band (Table 61). A circuit that cannot do the task on any band cannot show an advantage on one band, so for these models, there is nothing that can be concluded about the same-band advantage. This experiment is more informative for Pythia-160M, where the accuracy survives the zero ablation, and then the same-band advantage disappears.
- The main claim of the paper is a null result (no functional specialization found), and the pipeline was never shown to detect genuine specialization when it exists: both attempts of a positive control failed. The authors acknowledge this in Section 6.3, but this limitation should also be upfront in the paper, since it changes how the main claim should be read.

In general, I think that the central findings of the paper are interesting. If the critical items described in the Requested Changes are addressed, I expect to change this answer to Yes.

[1]: Haklay, Tal, et al. "Position-aware automatic circuit discovery." Proceedings of the 63rd Annual Meeting of the Association for Computational Linguistics (Volume 1: Long Papers). 2025.
[2]: Tigges, Curt, et al. "LLM circuit analyses are consistent across training and scale." Advances in Neural Information Processing Systems 37 (2024): 40699-40731.
[3]: Mueller, Aaron. "Missed causes and ambiguous effects: Counterfactuals pose challenges for interpreting neural networks." arXiv preprint arXiv:2407.04690 (2024).

**Requested Changes:**

## Critical changes
- Perform a major re-writing of the paper, with a unified storyline, especially on Sections 3-5. Most of the points to be improved were mentioned in the review, but something that I recommend is to restructure the results around the three or four major findings, and leave many small results to appendix summaries.
- Specify how the head roles were found or cite the source if prior work was used (e.g. [2]).
- Fix the zero-ablation claims. Acknowledge that this experiment is uninformative for the models which accuracy collapses to near chance. Another option is to replace it with a corruption method that keeps the models functional, such as mean ablation.
- Run the extraction on the very_low frequency tokens to see what the results of the functional specialization are for these circuits. Since the sample size may be small, this can be deferred to the appendix, and the main text can refer to it as a case for further future work.
- Say explicitly in Section 3.3 that the circuits are aggregated by position, and either run a position-aware variant on at least one model or add a limitation paragraph discussing how the aggregation could bias the results.
- Say upfront in the paper that the claim supported by the results is that no functional specialization was detectable at this granularity and sensitivity, not that it doesn’t exists in the model.

## Changes that would strengthen the work

- Connect the discussion in Section 6.1 to the overdetermination problem of counterfactual-based methods [3], which is very related to the proposed mechanism behind phantom specialization.
- Add a few-shot variant of the task, to see if the ICL can “reduce” the influence of the token frequency in the circuit finding.
- Discuss in more depth the difference between structure and function. These concepts may be mixed sometimes, and it’s good to clarify their difference. For instance, [4] find that structurally identical edges can still carry different signals, even considering the same token role, but still perform the same task in the end (e.g. moving a name).

[4]: Franco, Gabriel, et al. "Finding Interpretable Prompt-Specific Circuits in Language Models." arXiv preprint arXiv:2602.13483 (2026).

---

### Review · Reviewer_erZU · 2026-07-13

**Summary Of Contributions:**

In this work, the authors test whether in mechanistic interpretability, structural differences between discovered circuits can be interpreted as evidence of distinct underlying mechanisms. They fix a task (Literal Sequence Copying) and vary input statistics by drawing from four different token frequency bands from the Pile, controlled for frequency. They extract 75 circuits from five Pythia models (with size ranging from 70M to 1.4B parameters) using ACDC across five conditions and three independent draws.
The authors find that circuits are structurally distinct. (low-frequency circuits are larger, Jaccard overlap within a band is larger than inter-band, and band-specific edges are 5-263x more common than a noise null would predict) but functionally interchangeable: (i) band-specific edges transfer across bands 81-97% of the time (ii) edges that exist in 3+ conditions almost fully recover the accuracy of the full circuit (98-99%) (iii) cross-draw transfer ratios are ~1 even with low edge overlap (iv) interchange interventions on the residual stream yield IIA > 0.95 (v) boundless DAS finds that the band identity in <6% of the model's dimensions.
The authors call this asymmetrical finding "phantom specialization", and analyze two additional methodological issues: source-level evaluation inflates apparent circuit accuracy relative to edge-level evaluation, and the model scale is the dominant source of variance for almost all metrics, creating a Simpson's paradox in 52/80 structure-function and 68/80 structure-representation metric pairs. The authors conduct a zero-ablation study and determine that the residual same-band advantage is attributable to the corruption procedure.
The authors conclude with practical recommendations for researchers, such as cross-condition transfer tests, edge-level evaluation rather than source-level evaluation as the primary metric, multiple extractions with majority-vote aggregation and per-model statistics.

Key strengths:
- The experimental setup section is extremely rich, large, deep and statistically sound, which is visible from the size and extensiveness of the appendix. Some examples: the authors select a threshold once on the control band and then apply it uniformly, they apply many non-parametric tests with FDR corrections, report bootstrap confidence intervals, explicit power analyses...
- The amount of evidence across all four axes (structural, functional, representational and causal) is staggering, making the authors' claims very convincing
- The authors report not only successful experiments but also failed positive controls (e.g. in appendix I)
- The paper gives practical and actionable recommendations (e.g. source-level rather than edge-level)
- The authors replicate their core finding on a second axis (zero-distractor)

Key weaknesses:
- The first appendix shows that LSC is a single mechanism task on the studied Pythia models. As such, the functional-equivalence null is not very surprising
- The authors do not show examples where their pipeline can detect actual specialization when it is present
- The paper's presentation highly depends on the appendix and is quite hard to read as a result

**Additional Comments:**

It must be emphasized that the level of experimental detail and (honest) statistical reporting in this paper is excellent. This type of reporting is extremely useful to the community, yet it is *extremely* rare to find such in-depth studies. I commend the authors for their methodological hygiene.

**Audience:**

Yes

**Audience Explanation:**

The authors' findings have direct consequences on the standard practices used by the mechanistic interpretability community (part of TMLR's audience). In particular, many existing studies use single extractions, source-level evaluation, and structural overlap comparisons across conditions or models. This paper gives very clear reasons why those practices should not be trusted and proposes concrete and cheap alternatives to those.

**Claims And Evidence:**

Yes

**Claims Explanation:**

The central claims stated by the authors are overwhelmingly well-supported within the declared scope. For the main claim (structural divergence between discovered circuits is not sufficient evidence of distinct mechanisms (stated in 6.3, 7)), this is supported by the transfer efficiency of band-specific edges (table 2), shared-core sufficiency (fig 6), near perfect cross-draw transfer with low edge overlap (table 4), intervention interchange results (5.3.2), zero-ablation diagnostic to counteract the main counter-signal. The source-level vs edge-level gap and the Simpson's paradox analyses are convincing. The limitations section clearly mentions the missing positive control and the restriction to a single task and model family.

**Requested Changes:**

No critical changes required.

Strengthening changes:
- Having a positive control on a known multi mechanism setting would be beneficial in showing that the paper can detect actual specialization when this exists. This could be done on some of the stated tasks (factual recall vs counterfactual competition, clock vs. pizza), but I understand that this is expensive.
- In 6.1, the authors suggest that band-specific edges are near the pruning threshold and may be selected inconsistently with small input shifts. This could be tested with existing data, e.g. by checking if the patching score distributions of band-specific edges clusters near the threshold while those of universal edges do not.
- Some tables appear to be duplicated in the appendix, which should be avoided considering the size of the paper (Tables 2-49, 4-31, 5-56, 6-46, 7-51, 8-52, 10-62). In general, the main text references the appendix very heavily, considering that some parts contain rather importance evidence (e.g. C9, F9, H1, H2, H12). I recommend removing the duplicates and moving the cross-method comparison into the main text, perhaps by shortening section 4 instead.

---

### Comment · Action_Editor_ZvUz · 2026-05-29

Dear Authors,

I apologize for the delay in getting your paper under review. I have been having trouble finding reviewers that are suitable and available. I had hoped that after the NeurIPS deadline the number of available TMLR reviewers in the pool would increase. But that has not happened. Because of this, I was wondering if you could help by providing a list of a few reviewers who you think would be appropriate.

Thank you for your help.

Best,

AE